# Achieving Near-Optimal Convergence for Distributed Minimax Optimization with Adaptive Stepsizes

**Yan Huang**
College of Control Science and Engineering
Zhejiang University, China
huangyan5616@zju.edu.cn

**Xiang Li**
Department of Computer Science
ETH Zurich, Switzerland
xiang.li@inf.ethz.ch

**Yipeng Shen**
College of Control Science and Engineering
Zhejiang University, China
22332074@zju.edu.cn

**Niao He**
Department of Computer Science
ETH Zurich, Switzerland
niao.he@inf.ethz.ch

**Jinming Xu**
College of Control Science and Engineering
Zhejiang University, China
jimmyxu@zju.edu.cn

## Abstract

In this paper, we show that applying adaptive methods directly to distributed minimax problems can result in non-convergence due to inconsistency in locally computed adaptive stepsizes. To address this challenge, we propose D-AdaST, a Distributed Adaptive minimax method with Stepsize Tracking. The key strategy is to employ an adaptive stepsize tracking protocol involving the transmission of two extra (scalar) variables. This protocol ensures the consistency among stepsizes of nodes, eliminating the steady-state error due to the lack of coordination of stepsizes among nodes that commonly exists in vanilla distributed adaptive methods, and thus guarantees exact convergence. For nonconvex-strongly-concave distributed minimax problems, we characterize the specific transient times that ensure time-scale separation of stepsizes and quasi-independence of networks, leading to a near-optimal convergence rate of $\tilde{\mathcal{O}}\left(\epsilon^{-(4+\delta)}\right)$ for any small $\delta > 0$, matching that of the centralized counterpart. To our best knowledge, D-AdaST is the *first* distributed adaptive method achieving near-optimal convergence without knowing any problem-dependent parameters for nonconvex minimax problems. Extensive experiments are conducted to validate our theoretical results.

## 1 Introduction

Distributed optimization has seen significant research progress over the last decade, resulting in numerous algorithms (Nedic and Ozdaglar, 2009; Yuan et al., 2016; Lian et al., 2017; Pu and Nedić, 2021). However, the traditional focus of distributed optimization has primarily been on minimization tasks. With the rapid growth of machine learning research, various applications have emerged that go beyond simple minimization, such as Generative Adversarial Networks (GANs) (Goodfellow et al., 2014; Gulrajani et al., 2017), robust optimization (Mohri et al., 2019; Sinha et al., 2017), adversary training of neural networks (Wang et al., 2021), fair machine learning (Madras et al., 2018), and just

38th Conference on Neural Information Processing Systems (NeurIPS 2024).

to name a few. These tasks typically involve a minimax structure as follows:

$$\min_{x \in \mathcal{X}} \max_{y \in \mathcal{Y}} f(x, y),$$

where $\mathcal{X} \subseteq \mathbb{R}^p$, $\mathcal{Y} \subseteq \mathbb{R}^d$, and $x, y$ are the primal and dual variables to be learned, respectively. One of the simplest yet effective methods for solving the above minimax problem is Gradient Descent Ascent (GDA) (Dem'yanov and Pevnyi, 1972; Nemirovski et al., 2009) which alternately performs stochastic gradient descent for the primal variable and stochastic gradient ascent for the dual variable. This approach has demonstrated its effectiveness in solving minimax problems, especially for convex-concave objectives (Hsieh et al., 2021; Daskalakis et al., 2021; Antonakopoulos et al., 2021), i.e., the function $f(\cdot, y)$ is convex for any $y \in \mathcal{Y}$, and $f(x, \cdot)$ is concave for any $x \in \mathcal{X}$.

Adaptive gradient methods, such as AdaGrad (Duchi et al., 2011), Adam (Kingma and Ba, 2014), and AMSGrad (Reddi et al., 2018), are often integrated with GDA to effectively solve minimax problems with theoretical guarantees in convex-concave settings (Diakonikolas, 2020; Antonakopoulos et al., 2021; Ene and Lê Nguyen, 2022). These adaptive methods are capable of adjusting stepsizes based on historical gradient information, making it robust to hyper-parameters tuning and can converge without requiring to know problem-dependent parameters (a characteristic often referred to as being "parameter-agnostic"). However, in the nonconvex regime, it has been shown by Lin et al. (2020); Yang et al. (2022b) that it is necessary to have a time-scale separation in stepsizes between the minimization and maximization processes to ensure the convergence of GDA and GDA-based adaptive algorithms. In particular, the stepsize ratio between primal and dual variables needs to be smaller than a threshold depending on the properties of the problem such as the smoothness and strong-concavity parameters (Li et al., 2022; Guo et al., 2021; Huang et al., 2021), which are often unknown or difficult to estimate in real-world tasks, such as training deep neural networks.

Applying GDA-based adaptive methods into decentralized settings poses additional challenges due to the presence of inconsistency in locally computed adaptive stepsizes. In particular, it has been shown that the inconsistency of stepsizes can result in non-convergence in federated learning with heterogeneous computation speeds (Wang et al., 2020; Sharma et al., 2023). This is mainly due to the lack of a central node coordinating the stepsizes of nodes in distributed settings, making it difficult to converge, as observed in minimization problems (Liggett, 2022; Chen et al., 2023b). As a result, the following question arises naturally:

*"Can we design an adaptive minimax method that ensures the time-scale separation and consistency of stepsizes with provable convergence in fully distributed settings?"*

**Contributions.** In this paper, we aim to propose a distributed adaptive method for efficiently solving nonconvex-strongly-concave (NC-SC) minimax problems. The contributions are threefold:

- We construct counterexamples showing that directly applying adaptive methods designed for centralized problems will lead to inconsistencies in locally computed adaptive stepsizes, resulting in non-convergence in distributed settings. To tackle this issue, we propose the *first* distributed adaptive minimax method, named D-AdaST, that incorporates an efficient stepsize tracking mechanism to maintain consistency across local stepsizes, which involves transmission of merely two extra (scalar) variables. The proposed algorithm exhibits time-scale separation in stepsizes and parameter-agnostic capability in fully distributed settings.

- Theoretically, we prove that D-AdaST is able to achieve a near-optimal convergence rate of $\tilde{\mathcal{O}}\left(\epsilon^{-(4+\delta)}\right)$ with arbitrarily small $\delta > 0$ to find an $\epsilon$-stationary point for distributed NC-SC minimax problems. In contrast, we also prove the existence of a constant steady-state error in both the lower and upper bounds for GDA-based distributed minimax algorithms when being directly integrated with the adaptive stepsize rule without the stepsize tracking mechanism. Moreover, we explicitly characterize the transient times that ensure time-scale separation and quasi-independence of network, respectively.

- We conduct extensive experiments on real-world datasets to verify our theoretical findings and the effectiveness of D-AdaST on a variety of tasks, including robust training of neural networks and optimizing Wasserstein GANs. In all tasks, we demonstrate the superiority of D-AdaST over several vanilla distributed adaptive methods across various graphs, initial stepsizes and data distributions (see also additional experiments in Appendix A).

## 1.1 Related Works

**Distributed nonconvex minimax methods.** In the realm of federated learning, Deng and Mahdavi (2021) introduce Local SGDA algorithm combining FedAvg/Local SGD with stochastic GDA and show an $\tilde{\mathcal{O}}\left(\epsilon^{-6}\right)$ sample complexity for NC-SC objective functions. Sharma et al. (2022) provide improved complexity result of $\tilde{\mathcal{O}}\left(\epsilon^{-4}\right)$ matching that of the lower bound of first-order algorithms for both NC-SC and nonconvex-Polyak-Lojasiewicz (NC-PL) settings (Li et al., 2021; Zhang et al., 2021a) . Yang et al. (2022a) integrate Local SGDA with stochastic gradient estimators to eliminate the data heterogeneity. More recently, Zhang et al. (2023) adopt compressed momentum methods with Local SGD to increase the communication efficiency of the algorithm. For decentralized nonconvex minimax problems, Liu et al. (2020) study the training of GANs using decentralized optimistic stochastic gradient and provide non-asymptotic convergence with fixed stepsizes. Tsaknakis et al. (2020) propose a double-loop decentralized SGDA algorithm with gradient tracking techniques (Pu and Nedić, 2021) and achieve $\tilde{\mathcal{O}}\left(\epsilon^{-4}\right)$ sample complexity. With a stronger assumption of average smoothness, some studies employ variance reduction techniques to accelerate convergence (Zhang et al., 2021b; Chen et al., 2022; Xian et al., 2021; Tarzanagh et al., 2022; Wu et al., 2023; Chen et al., 2024; Zhang et al., 2024), which require more memory and computational resources due to the need for larger batch-sizes or full gradient evaluations. However, all the above-mentioned methods use a fixed or uniformly decaying stepsize, requiring the prior knowledge of smoothness and concavity.

**(Distributed) adaptive minimax methods.** For centralized nonconvex minimax problems, Yang et al. (2022b) show that, even in deterministic settings, GDA-based methods necessitate the time-scale separation of the stepsizes for primal and dual updates. Many attempts have been made for ensuring the time-scale separation requirement (Lin et al., 2020; Yang et al., 2022c; Boţ and Böhm, 2023; Huang et al., 2023). However, these methods typically come with the prerequisite of having knowledge about problem-dependent parameters, which can be a significant drawback in practical scenarios. To this end, Yang et al. (2022b) introduce a nested adaptive algorithm named NeAda that achieves parameter-agnosticism by incorporating an inner loop to effectively maximize the dual variable, which can obtain an optimal sample complexity of $\tilde{\mathcal{O}}\left(\epsilon^{-4}\right)$ when the strong-concavity parameter is known. More recently, Li et al. (2023) introduce TiAda, a single-loop parameter-agnostic adaptive algorithm for nonconvex minimax optimization which employs separated exponential factors on the adaptive primal and dual stepsizes, improving upon NeAda on the noise-adaptivity. There has been few works dedicated to adaptive minimax optimization in federated learning settings. For instance, Huang et al. (2024) introduces a federated adaptive algorithm that integrates the stepsize rule of Adam with full-client participation, resembling the centralized counterpart. Ju et al. (2023) study a federated Adam algorithm for fair federated learning where the objective function is properly weighted to account for heterogeneous updates among nodes. To the best of our knowledge, it is still unknown how one can design an adaptive minimax method capable of fulfilling the time-scale separation requirement and being parameter-agnostic in *fully distributed settings*.

**Notations.** Throughout this paper, we denote by $\mathbb{E}\left[\cdot\right]$ the expectation of a random variable, $\|\cdot\|$ the Frobenius norm, $\langle\cdot,\cdot\rangle$ the inner product of two vectors, $\odot$ the Hadamard product (entry wise), $\otimes$ the Kronecker product. We denote by $\mathbf{1}$ the all-ones vector, $\mathbf{I}$ the identity matrix and $\mathbf{J} = \mathbf{1}\mathbf{1}^T/n$ the averaging matrix with $n$ dimension. For a vector or matrix $A$ and constant $\alpha$, we denote $A^\alpha$ the entry-wise exponential operations. We denote $\Phi\left(x\right) := f\left(x, y^*\left(x\right)\right)$ as the primal function where $y^*\left(x\right) = \underset{y \in \mathcal{Y}}{\arg\max} f\left(x, y\right)$, and $\mathcal{P}_\mathcal{Y}\left(\cdot\right)$ as the projection operation onto set $\mathcal{Y}$.

## 2 Distributed Adaptive Minimax Methods

We consider the distributed minimax problem collaboratively solved by a set of agents over a network. The overall objective of the agents is to solve the following finite-sum problem:

$$\min_{x \in \mathbb{R}^p} \max_{y \in \mathcal{Y}} f\left(x, y\right) = \frac{1}{n} \sum_{i=1}^{n} \underbrace{\mathbb{E}_{\xi_i \sim \mathcal{D}_i}\left[F_i\left(x, y; \xi_i\right)\right]}_{:= f_i(x,y)}, \tag{1}$$

where $f_i : \mathbb{R}^{p+d} \to \mathbb{R}$ is the local private loss function accessible only by the associated node $i \in \mathcal{N} = \{1, 2, \cdots, n\}$, $\mathcal{Y} \subset \mathbb{R}^d$ is closed and convex, and $\xi_i \sim \mathcal{D}_i$ denotes the data sample locally stored at node $i \in \mathcal{N}$ with distribution $\mathcal{D}_i$. We consider a graph $\mathcal{G} = (\mathcal{V}, \mathcal{E})$, here, $\mathcal{V} = \{1, 2, ..., n\}$ represents the set of agents, and $\mathcal{E} \subseteq \mathcal{V} \times \mathcal{V}$ denotes the set of edges consisting of ordered pairs $(i, j)$

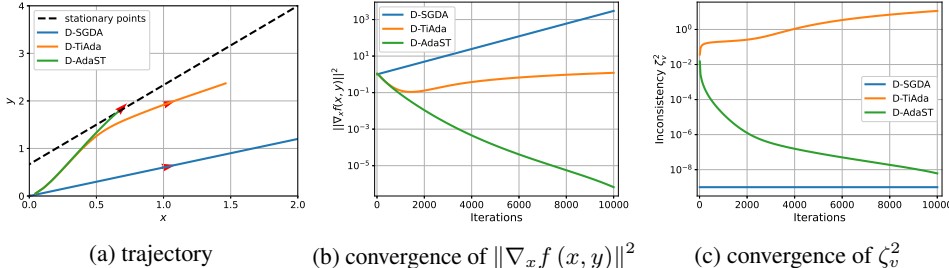

| (a) trajectory | (b) convergence of $\|\nabla_x f(x,y)\|^2$ | (c) convergence of $\zeta_v^2$ |

Figure 1: Comparison among D-SGDA, D-TiAda and D-AdaST for NC-SC quadratic objective function (6) with $n = 2$ nodes and $\gamma_x = \gamma_y$. In (a), it shows the trajectories of primal and dual variables of the algorithms, the points on the black dash line are stationary points of $f$. In (b), it shows the convergence of $\|\nabla_x f(x_k, y_k)\|^2$ over the iterations. In (c), it shows the convergence of the inconsistency of stepsizes, $\zeta_v^2$ defined in (8), over the iterations. Notably, $\zeta_v^2$ fails to converge for D-TiAda and $\zeta_v^2 = 0$ for non-adaptive D-SGDA.

representing the communication link from node $j$ to node $i$. For node $i$, we define $\mathcal{N}_i = \{j \mid (i, j) \in \mathcal{E}\}$ as the set of its neighboring nodes. Before proceeding to the discussion of distributed algorithms, we first introduce the following notations for brevity:

$$\mathbf{x}_k := [x_{1,k}, x_{2,k}, \cdots, x_{n,k}]^T \in \mathbb{R}^{n \times p}, \ \mathbf{y}_k := [y_{1,k}, y_{2,k}, \cdots, y_{n,k}]^T \in \mathbb{R}^{n \times d},$$

where $x_{i,k} \in \mathbb{R}^p, y_{i,k} \in \mathcal{Y}$ denote the primal and dual variable of node $i$ at each iteration $k$, and

$$\nabla_x F(\mathbf{x}_k, \mathbf{y}_k; \xi_k^x) := \left[\cdots, \nabla_x F_i(x_{i,k}, y_{i,k}; \xi_{i,k}^x), \cdots\right]^T,$$
$$\nabla_y F(\mathbf{x}_k, \mathbf{y}_k; \xi_k^y) := \left[\cdots, \nabla_y F_i(x_{i,k}, y_{i,k}; \xi_{i,k}^y), \cdots\right]^T$$

are the corresponding partial stochastic gradients with i.i.d. samples $\xi_k^x, \xi_k^y$ in a compact form.

Next, we will first explain the pitfalls of directly applying centralized adaptive stepsize rules to decentralized settings, and then introduce our newly proposed solution to address the challenge.

## 2.1 Non-Convergence of Direct Extensions

For the distributed minimax optimization problem as depicted in (1) involving NC-SC objective functions, we will show shortly that the Distributed Stochastic Gradient Descent Ascent (D-SGDA) method may not converge due to the inability of time-scale separation with constant stepsizes (c.f., Figure 1), which is also observed in centralized settings (Lin et al., 2020; Yang et al., 2022b). To address this issue, one can adopt the adaptive stepsize rule used in centralized TiAda (Li et al., 2023) for each individual node, which is renowned for its ability to adaptively fulfill the time-scale separation requirements. As a result, we arrive at the following Distributed TiAda (D-TiAda) algorithm.

$$\mathbf{x}_{k+1} = W\left(\mathbf{x}_k - \gamma_x V_{k+1}^{-\alpha} \nabla_x F(\mathbf{x}_k, \mathbf{y}_k; \xi_k^x)\right), \tag{2a}$$

$$\mathbf{y}_{k+1} = \mathcal{P}_{\mathcal{Y}}\left(W\left(\mathbf{y}_k + \gamma_y U_{k+1}^{-\beta} \nabla_y F(\mathbf{x}_k, \mathbf{y}_k; \xi_k^y)\right)\right), \tag{2b}$$

where $\gamma_x$ and $\gamma_y$ are the stepsizes, $W$ is a doubly-stochastic weight matrix induced by graph $\mathcal{G}$ (Xiao et al., 2006) (c.f., Assumption 4), and

$$V_{k+1}^{-\alpha} = \text{diag}\left\{v_{i,k+1}^{-\alpha}\right\}_{i=1}^n, \quad U_{k+1}^{-\beta} = \text{diag}\left\{u_{i,k+1}^{-\beta}\right\}_{i=1}^n, \tag{3}$$

with $v_{i,k+1} = \max\left\{m_{i,k+1}^x, m_{i,k+1}^y\right\}, u_{i,k+1} = m_{i,k+1}^y$, and

$$m_{i,k+1}^x = m_{i,k}^x + \left\|\nabla_x F_i(x_{i,k}, y_{i,k}; \xi_{i,k}^x)\right\|^2, \ m_{i,k+1}^y = m_{i,k}^y + \left\|\nabla_y F_i(x_{i,k}, y_{i,k}; \xi_{i,k}^y)\right\|^2 \tag{4}$$

are the local accumulated gradient norm. Note that we impose a maximum operator in the preconditioner $v_{i,k}$, and employ different stepsize decaying rates, i.e., $0 < \beta < \alpha < 1$, for the primal and

dual variables, respectively. Such design allows to balance the updates of $x$ and $y$, and achieves the desired time-scale separation without requiring any knowledge of parameters (Li et al., 2023).

However, in the distributed setting, such direct extension may fail to converge to a stationary point because $v_{i,k}$ and $u_{i,k}$ can be inconsistent due to the difference of local objective functions $f_i$, In particular, we can rewrite the above vanilla distributed optimization algorithm (2) in the sense of average system of primal variables as below,

$$\bar{x}_{k+1} = \underbrace{\bar{x}_k - \gamma_x \bar{v}_k^{-\alpha} \frac{\mathbf{1}^T}{n} \nabla_x F(\mathbf{x}_k, \mathbf{y}_k; \xi_k^x)}_{\text{adaptive descent}} - \underbrace{\gamma_x \frac{\left(\tilde{\boldsymbol{v}}_{k+1}^{-\alpha}\right)^T}{n} \nabla_x F(\mathbf{x}_k, \mathbf{y}_k; \xi_k^x)}_{\text{inconsistancy}}, \tag{5}$$

where $\left(\tilde{\boldsymbol{v}}_k^{-\alpha}\right)^T := \left[\cdots, v_{i,k}^{-\alpha} - \bar{v}_k^{-\alpha}, \cdots\right]$, $\bar{x}_k := \mathbf{1}^T \mathbf{x}_k / n$ and $\bar{v}_k := 1/n \sum_{i=1}^n v_{i,k}$.

It is evident that, in comparison to centralized adaptive methods, an unexpected term (i.e., $\tilde{\boldsymbol{v}}_k$) on the right-hand side (RHS) arises due to inconsistencies. This term introduces inaccuracies in the directions of gradient descent, degrading the optimization performance. The theorem presented below reveals a gap near the stationary points in a properly designed counterexample, indicating the non-convergence of D-TiAda. The proof is available in Appendix B.3.

**Theorem 1.** *There exists a distributed minimax problem in the form of Problem (1) and certain initialization such that after running D-TiAda with any $0 < \beta < 0.5 < \alpha < 1$ and $\gamma_x, \gamma_y > 0$, it holds that for any $t = 0, 1, 2, \ldots$, we have,*

$$\| \nabla_x f(x_t, y_t) \| = \| \nabla_x f(x_0, y_0) \|, \quad \| \nabla_y f(x_t, y_t) \| = \| \nabla_y f(x_0, y_0) \|,$$

*where $\|\nabla_x f(x_0, y_0)\|$ and $\|\nabla_y f(x_0, y_0)\|$ can be arbitrarily large depending on the initialization.*

**Remark 1.** *The counterexample we constructed consists of three nodes, forming a complete graph. Without the stepsize tracking, D-TiAda will remain stationary, and the iterates will not progress if initiated along a specific line. In this counterexample, the only stationary point is at $(0, 0)$, but initial points along the line (c.f., Eq. (72)) can be positioned arbitrarily far away from this stationary point, implying the non-convergence of D-TiAda with certain initialization.*

Apart from the counterexample discussed in Theorem 1, we also experimentally observe the divergence of of D-SGDA and D-TiAda even in a simple scenario involving only two connected agents. This phenomenon is illustrated in Figure 1 and the functions are depicted as follows:

$$\begin{aligned} f_1(x, y) &= -\frac{9}{20}y^2 + \frac{3}{5}y - x + xy - \frac{1}{2}x^2, \\ f_2(x, y) &= -\frac{9}{20}y^2 + \frac{3}{5}y - x + 2xy - 2x^2. \end{aligned} \tag{6}$$

It is not difficult to verify that the points on the line $3y = 5x + 2$ are stationary points of $f(x, y) = 1/2\left(f_1(x, y) + f_2(x, y)\right)$. It follows from Figure 1(a) and 1(b) that D-SGDA does not converge to a stationary point because of the lack of time-scale separation, and D-TiAda also fails to converge due to stepsize inconsistency, as shown in Figure 1(c). In contrast, the utilization of the stepsize tracking protocol in D-AdaST ensures convergence to a stationary point, with the inconsistency in stepsizes gradually diminishing (c.f., Lemma 9). These two motivating examples effectively highlight the challenges associated with applying adaptive minimax algorithms to distributed settings.

## 2.2 The Proposed D-AdaST Algorithm

To address the issue of stepsize inconsistency across different nodes, we propose the following Distributed Adaptive minimax optimization algorithm with Stepsize Tracking protocol, termed D-AdaST, which allows us to asymptotically eliminate the stepsize inconsistency in a decentralized manner over networks. The pseudo-code for the algorithm is summarized in Algorithm 1, and can be rewritten in a compact form as follows:

$$\mathbf{m}_{k+1}^x = W\left(\mathbf{m}_k^x + \mathbf{h}_k^x\right), \tag{7a}$$

$$\mathbf{m}_{k+1}^y = W\left(\mathbf{m}_k^y + \mathbf{h}_k^y\right), \tag{7b}$$

$$\mathbf{x}_{k+1} = W\left(\mathbf{x}_k - \gamma_x V_{k+1}^{-\alpha} \nabla_x F(\mathbf{x}_k, \mathbf{y}_k; \xi_k^x)\right), \tag{7c}$$

$$\mathbf{y}_{k+1} = \mathcal{P}_{\mathcal{Y}}\left(W\left(\mathbf{y}_k + \gamma_y U_{k+1}^{-\beta} \nabla_y F(\mathbf{x}_k, \mathbf{y}_k; \xi_k^y)\right)\right), \tag{7d}$$

---

**Algorithm 1 Distributed Adaptive Minimax Method with Stepsize Tracking (D-AdaST)**

---

**Initialization:** $x_{i,0} \in \mathbb{R}^p$, $y_{i,0} \in \mathcal{Y}$, buffers $m_{i,0}^x = m_{i,0}^y = c > 0$, stepsizes $\gamma_x, \gamma_y > 0$, exponential factors $0 < \beta < \alpha < 1$ and weight matrix $W$.

1: **for** iteration $k = 0, 1, \cdots$, each node $i \in [n]$, **do**

2:    Sample i.i.d. $g_{i,k}^x = \nabla_x F_i \left( x_{i,k}, y_{i,k}; \xi_{i,k}^x \right)$ and $g_{i,k}^y = \nabla_y F_i \left( x_{i,k}, y_{i,k}; \xi_{i,k}^y \right)$.

3:    Accumulate the gradient norm:
$$m_{i,k+1}^x = m_{i,k}^x + \|g_{i,k}^x\|^2, \ m_{i,k+1}^y = m_{i,k}^y + \|g_{i,k}^y\|^2.$$

4:    Compute the ratio:
$$\psi_{i,k+1} = (m_{i,k+1}^x)^\alpha / \max \left\{ (m_{i,k+1}^x)^\alpha, (m_{i,k+1}^y)^\alpha \right\} \leqslant 1.$$

5:    Update primal and dual variables locally:
$$x_{i,k+1} = x_{i,k} - \gamma_x \psi_{i,k+1} \left( m_{i,k+1}^x \right)^{-\alpha} g_{i,k}^x, \ y_{i,k+1} = y_{i,k} + \gamma_y (m_{i,k+1}^y)^{-\beta} g_{i,k}^y.$$

6:    Communicate adaptive stepsizes and decision variables with neighbors:
$$\left\{ m_{i,k+1}^x, m_{i,k+1}^y, x_{i,k+1}, y_{i,k+1} \right\} \leftarrow \sum_{j \in \mathcal{N}_i} W_{i,j} \left\{ m_{j,k+1}^x, m_{j,k+1}^y, x_{j,k+1}, y_{j,k+1} \right\}.$$

7:    Projection of dual variable on the set $\mathcal{Y}$: $y_{i,k+1} \leftarrow \mathcal{P}_{\mathcal{Y}} \left( y_{i,k+1} \right)$.

8: **end for**

---

where $\mathbf{m}_k^x = [\cdots, m_{i,k}^x, \cdots]^T$, $\mathbf{m}_k^y = [\cdots, m_{i,k}^y, \cdots]^T$ denote the tracking variables for the accumulated global gradient norm, i.e., for $z \in \{x, y\}$,

$$\frac{\mathbf{1}^T}{n} \mathbf{m}_{k+1}^z = \frac{1}{n} \sum_{i=1}^n \left( \sum_{t=0}^k \left\| g_{i,t}^z \right\|^2 + m_{i,0}^z \right)$$

while $\boldsymbol{h}_k^z = [\cdots, \| g_{i,k}^z \|^2, \cdots]^T$, and $V_k, U_k$ are diagonal matrices with $v_{i,k} = \max \left\{ m_{i,k}^x, m_{i,k}^y \right\}$ and $u_{i,k} = m_{i,k}^x$. Note that we also provide a variant of D-AdaST with coordinate-wise adaptive stepsizes in Algorithm 2, along with its convergence analysis in Appendix B.5.

## 3 Convergence Analysis

In this section, we present the main convergence results for the proposed D-AdaST algorithm and compare it with D-TiAda to show the effectiveness of the proposed stepsize tracking protocol.

To this end, letting $\bar{u}_k := 1/n \sum_{i=1}^n u_{i,k}$, we define the following metrics to evaluate the level of inconsistency of stepsizes among nodes, which are ensured to be bounded by Assumption 3.

$$\zeta_v^2 := \sup_{i \in [n], k > 0} \left\{ \left( v_{i,k}^{-\alpha} - \bar{v}_k^{-\alpha} \right)^2 / \left( \bar{v}_k^{-\alpha} \right)^2 \right\}, \ \zeta_u^2 := \sup_{i \in [n], k > 0} \left\{ \left( u_{i,k}^{-\beta} - \bar{u}_k^{-\beta} \right)^2 / \left( \bar{u}_k^{-\beta} \right)^2 \right\}. \quad (8)$$

### 3.1 Assumptions

We consider the NC-SC setting of Problem (1) with the following assumptions that are commonly used in the existing works (c.f., Remark 2 and Remark 3). Notably, for the function and algorithm class determined by the assumptions of this work, Li et al. (2021) derived a lower complexity bound of $\Omega \left( \epsilon^{-4} \right)$ and proved that such a dependency on $\epsilon$ is optimal (c.f., Remark 2).

**Assumption 1** ($\mu$-strong concavity in $y$). *Each objective function $f_i (x, y)$ is $\mu$-strongly concave in $y$, i.e., $\forall x \in \mathbb{R}^p$, $\forall y, y' \in \mathcal{Y}$ and $\mu > 0$,*

$$f_i (x, y) - f_i (x, y') \geqslant \langle \nabla_y f_i (x, y), y - y' \rangle + \frac{\mu}{2} \| y - y' \|^2. \quad (9)$$

**Assumption 2** (Joint smoothness). *Each objective function $f_i(x, y)$ is L-smooth in $x$ and $y$, i.e., $\forall x, x' \in \mathbb{R}^p$ and $\forall y, y' \in \mathcal{Y}$, there exists a constant $L$ such that for $z \in \{x, y\}$,*

$$\|\nabla_z f_i(x, y) - \nabla_z f_i(x', y')\|^2 \leqslant L^2 \left( \|x - x'\|^2 + \|y - y'\|^2 \right). \tag{10}$$

*Furthermore, $f_i$ is second-order Lipschitz continuous for $y$, i.e., for $z \in \{x, y\}$,*

$$\left\|\nabla_{zy}^2 f_i(x, y) - \nabla_{zy}^2 f_i(x', y')\right\|^2 \leqslant L^2 \left( \|x - x'\|^2 + \|y - y'\|^2 \right). \tag{11}$$

**Remark 2.** *Assumption 1 does not require the convexity in $x$ and the objective function thus can be nonconvex. Assumption 1 and 2 ensure that $y^*(\cdot)$ is smooth (c.f., Lemma 2), which is essential for achieving (near) optimal convergence rate (Chen et al., 2021; Li et al., 2023). Besides, it can be verified that the constructed 'hard' examples for obtaining the lower complexity bound in Li et al. (2021) satisfy the above second-order Lipschitz continuity (11) on $y$, implying that the achievable optimal complexity for the function and algorithm class considered in this work is $\mathcal{O}\left(\epsilon^{-4}\right)$.*

**Assumption 3** (Stochastic gradient). *For i.i.d. sample $\xi_i$, the stochastic gradient of each $i$ is unbiased, i.e., $\forall x \in \mathbb{R}^p, y \in \mathcal{Y}$, $\mathbb{E}_{\xi_i}\left[\nabla_z F_i(x, y; \xi_i)\right] = \nabla_z f_i(x, y)$, for $z \in \{x, y\}$, and there is a constant $C > 0$ such that $\|\nabla_z F_i(x, y; \xi_i)\| \leqslant C$.*

**Remark 3.** *Assumption 3 on unbiased stochastic gradient is widely used for establishing convergence rates of both minimization and minimax optimization methods with AdaGrad (Kavis et al., 2022; Li et al., 2023) or Adam (Zou et al., 2019; Chen et al., 2023a; Huang et al., 2024) adaptive stepsize. We note that under Assumption 2, this assumption can be easily satisfied in many real-world tasks by imposing constraints on the compact domain of $f$, e.g., neural networks with rectified activation (Dinh et al., 2017) and GANs with projections on the critic (Gulrajani et al., 2017).*

Next, we make the following assumption on the underlying graph to ensure its connectivity.

**Assumption 4** (Graph connectivity). *The weight matrix $W$ induced by graph $\mathcal{G}$ is doubly stochastic, i.e., $W\mathbf{1} = \mathbf{1}, \mathbf{1}^T W = \mathbf{1}^T$ and $\rho_W := \|W - \mathbf{J}\|_2^2 < 1$.*

Note that one can always find a proper weight matrix $W$ compliant to the graph that satisfies Assumption 4 once the underlying graph is undirected and connected. For instance, the weight matrix can be easily determined based on the Metropolis-Hastings protocol (Xiao et al., 2006). Moreover, this assumption is more general than that in Lian et al. (2017); Borodich et al. (2021) in the sense that $W$ is not required to be symmetric, implying that certain directed graphs can be included in this assumption, e.g., directed ring and exponential graphs (Ying et al., 2021).

### 3.2 Main Results

We are now ready to present the key convergence results in terms of the primal function $\Phi(x) := f(x, y^*(x))$ with $y^*(x) = \underset{y \in \mathcal{Y}}{\arg\max} f(x, y)$, whose proofs can be found in Appendix B.4.

**Theorem 2.** *Suppose Assumption 1-4 hold. Let $0 < \beta < \alpha < 1$ and the total iteration $K$ satisfy*

$$\Omega\left( \max\left\{ \left( \frac{\gamma_x^2 \kappa^4}{\gamma_y^2} \right)^{\frac{1}{\alpha - \beta}}, \left( \frac{1}{(1 - \rho_W)^2} \right)^{\max\left\{\frac{1}{\alpha}, \frac{1}{\beta}\right\}} \right\} \right) \tag{12}$$

*with $\kappa := L/\mu$ to ensure time-scale separation and quasi-independence of the network. For D-AdaST, we have [1]*

$$\frac{1}{K} \sum_{k=0}^{K-1} \mathbb{E}\left[ \|\nabla \Phi(\bar{x}_k)\|^2 \right] = \tilde{\mathcal{O}}\left( \frac{1}{K^{1-\alpha}} + \frac{1}{(1 - \rho_W)^\alpha K^\alpha} \right) + \tilde{\mathcal{O}}\left( \frac{1}{K^{1-\beta}} + \frac{1}{(1 - \rho_W) K^\beta} \right). \tag{13}$$

**Remark 4** (Near-optimal convergence). *Theorem 2 implies that if the total number of iterations satisfies the conditions (12), the proposed D-AdaST algorithm converges to a stationary point exactly for Problem (1) with an $\tilde{\mathcal{O}}\left(\epsilon^{-(4+\delta)}\right)$ sample complexity for arbitrarily small $\delta > 0$, e.g., letting*

---

[1] The complete convergence result can be found in (75) in Appendix.

$\alpha = 0.5 + \delta/(8 + 2\delta)$ *and* $\beta = 0.5 - \delta/(8 + 2\delta)$. *It is worth noting that this rate is near-optimal compared to the existing lower bound of* $\Omega(\epsilon^{-4})$ *(Li et al., 2021) for a class of smooth NC-SC functions. Moreover, this result recovers the centralized TiAda algorithm (Li et al., 2023) as a special case, i.e., setting* $\rho_W = 0$, *without assuming the existence of interior optimal point (c.f., Assumption 3.3 Li et al. (2023)). To the best of our knowledge, there is no existing fully parameter-agnostic method that achieves a convergence rate of* $\tilde{\mathcal{O}}(\epsilon^{-4})$, *even in a centralized setting.*

**Remark 5** (Parameter-agnostic property and transient times)**.** *The above results show that D-AdaST converges without requiring to know any problem-dependent parameters, i.e., $L$, $\mu$ and $\rho_W$, or tuning the initial stepsize $\gamma_x$ and $\gamma_y$, and is thus parameter-agnostic. Moreover, we explicitly characterize the transient times (c.f., Eq. (12)) that ensure time-scale separation and quasi-independence of the network, respectively. Indeed, we can see that if $\alpha$ and $\beta$ are close to each other, the time required for time-scale separation to occur increases significantly, which has been observed in (Li et al., 2023). On the other hand, if $\alpha$ and $\beta$ are relatively large, then $\tilde{\mathcal{O}}\left(1/K^{1-\alpha} + 1/K^{1-\beta}\right)$ dominates the other terms, indicating independence on the network. These observations highlight the trade-offs between the convergence rate and the required duration of the transition phase.*

For proper comparison, we also derive an upper bound for D-TiAda as follows. Together with the lower bound in Theorem 1, we demonstrate that without the stepsize tracking mechanism, the inconsistency among local stepsizes prevents D-TiAda from converging in the distributed setting.

**Corollary 1.** *Under the same conditions of Theorem 2. For the proposed D-TiAda, we have*

$$
\begin{aligned}
\frac{1}{K} \sum_{k=0}^{K-1} \mathbb{E}\left[\|\nabla\Phi(\bar{x}_k)\|^2\right] &= \tilde{\mathcal{O}}\left(\frac{1}{K^{1-\alpha}} + \frac{1}{(1-\rho_W)^\alpha K^\alpha}\right) \\
&+ \tilde{\mathcal{O}}\left(\frac{1}{K^{1-\beta}} + \frac{1}{(1-\rho_W)K^\beta}\right) + \tilde{\mathcal{O}}\left(\left(\zeta_v^2 + \kappa^2\zeta_u^2\right)C^2\right).
\end{aligned}
\tag{14}
$$

## 4 Experiments

In this section, we conduct experiments to validate the theoretical findings and demonstrate the effectiveness of the proposed algorithm on real-world machine learning tasks. We compare the proposed D-AdaST with the distributed variants of AdaGrad (Duchi et al., 2011), TiAda (Li et al., 2023) and NeAda (Yang et al., 2022b), namely D-AdaGrad, D-TiAda and D-NeAda, respectively. These experiments run across multiple nodes with different networks, and we consider heterogeneous distributions of local objective functions/datasets. For example, each node can only access samples with a subset of labels on MNIST and CIFAR-10 datasets, which is a common scenario in decentralized and federated learning tasks (Sharma et al., 2023; Huang et al., 2022). The experiments cover three main tasks: synthetic function, robust training of the neural network, and training of Wasserstein GANs (Heusel et al., 2017). For the exponential factors of stepsize, we set $\alpha = 0.6$ and $\beta = 0.4$ for both D-TiAda and D-AdaST. More detailed settings and additional experiments with different initial stepsizes, data distributions and choices of $\alpha$ and $\beta$ can be found in Appendix A.

**Synthetic example.** We consider a distributed minimax problem with the following NC-SC local objective functions over exponential networks with $n = 50$ ($\rho_W = 0.71$) and $n = 100$ ($\rho_W = 0.75$).

$$
f_i(x, y) = -\frac{1}{2}y^2 + L_i xy - \frac{L_i^2}{2}x^2 - 2L_i x + L_i y,
\tag{15}
$$

where $L_i \sim \mathcal{U}(1.5, 2.5)$. The local gradient of each node is computed with an additive $\mathcal{N}(0, 0.1)$ Gaussian noise. It follows from Figure 2 (a) and 2 (b) that the proposed D-AdaST algorithm outperforms other distributed adaptive methods for both initial stepsize settings, especially in cases with a favorable initial stepsize ratio, as illustrated in plots (b) and (d) where $\gamma_x/\gamma_y = 0.2$. Similar observation can be found in Figure 2 (c) and 2 (d), demonstrating the effectiveness of D-AdaST.

**Robust training of neural networks.** Next, we consider the task of robust training of neural networks, in the presence of adversarial perturbations on data samples (Sharma et al., 2022; Deng and Mahdavi, 2021). The problem can be formulated as $\min_x \max_y 1/n \sum_{i=1}^n f_i(x; \xi_i + y) - \eta\|y\|^2$,

where $x$ denotes the parameters of the model, $y$ denotes the perturbation and $\xi_i$ denotes the data sample of node $i$. Note that if $\eta$ is large enough, the problem is NC-SC. We conduct experiments on

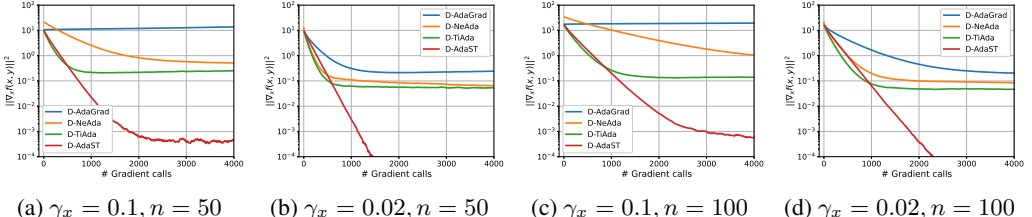

(a) $\gamma_x = 0.1, n = 50$    (b) $\gamma_x = 0.02, n = 50$    (c) $\gamma_x = 0.1, n = 100$    (d) $\gamma_x = 0.02, n = 100$

Figure 2: Performance comparison of algorithms on quadratic functions over exponential graphs with node counts $n = \{50, 100\}$ and *different initial stepsizes* ($\gamma_y = 0.1$).

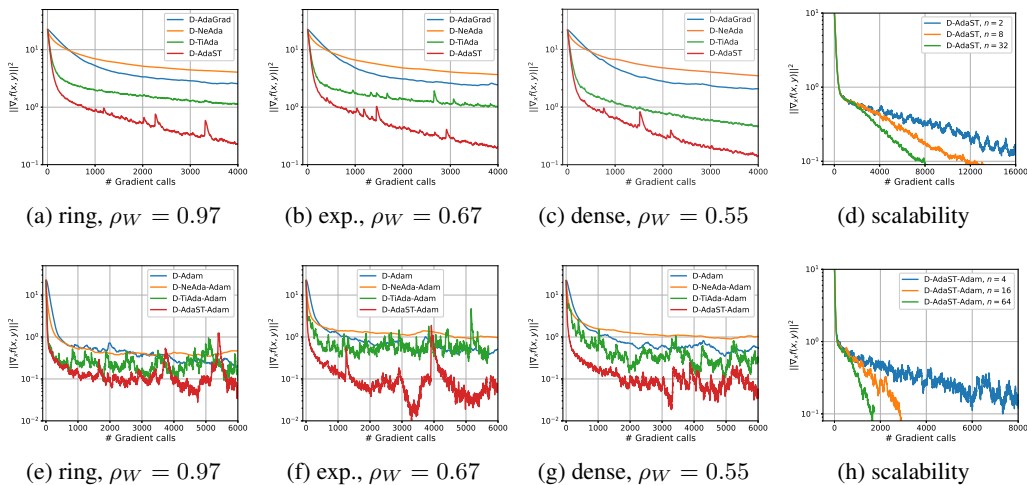

(a) ring, $\rho_W = 0.97$    (b) exp., $\rho_W = 0.67$    (c) dense, $\rho_W = 0.55$    (d) scalability

(e) ring, $\rho_W = 0.97$    (f) exp., $\rho_W = 0.67$    (g) dense, $\rho_W = 0.55$    (h) scalability

Figure 3: Comparison of the algorithms on training robust CNN on MNIST dataset. The first row shows the results of AdaGrad-like stepsize, and the second row is for Adam-like stepsize. For the first three columns, we compare the algorithms on *different graphs* with $n = 20$. For the last column, we show the scalability of D-AdaST in terms of number of nodes. Initial stepsizes are set as $\gamma_x = 0.01, \gamma_y = 0.1$ for AdaGrad-like stepsize, and $\gamma_x = 0.1, \gamma_y = 0.1$ for Adam-like stepsize.

MNIST dataset over different networks, e.g., ring graph, exponential (exp.) graph (Ying et al., 2021) and dense graph with $n/2$ edges for each node. We consider a heterogeneous scenario in which each node possesses only two distinct classes of labeled samples, resulting in heterogeneity among the local datasets across nodes, while the data is i.i.d within each node.

In Figure 3, we compare D-AdaST with D-AdaGrad, D-TiAda and D-NeAda, using adaptive stepsizes in AdaGrad (first row) and Adam (second row, name suffixed with Adam) respectively, it can be observed from the first three columns that the proposed D-AdaST outperforms the others on three different graphs and it is not very sensitive to the graph connectivity (i.e., $\rho_W$), demonstrating the quasi-independence of network as indicated in Theorem 2. It should be noted that Adam-like algorithms exhibit more fluctuations in the later stages of optimization as the gradient norm vanishes, leading to an inevitable increase in the Adam stepsize as the optimization process converges (Kingma and Ba, 2014). In plots (d) and (h), we further demonstrate that D-AdaST can scale efficiently with respect to the number of nodes, while keeping a constant batch-size of 64 for each node. This showcases the algorithm's ability to handle large-scale distributed scenarios effectively.

**Generative Adversarial Networks.** We further illustrate the effectiveness of D-AdaST on another popular task of training GANs, which has a generator and a discriminator used to generate and distinguish samples respectively (Goodfellow et al., 2014). In this experiment, we train Wasserstein GANs (Gulrajani et al., 2017) on CIFAR-10 dataset in a decentralized setting where each discriminator is 1-Lipschitz and has access to only two classes of samples. We compare the inception score of D-AdaST with D-Adam and D-TiAda adopting Adam-like stepsizes in Figure 4. It can be observed from the figure that D-AdaST achieves higher inception scores in three cases with different initial stepsizes, and has a small score loss as the initial step size changes. We believe that this example shows the great potential of D-AdaST in solving real-world problems.

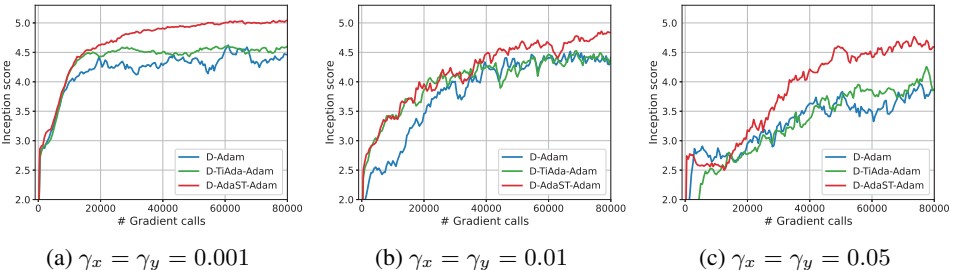

| (a) $\gamma_x = \gamma_y = 0.001$ | (b) $\gamma_x = \gamma_y = 0.01$ | (c) $\gamma_x = \gamma_y = 0.05$ |

Figure 4: Training GANs on CIFAR-10 dataset over exponential graphs with $n = 10$ nodes.

## 5   Conclusion

We introduced a new distributed adaptive minimax method, D-AdaST, designed to tackle the issue of non-convergence in nonconvex-strongly-concave minimax problems caused by the inconsistencies among locally computed adaptive stepsizes. Vanilla distributed adaptive methods could suffer from such inconsistencies, as highlighted by the carefully designed counterexamples for demonstrating their potential non-convergence. In contrast, our proposed method employs an efficient adaptive stepsize tracking protocol that not only ensures the time-scale separation, but also guarantees stepsize consistency among nodes and thus effectively eliminates steady-state errors. Theoretically, we showed that D-AdaST can achieve a near-optimal convergence rate of $\tilde{\mathcal{O}}\left(\epsilon^{-(4+\delta)}\right)$ with any arbitrarily small $\delta > 0$. Extensive experiments on both real-world and synthetic datasets have been conducted to validate our theoretical findings across various scenarios.

## Acknowledgments

The work of Huang, Shen and Xu has been supported by the National Key R&D Program of China under Grant No. 2022YFB3102100, and in parts by National Natural Science Foundation of China under Grants 62373323, 62088101. The work of Li and He has been supported by the ETH research grant and Swiss National Science Foundation (SNSF) Starting Grant.

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

# A Additional Experiments

In this section, we provide detailed experimental settings and perform additional experiments on the task of training robust neural networks with different choices of hyper-parameters. All experiments are deployed in a server with Intel Xeon E5-2680 v4 CPU @ 2.40GHz and 8 Nvidia RTX 3090 GPUs, and implemented using distributed communication package *torch.distributed* in PyTorch 2.0, where each process serves as a node, and we use inter-process communication to mimic communication between nodes. For the AdaGrad-like algorithms considered in the experiments of training neural networks, similar to the Adam-like stepsize, we adopt a coordinate-wise adaptive stepsize rule as commonly used in existing centralized adaptive methods (Yang et al., 2022b; Li et al., 2023). Moreover, since we attempt to develop a parameter-agnostic algorithm that does not need much effort in tuning hyper-parameters, we set $\alpha = 0.6$ and $\beta = 0.4$ for all tasks in the main text, and evaluate the effect of the choices of $\alpha$ and $\beta$ on the performance of D-AdaST individually in an additional experiment on the synthetic objective function as shown in Appendix A.4.

## A.1 Experimental details

**Communication topology.** For the experiments in the main text, we utilize three commonly used communication topologies: indirect ring, exponential graph and dense graph. An indirect ring is a sparse graph in which each node is sequentially connected to form a ring, with only two neighbors per node. Exponential graph (Ying et al., 2021) is a directed graph where each node is connected to nodes at distances of $2^0, 2^1 ..., 2^{\log n}$. Exponential graphs achieve a good balance between the degree and connectivity of the graph. A dense graph is an indirect graph where each node is connected to nodes at distances of $1, 2, 4, ..., n$. We also consider directed ring and fully connected graphs, which are more sparsely and densely connected, respectively, in the additional experiments.

**Robust training of neural network.** In this task, we train CNNs with three convolutional layers and one fully connected layer on MNIST dataset containing images of 10 classes. Each layer adopts batch normalization and ELU activation. The total batch-size is 1280, and the batch-size of each node during training is $1280/n$. For Adam-like algorithms, we set the first and second moment parameters as $\beta_1 = 0.9, \beta_2 = 0.999$ respectively. Since NeAda is a double-loop algorithm, for fair comparison, we imply D-AdaGrad and D-Adam using 15 iterations of inner loop in this task.

**Generative Adversarial Networks.** In this task, we train Wasserstein GANs on CIFAR-10 dataset, where the model used for discriminator is a four layer CNNs, and for generator is a four layer CNNs with transpose convolution layers. The total batch-size is 1280, and the batch-size of each node during training is 128 with 10 nodes. For Adam-like algorithms, we use $\beta_1 = 0.5, \beta_2 = 0.9$. To obtain the inception score, we use 8000 artificially generated samples to feed the previously trained inception network.

## A.2 Additional experiments on robust training of neural network.

In this part, we conduct additional experiments on robust training of CNNs on MNIST dataset considering a variety of settings. We compare the convergence performance of D-AdaST with D-AdaGrad, D-TiAda and D-NeAda using adaptive stepsizes of AdaGrad and Adam. Unless otherwise specified, the total batch-size is set to 1280; the initial stepsizes for $x$ and $y$ are assigned as $\gamma_x = 0.01, \gamma_y = 0.1$ for AdaGrad-like algorithms, and $\gamma_x = \gamma_y = 0.1$ for Adam-like algorithms. Specifically, we consider two extra graphs that are more sparse and more dense, respectively in Figure 5, e.g., directed ring and fully-connected (fc) graphs. We consider more initial stepsizes settings for $x$ and $y$ respectively in Figure 6. Further, we also consider different data distributions where each node has samples from 4 of the 10 classes in Figure 7. Finally, we perform a comparison experiment with 40 nodes in Figure 8. Under all settings, the proposed D-AdaST outperforms the others, demonstrating the superiority of D-AdaST.

## A.3 Additional experiments on training GANs

We provide additional experiments of training GANs on a more complicated dataset CIFAR-100 to further illustrate the effectiveness of the proposed D-AdaST, as shown in Figure 9. We use the entire training set of CIFAR-100 with coarse labels (20 classes) to train GANs over networks, where each node is assigned with four distinct classes of labeled samples. Under the same settings as in

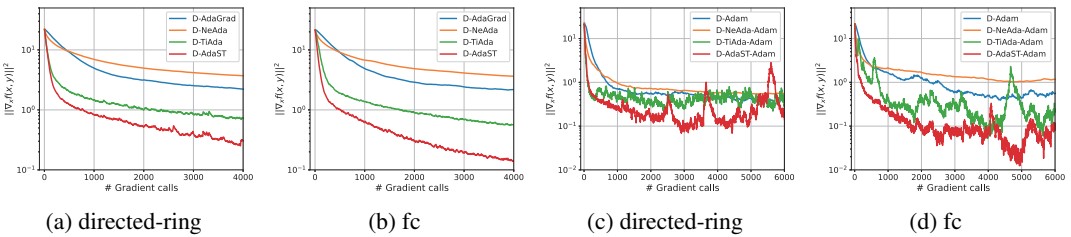

(a) directed-ring      (b) fc      (c) directed-ring      (d) fc

Figure 5: Performance comparison of training CNN on MNIST with $n = 20$ nodes over *directed ring and fully connected graphs*.

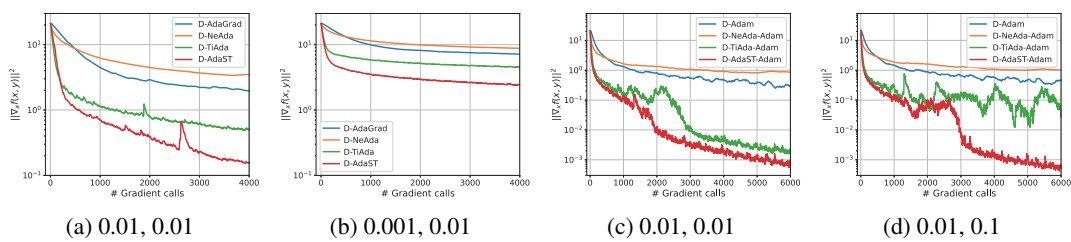

(a) 0.01, 0.01      (b) 0.001, 0.01      (c) 0.01, 0.01      (d) 0.01, 0.1

Figure 6: Performance comparison of training CNN on MNIST with $n = 20$ nodes with *different initial stepsizes $\gamma_x$ and $\gamma_y$*.

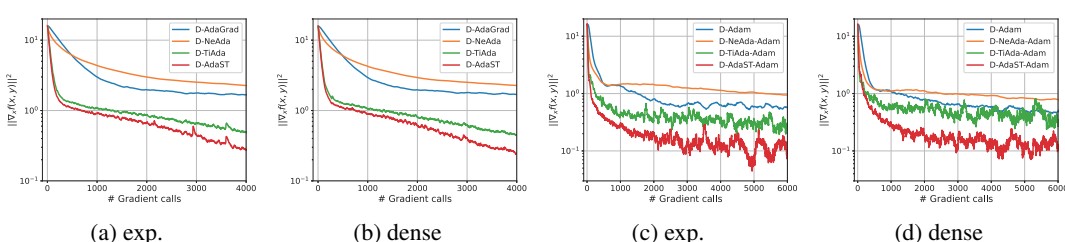

(a) exp.      (b) dense      (c) exp.      (d) dense

Figure 7: Performance comparison of training CNN on MNIST with $n = 20$ nodes over exponential and dense graphs where each node has *4 sample classes*.

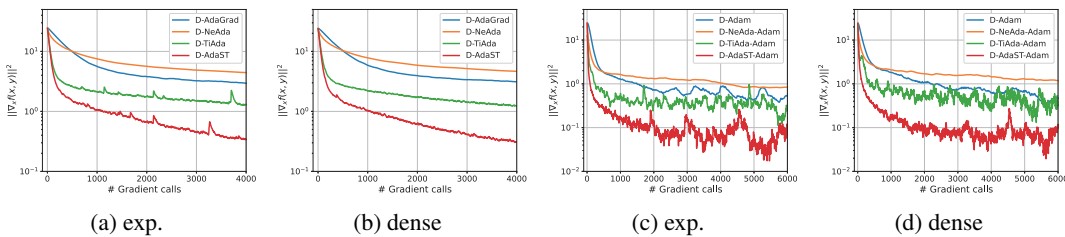

(a) exp.      (b) dense      (c) exp.      (d) dense

Figure 8: Performance comparison of training CNN on MNIST with $n = 40$ nodes over exponential and dense graphs.

Figure 4 (a), it can be observed that D-AdaST outperforms the others in terms of the inception score. Together with other experimental results in the main text, we believe that we have demonstrated the effectiveness of the proposed D-AdaST method and its potential for further real-world applications.

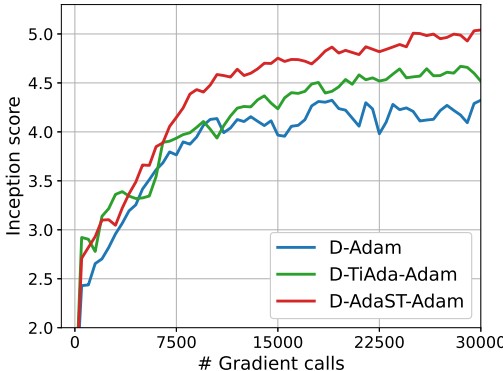

Figure 9: Performance comparison of D-AdaST with D-Adam and D-TiAda adopting Adam-like stepsizes for training GANs on CIFAR-100 with coarse labels over the exponential graph consisting of $n = 10$ nodes under initial stepsizes $\gamma_x = \gamma_y = 0.001$.

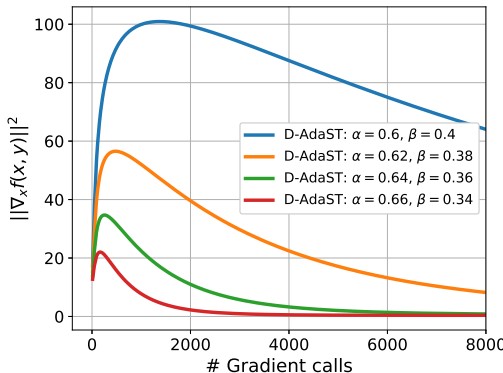

Figure 10: Performance comparison of D-AdaST on quadratic functions over an exponential graph of $n = 50$ nodes with different choices of $\alpha$ and $\beta$.

## A.4   Additional experiments with different choices of $\alpha$ and $\beta$

In this part, we evaluate the effect of the choices of $\alpha$ and $\beta$ on the performance of D-AdaST. In particular, we provide an additional experimental result on the synthetic quadratic objective functions (15) with a larger ratio of initial stepsizes, i.e., $\gamma_x/\gamma_y = 20$ (indicating faster minimization and slower maximization processes at the beginning). As shown in Figure 10, it can be observed that the transient time (iteration before the inflection point) becomes longer as $\alpha - \beta$ decreases, while the convergence rate is relatively faster, which is consistent with Theorem 2 and the result in the centralized TiAda algorithm (c.f., Figure 5, Li et al., 2023).

## B   Proof of the main results

We recall here some definitions used in the main text. The averaged variables and the inconsistency are defined as follows:

$$
\bar{x}_k := \frac{\mathbf{1}^T}{n}\mathbf{x}_k, \quad \bar{v}_k := \frac{1}{n}\sum_{i=1}^{n} v_{i,k}, \quad \left(\tilde{\boldsymbol{v}}_k^{-\alpha}\right)^T := \left[\cdots, v_{i,k}^{-\alpha} - \bar{v}_k^{-\alpha}, \cdots\right],
$$

$$
\bar{y}_k := \frac{\mathbf{1}^T}{n}\mathbf{y}_k, \quad \bar{u}_k := \frac{1}{n}\sum_{i=1}^{n} u_{i,k}, \quad \left(\tilde{\boldsymbol{u}}_k^{-\beta}\right)^T := \left[\cdots, u_{i,k}^{-\beta} - \bar{u}_k^{-\beta}, \cdots\right].
$$

The inconsistency of stepsizes of the primal and dual variables is defined as follows:

$$\zeta_v^2 := \sup_{i \in [n], k > 0} \left\{ \left( v_{i,k}^{-\alpha} - \bar{v}_k^{-\alpha} \right)^2 / \left( \bar{v}_k^{-\alpha} \right)^2 \right\}, \ \zeta_u^2 := \sup_{i \in [n], k > 0} \left\{ \left( u_{i,k}^{-\beta} - \bar{u}_k^{-\beta} \right)^2 / \left( \bar{u}_k^{-\beta} \right)^2 \right\}.$$

**Proof Sketch.** The convergence analysis of the main results in Theorem 2 is mainly based on carefully analyzing the average system as shown in (5), and the difference between the distributed system and the averaged system. In general, under Assumption 1-4, we first give a telescoped descent lemma from 0 to $K-1$ iterations in Lemma 3, which is upper bounded by the following key error terms:

- $S_1 := \frac{1}{nK} \sum_{k=0}^{K-1} \mathbb{E}\left[ \bar{v}_{k+1}^{-\alpha} \| \nabla_x F(\mathbf{x}_k, \mathbf{y}_k; \xi_k^x) \|^2 \right]$: The asymptotically decaying terms by adopting adaptive stepsize;

- $S_2 := \frac{1}{nK} \sum_{k=0}^{K-1} \mathbb{E}\left[ \| \mathbf{x}_k - \mathbf{1}\bar{x}_k \|^2 + \| \mathbf{y}_k - \mathbf{1}\bar{y}_k \|^2 \right]$: The consensus error of $x$ and $y$ between the distributed system and the average system;

- $S_3 := \frac{1}{K} \sum_{k=0}^{K-1} \mathbb{E}\left[ f(\bar{x}_k, y^*(\bar{x}_k)) - f(\bar{x}_k, \bar{y}_k) \right]$: The optimality gap in dual variable $y$;

- $S_4 := \frac{1}{K} \sum_{k=0}^{K-1} \mathbb{E}\left[ \left\| \frac{(\tilde{\mathbf{v}}_{k+1}^{-\alpha})^T}{n\bar{v}_{k+1}^{-\alpha}} \nabla_x F(\mathbf{x}_k, \mathbf{y}_k; \xi_k^x) \right\|^2 \right]$: The inconsistency of stepsize of $x$.

Next, we prove the contraction properties of these terms in Lemma 4-8 and Lemma 9 respectively. Finally, these results are integrated into the descent lemma to complete the proof. We note that the proof is not trivial in the sense that these terms are coupled and therefore are needed to be carefully analyzed. This proof can also be adapted to analyze the coordinate-wise adaptive stepsize variant of D-AdaST as explained in Appendix B.5, which is of independent interest.

## B.1 Supporting lemmas

In this part, we provide several supporting lemmas that have been shown in the existing literature, which are essential to the subsequent convergence analysis.

**Lemma 1** (Lemma A.2 in Yang et al. (2022b)). *Let $\{x_t\}_{t=0}^{T-1}$ be a sequence of non-negative real numbers, $x_0 > 0$ and $\alpha \in (0, 1)$. Then we have,*

$$\left( \sum_{t=0}^{T-1} x_t \right)^{1-\alpha} \leqslant \sum_{t=0}^{T-1} \frac{x_t}{\left( \sum_{k=0}^t x_k \right)^\alpha} \leqslant \frac{1}{1-\alpha} \left( \sum_{t=0}^{T-1} x_t \right)^{1-\alpha}. \tag{16}$$

*When $\alpha = 0$, we have*

$$\sum_{t=0}^{T-1} \frac{x_t}{\left( \sum_{k=0}^t x_k \right)^\alpha} \leqslant 1 + \log\left( \frac{\sum_{t=0}^{T-1} x_t}{x_0} \right). \tag{17}$$

**Lemma 2.** *Suppose Assumption 1 and 2 hold. Define $\Phi(x) := f(x, y^*(x))$ as the envelope function and $y^*(x) = \arg\max_{y \in \mathcal{Y}} f(x, y)$. Then, we have,*

- *$\Phi(\cdot)$ is $L_\Phi$-smooth with $L_\Phi = L(1 + \kappa)$, and $\nabla\Phi(x) = \nabla_x f(x, y^*(x))$ (c.f., Lemma 4.3 in Lin et al. (2020));*

- *$y^*(\cdot)$ is $\kappa$-Lipschitz and $\hat{L}$-smooth with $\hat{L} = \kappa(1 + \kappa)^2$ (c.f., Lemma 2 in Chen et al. (2021)).*

## B.2 Key Lemmas

In this subsection, we give the key lemmas to help the analysis of the main results. For simplicity, we define $\Delta_k := \| \mathbf{x}_k - \mathbf{1}\bar{x}_k \|^2 + \| \mathbf{y}_k - \mathbf{1}\bar{y}_k \|^2$ as the consensus error for primal and dual variables. Then, we have the following lemmas.

**Lemma 3** (Descent lemma). *Suppose Assumption 1-4 hold. Then, we have*

$$\frac{1}{K}\sum_{k=0}^{K-1}\mathbb{E}\left[\|\nabla\Phi\left(\bar{x}_k\right)\|^2\right]$$

$$\leqslant \frac{8C^{2\alpha}\left(\Phi^{\max}-\Phi^*\right)}{\gamma_x K^{1-\alpha}} - \frac{4}{K}\sum_{k=0}^{K-1}\mathbb{E}\left[\|\nabla_x f\left(\bar{x}_k,\bar{y}_k\right)\|^2\right]$$

$$+8\gamma_x L_\Phi\left(1+\zeta_v^2\right)\underbrace{\frac{1}{nK}\sum_{k=0}^{K-1}\mathbb{E}\left[\bar{v}_{k+1}^{-\alpha}\|\nabla_x F\left(\mathbf{x}_k,\mathbf{y}_k;\xi_k^x\right)\|^2\right]}_{S_1} + 8L^2\underbrace{\frac{1}{nK}\sum_{k=0}^{K-1}\mathbb{E}\left[\Delta_k\right]}_{S_2}$$

$$+8\kappa L\underbrace{\frac{1}{K}\sum_{k=0}^{K-1}\mathbb{E}\left[f\left(\bar{x}_k,y^*\left(\bar{x}_k\right)\right)-f\left(\bar{x}_k,\bar{y}_k\right)\right]}_{S_3} + 16\underbrace{\frac{1}{K}\sum_{k=0}^{K-1}\mathbb{E}\left[\left\|\frac{\left(\tilde{\boldsymbol{v}}_{k+1}^{-\alpha}\right)^T}{n\bar{v}_{k+1}^{-\alpha}}\nabla_x F\left(\mathbf{x}_k,\mathbf{y}_k;\xi_k^x\right)\right\|^2\right]}_{S_4},$$

$$(18)$$

*where $\kappa := L/\mu$ is the condition number of the function in $y$, $\Phi^{\max}=\max\limits_x \Phi\left(x\right)$, $\Phi^*=\min\limits_x \Phi\left(x\right)$.*

*Proof.* By the smoothness of $\Phi$ given in Lemma 2, i.e.,

$$\Phi\left(\bar{x}_{k+1}\right)-\Phi\left(\bar{x}_k\right) \leqslant \langle\nabla\Phi\left(\bar{x}_k\right),\bar{x}_{k+1}-\bar{x}_k\rangle + \frac{L_\Phi}{2}\|\bar{x}_{k+1}-\bar{x}_k\|^2,$$

and noticing that the scalar $\bar{v}_k,\bar{u}_k$ are random variables, we have

$$\mathbb{E}\left[\frac{\Phi\left(\bar{x}_{k+1}\right)-\Phi\left(\bar{x}_k\right)}{\gamma_x\bar{v}_{k+1}^{-\alpha}}\right]$$

$$\leqslant -\mathbb{E}\left[\left\langle\nabla\Phi\left(\bar{x}_k\right),\frac{\mathbf{1}^T}{n}\nabla_x F\left(\mathbf{x}_k,\mathbf{y}_k;\xi_k\right)\right\rangle\right] - \mathbb{E}\left[\left\langle\nabla\Phi\left(\bar{x}_k\right),\frac{\left(\tilde{\boldsymbol{v}}_{k+1}^{-\alpha}\right)^T}{n\bar{v}_{k+1}^{-\alpha}}\nabla_x F\left(\mathbf{x}_k,\mathbf{y}_k;\xi_k^x\right)\right\rangle\right]$$

$$+\frac{\gamma_x L_\Phi}{2}\mathbb{E}\left[\frac{1}{\bar{v}_{k+1}^{-\alpha}}\left\|\left(\frac{\bar{v}_{k+1}^{-\alpha}\mathbf{1}^T}{n}+\frac{\left(\tilde{\boldsymbol{v}}_{k+1}^{-\alpha}\right)^T}{n}\right)\nabla_x F\left(\mathbf{x}_k,\mathbf{y}_k;\xi_k^x\right)\right\|^2\right],$$

$$(19)$$

where we have used the definition of $\bar{x}_{k+1}$ as presented in (5). Then, we bound the inner-product terms on the RHS. Firstly,

$$-\mathbb{E}\left[\left\langle\nabla\Phi\left(\bar{x}_k\right),\frac{\mathbf{1}^T}{n}\nabla_x F\left(\mathbf{x}_k,\mathbf{y}_k;\xi_k^x\right)\right\rangle\right]$$

$$= -\mathbb{E}\left[\left\langle\nabla\Phi\left(\bar{x}_k\right),\frac{\mathbf{1}^T}{n}\nabla_x F\left(\mathbf{x}_k,\mathbf{y}_k\right)-\frac{\mathbf{1}^T}{n}\nabla_x F\left(\mathbf{1}\bar{x}_k,\mathbf{1}\bar{y}_k\right)+\frac{\mathbf{1}^T}{n}\nabla_x F\left(\mathbf{1}\bar{x}_k,\mathbf{1}\bar{y}_k\right)\right\rangle\right]$$

$$\leqslant \frac{1}{4}\mathbb{E}\left[\|\nabla\Phi\left(\bar{x}_k\right)\|^2\right] + \mathbb{E}\left[\left\|\frac{\mathbf{1}^T}{n}\nabla_x F\left(\mathbf{x}_k,\mathbf{y}_k\right)-\frac{\mathbf{1}^T}{n}\nabla_x F\left(\mathbf{1}\bar{x}_k,\mathbf{1}\bar{y}_k\right)\right\|^2\right] \qquad (20)$$

$$+\frac{1}{2}\left(\mathbb{E}\left[\|\nabla\Phi\left(\bar{x}_k\right)-\nabla_x f\left(\bar{x}_k,\bar{y}_k\right)\|^2\right]-\mathbb{E}\left[\|\nabla\Phi\left(\bar{x}_k\right)\|^2\right]-\mathbb{E}\left[\|\nabla_x f\left(\bar{x}_k,\bar{y}_k\right)\|^2\right]\right)$$

$$\leqslant -\frac{1}{4}\mathbb{E}\left[\|\nabla\Phi\left(\bar{x}_k\right)\|^2\right]+\frac{L^2}{n}\mathbb{E}\left[\Delta_k\right]+\frac{L^2}{2}\mathbb{E}\left[\|\bar{y}_k-y^*\left(\bar{x}_k\right)\|^2\right]-\frac{1}{2}\mathbb{E}\left[\|\nabla_x f\left(\bar{x}_k,\bar{y}_k\right)\|^2\right].$$

wherein the last inequality we have used the smoothness of the objective functions. Then, for the second inner-product in (19), using Young's inequality we get

$$-\mathbb{E}\left[\left\langle\nabla\Phi\left(\bar{x}_k\right),\frac{\left(\tilde{\boldsymbol{v}}_{k+1}^{-\alpha}\right)^T}{n\bar{v}_{k+1}^{-\alpha}}\nabla_x F\left(\mathbf{x}_k,\mathbf{y}_k;\xi_k^x\right)\right\rangle\right]$$

$$\leqslant \frac{1}{8}\mathbb{E}\left[\|\nabla\Phi\left(\bar{x}_k\right)\|^2\right]+2\mathbb{E}\left[\left\|\frac{\left(\tilde{\boldsymbol{v}}_{k+1}^{-\alpha}\right)^T}{n\bar{v}_{k+1}^{-\alpha}}\nabla_x F\left(\mathbf{x}_k,\mathbf{y}_k;\xi_k^x\right)\right\|^2\right]. \qquad (21)$$

Then, for the last term on the RHS of (18), recalling the definition of stepsize inconsistency in (8), we have

$$
\frac{\gamma_x L_\Phi}{2} \mathbb{E}\left[\frac{1}{\bar{v}_{k+1}^{-\alpha}} \left\| \left( \frac{\bar{v}_{k+1}^{-\alpha} \mathbf{1}^T}{n} + \frac{\left(\tilde{\boldsymbol{v}}_{k+1}^{-\alpha}\right)^T}{n} \right) \nabla_x F\left(\mathbf{x}_k, \mathbf{y}_k; \xi_k^x\right) \right\|^2\right]
$$
$$
\leqslant \frac{\gamma_x L_\Phi \left(1 + \zeta_v^2\right)}{n} \mathbb{E}\left[\bar{v}_{k+1}^{-\alpha} \left\| \nabla_x F\left(\mathbf{x}_k, \mathbf{y}_k; \xi_k^x\right) \right\|^2\right].
$$

(22)

Plugging the obtained inequalities into (18) and telescoping the terms, we get

$$
\sum_{k=0}^{K-1} \mathbb{E}\left[\left\| \nabla \Phi\left(\bar{x}_k\right) \right\|^2\right]
$$
$$
\leqslant 8 \sum_{k=0}^{K-1} \mathbb{E}\left[\frac{\Phi\left(\bar{x}_k\right) - \Phi\left(\bar{x}_{k+1}\right)}{\gamma_x \bar{v}_k^{-\alpha}}\right] - 4 \sum_{k=0}^{K-1} \mathbb{E}\left[\left\| \nabla_x f\left(\bar{x}_k, \bar{y}_k\right) \right\|^2\right]
$$
$$
+ 4L^2 \sum_{k=0}^{K-1} \mathbb{E}\left[\left\| \bar{y}_k - \bar{y}^* \right\|^2\right] + \frac{8L^2}{n} \sum_{k=0}^{K-1} \mathbb{E}\left[\Delta_k\right]
$$
$$
+ \frac{8\gamma_x L_\Phi \left(1 + \zeta_v^2\right)}{n} \sum_{k=0}^{K-1} \mathbb{E}\left[\bar{v}_k^{-\alpha} \left\| \nabla_x F\left(\mathbf{x}_k, \mathbf{y}_k; \xi_k^x\right) \right\|^2\right]
$$
$$
+ 16 \sum_{k=0}^{K-1} \mathbb{E}\left[\left\| \frac{\left(\tilde{\boldsymbol{v}}_{k+1}^{-\alpha}\right)^T}{n\bar{v}_{k+1}^{-\alpha}} \nabla_x F\left(\mathbf{x}_k, \mathbf{y}_k; \xi_k^x\right) \right\|^2\right].
$$

(23)

Now it remains to bound the first term on the RHS of the above inequality. With the help of Assumption 3, we have

$$
\sum_{k=0}^{K-1} \mathbb{E}\left[\frac{\Phi\left(\bar{x}_k\right) - \Phi\left(\bar{x}_{k+1}\right)}{\gamma_x \bar{v}_{k+1}^{-\alpha}}\right]
$$
$$
= \sum_{k=0}^{K-1} \mathbb{E}\left[\frac{\Phi\left(\bar{x}_k\right)}{\gamma_x \bar{v}_k^{-\alpha}} - \frac{\Phi\left(\bar{x}_{k+1}\right)}{\gamma_x \bar{v}_{k+1}^{-\alpha}} + \Phi\left(\bar{x}_k\right) \left( \frac{1}{\gamma_x \bar{v}_{k+1}^{-\alpha}} - \frac{1}{\gamma_x \bar{v}_k^{-\alpha}} \right)\right]
$$
$$
\leqslant \mathbb{E}\left[\frac{\Phi_{\max}}{\gamma_x \bar{v}_0^{-\alpha}} - \frac{\Phi^*}{\gamma_x \bar{v}_K^{-\alpha}}\right] + \sum_{k=0}^{K-1} \mathbb{E}\left[\Phi_{\max} \left( \frac{1}{\gamma_x \bar{v}_{k+1}^{-\alpha}} - \frac{1}{\gamma_x \bar{v}_k^{-\alpha}} \right)\right]
$$
$$
\leqslant \frac{\left(\Phi_{\max} - \Phi^*\right)}{\gamma_x} \mathbb{E}\left[\bar{v}_K^\alpha\right]
$$
$$
\leqslant \frac{\left(\Phi_{\max} - \Phi^*\right) \left(KC^2\right)^\alpha}{\gamma_x}.
$$

(24)

Noticing that $\mathbb{E}\left[\left\| \bar{y}_k - y^*\left(\bar{x}_k\right) \right\|^2\right] \leqslant \frac{2}{\mu} \mathbb{E}\left[f\left(\bar{x}_k, y^*\left(\bar{x}_k\right)\right) - f\left(\bar{x}_k, \bar{y}_k\right)\right]$, we thus complete the proof.

$\square$

Next, we need to bound the last four terms $S_1$-$S_4$ in (18) respectively. For $S_1$, we have the asymptotic convergence for both primal and dual variables in the following lemma.

**Lemma 4.** *Suppose Assumption 1-4 hold. Then, we have*

$$
\frac{1}{nK} \sum_{k=0}^{K-1} \mathbb{E}\left[\bar{v}_{k+1}^{-\alpha} \left\| \nabla_x F\left(\mathbf{x}_k, \mathbf{y}_k; \xi_k^x\right) \right\|^2\right] \leqslant \frac{C^{2-2\alpha}}{(1-\alpha) K^\alpha},
$$

(25)

*and*

$$
\frac{1}{nK} \sum_{k=0}^{K-1} \mathbb{E}\left[\bar{u}_{k+1}^{-\beta} \left\| \nabla_y F\left(\mathbf{x}_k, \mathbf{y}_k; \xi_k^y\right) \right\|^2\right] \leqslant \frac{C^{2-2\beta}}{(1-\beta) K^\beta}.
$$

(26)

*Proof.* With the help of Lemma 1 and Assumption 3, taking the primal variable $x$ as an example, and noticing that $v_{i,0} > 0, i \in [n]$, we have

$$\frac{1}{K} \sum_{k=0}^{K-1} \mathbb{E}\left[\bar{v}_{k+1}^{-\alpha} \|\nabla_x F(\mathbf{x}_k, \mathbf{y}_k; \xi_k^x)\|^2\right]$$

$$= \frac{1}{K} \sum_{k=0}^{K-1} \frac{1}{n} \sum_{i=1}^{n} \frac{\left\|\nabla_x F_i\left(x_{i,k}, y_{i,k}; \xi_{i,k}^x\right)\right\|^2}{\bar{v}_{k+1}^{\alpha}}$$

$$\leqslant \frac{1}{K} \sum_{k=0}^{K-1} \frac{1}{n} \sum_{i=1}^{n} \frac{\left\|\nabla_x F_i\left(x_{i,k}, y_{i,k}; \xi_{i,k}^x\right)\right\|^2}{\left(\sum_{t=0}^{k} \frac{1}{n} \sum_{j=1}^{n} \left\|\nabla_x F_j\left(x_{j,t}, y_{j,t}; \xi_{j,t}^x\right)\right\|^2\right)^{\alpha}}$$

$$\leqslant \frac{1}{1-\alpha} \frac{1}{K} \left(\sum_{k=0}^{K-1} \frac{1}{n} \sum_{i=1}^{n} \left\|\nabla_x F_i\left(x_{i,k}, y_{i,k}; \xi_{i,k}^x\right)\right\|^2\right)^{1-\alpha} \leqslant \frac{C^{2-2\alpha}}{(1-\alpha) K^{\alpha}}.$$

The similar result can be obtained for dual variable $y$ and we thus complete the proof. $\qquad \square$

Next, we bound the the consensus error term $S_2$ in the following lemma.

**Lemma 5.** *Suppose Assumption 1-4 hold. Then, we have*

$$\frac{1}{K} \sum_{k=0}^{K} \mathbb{E}\left[\Delta_k\right] \leqslant \frac{2\mathbb{E}\left[\Delta_0\right]}{(1-\rho_W) K}$$

$$+ \frac{8n\rho_W \gamma_x^2 \left(1 + \zeta_v^2\right)}{(1-\rho_W)^2} \left(\frac{C^{2-4\alpha}}{(1-2\alpha) K^{2\alpha}} \mathbb{I}_{\alpha < 1/2} + \frac{1 + \log v_K - \log v_1}{K \bar{v}_1^{2\alpha-1}} \mathbb{I}_{\alpha \geqslant 1/2}\right) \quad (27)$$

$$+ \frac{8n\rho_W \gamma_y^2 \left(1 + \zeta_u^2\right)}{(1-\rho_W)^2} \left(\frac{C^{2-4\beta}}{(1-2\beta) K^{2\beta}} \mathbb{I}_{\beta < 1/2} + \frac{1 + \log u_K - \log u_1}{K \bar{u}_1^{2\beta-1}} \mathbb{I}_{\beta \geqslant 1/2}\right),$$

*where $\mathbb{I}_{[\cdot]} \in \{0, 1\}$ is the indicator for specific condition, and the initial consensus error $\Delta_0$ can be set to 0 with proper initialization.*

*Proof.* By the updating rule of the primal variable, we have

$$\mathbb{E}\left[\|\mathbf{x}_{k+1} - \mathbf{1}\bar{x}_{k+1}\|^2\right]$$

$$= \mathbb{E}\left[\left\|W\left(\mathbf{x}_k - \gamma_x V_{k+1}^{-\alpha} \nabla_x F(\mathbf{x}_k, \mathbf{y}_k; \xi_k^x)\right) - \mathbf{J}\left(\mathbf{x}_k - \gamma_x V_{k+1}^{-\alpha} \nabla_x F(\mathbf{x}_k, \mathbf{y}_k; \xi_k^x)\right)\right\|^2\right]$$

$$\leqslant \frac{1+\rho_W}{2} \mathbb{E}\left[\|\mathbf{x}_k - \mathbf{1}\bar{x}_k\|^2\right] + \frac{2\gamma_x^2 (1+\rho_W) \rho_W}{1-\rho_W} \mathbb{E}\left[\bar{v}_{k+1}^{-2\alpha} \|\nabla_x F(\mathbf{x}_k, \mathbf{y}_k; \xi_k^x)\|^2\right] \quad (28)$$

$$+ \frac{2\gamma_x^2 (1+\rho_W) \rho_W}{1-\rho_W} \mathbb{E}\left[\left\|\left(V_{k+1}^{-\alpha} - \bar{v}_{k+1}^{-\alpha}\mathbf{I}\right) \nabla_x F(\mathbf{x}_k, \mathbf{y}_k; \xi_k^x)\right\|^2\right],$$

where we have used Young's inequality. Then, by the definition of $\zeta_v$ in (8), we have

$$\mathbb{E}\left[\left\|\left(V_{k+1}^{-\alpha} - \bar{v}_{k+1}^{-\alpha}\mathbf{I}\right) \nabla_x F(\mathbf{x}_k, \mathbf{y}_k; \xi_k^x)\right\|^2\right] \leqslant \zeta_v^2 \mathbb{E}\left[\bar{v}_{k+1}^{-2\alpha} \|\nabla_x F(\mathbf{x}_k, \mathbf{y}_k; \xi_k^x)\|^2\right], \quad (29)$$

and thus

$$\sum_{k=0}^{K-1} \mathbb{E}\left[\|\mathbf{x}_{k+1} - \mathbf{1}\bar{x}_{k+1}\|^2\right]$$

$$\leqslant \frac{2}{1-\rho_W} \mathbb{E}\left[\|\mathbf{x}_k - \mathbf{1}\bar{x}_k\|^2\right] + \frac{8\gamma_x^2 \rho_W \left(1 + \zeta_v^2\right)}{(1-\rho_W)^2} \sum_{k=0}^{K-1} \mathbb{E}\left[\bar{v}_{k+1}^{-2\alpha} \|\nabla_x F(\mathbf{x}_k, \mathbf{y}_k; \xi_k^x)\|^2\right]. \quad (30)$$

Then, we bound the last term on the RHS of the above inequality by Lemma 4. For the case $\alpha < 1/2$, by Assumption 3 we have

$$\sum_{k=0}^{K-1} \mathbb{E}\left[\bar{v}_{k+1}^{-2\alpha} \left\|\nabla_x F\left(\mathbf{x}_k, \mathbf{y}_k; \xi_k^x\right)\right\|^2\right]$$

$$= \sum_{k=0}^{K-1} \sum_{i=1}^{n} \mathbb{E}\left[\frac{\left\|\nabla_x F_i\left(x_{i,k}, y_{i,k}; \xi_{i,k}^x\right)\right\|^2}{\bar{v}_{k+1}^{2\alpha}}\right] \leqslant \frac{n\left(KC^2\right)^{1-2\alpha}}{(1-2\alpha)}. \tag{31}$$

For the case $\alpha \geqslant 1/2$, with the help of Lemma 1, we have

$$\sum_{k=0}^{K-1} \mathbb{E}\left[\bar{v}_{k+1}^{-2\alpha} \left\|\nabla_x F\left(\mathbf{x}_k, \mathbf{y}_k; \xi_k^x\right)\right\|^2\right]$$

$$= \sum_{k=0}^{K-1} \sum_{i=1}^{n} \mathbb{E}\left[\frac{\left\|\nabla_x F_i\left(x_{i,k}, y_{i,k}; \xi_{i,k}^x\right)\right\|^2}{\bar{v}_{k+1} \cdot \bar{v}_{k+1}^{2\alpha-1}}\right] \leqslant \frac{n\left(1 + \log v_T - \log v_1\right)}{\bar{v}_1^{2\alpha-1}}. \tag{32}$$

For the dual variable, we have

$$\mathbf{y}_{k+1} = \mathcal{P}_{\mathcal{Y}}\left(W\left(\mathbf{y}_k + \gamma_y U_{k+1}^{-\beta} \nabla_y F\left(\mathbf{x}_k, \mathbf{y}_k; \xi_k^y\right)\right)\right)$$

$$= W\mathbf{y}_k + \gamma_y \nabla_y \hat{G}$$

where

$$\nabla_y \hat{G} = \frac{1}{\gamma_y}\left(\mathcal{P}_{\mathcal{Y}}\left(W\left(\mathbf{y}_k + \gamma_y U_{k+1}^{-\beta} \nabla_y F\left(\mathbf{x}_k, \mathbf{y}_k; \xi_k^y\right)\right)\right) - W\mathbf{y}_k\right).$$

Then, using Young's inequality with parameter $\lambda$, we have

$$\mathbb{E}\left[\left\|\mathbf{y}_{k+1} - \mathbf{1}\bar{y}_{k+1}\right\|^2\right]$$

$$= \mathbb{E}\left[\left\|W\mathbf{y}_k + \gamma_y \nabla_y \hat{G} - \mathbf{J}\left(W\mathbf{y}_k + \gamma_y \nabla_y \hat{G}\right)\right\|^2\right]$$

$$\leqslant (1+\lambda)\rho_W \mathbb{E}\left[\left\|\mathbf{y}_k - \mathbf{J}\mathbf{y}_k\right\|^2\right]$$

$$+ \left(1 + \frac{1}{\lambda}\right)\mathbb{E}\left[\left\|\mathcal{P}_{\mathcal{Y}}\left(W\left(\mathbf{y}_k + \gamma_y U_{k+1}^{-\beta} \nabla_y F\left(\mathbf{x}_k, \mathbf{y}_k; \xi_k^y\right)\right)\right) - W\mathbf{y}_k\right\|^2\right]$$

$$\leqslant \frac{1+\rho_W}{2}\mathbb{E}\left[\left\|\mathbf{y}_k - \mathbf{J}\mathbf{y}_k\right\|^2\right]$$

$$+ \frac{1+\rho_W}{1-\rho_W}\mathbb{E}\left[\left\|\mathcal{P}_{\mathcal{Y}}\left(W\left(\mathbf{y}_k + \gamma_y U_{k+1}^{-\beta} \nabla_y F\left(\mathbf{x}_k, \mathbf{y}_k; \xi_k^y\right)\right)\right) - W\mathbf{y}_k\right\|^2\right].$$

Noticing that $W\mathbf{y}_k = \mathcal{P}_{\mathcal{Y}}\left(W\mathbf{y}_k\right)$ holds for convex set $\mathcal{Y}$, we get

$$\mathbb{E}\left[\left\|\mathbf{y}_{k+1} - \mathbf{1}\bar{y}_{k+1}\right\|^2\right]$$

$$\leqslant \frac{1+\rho_W}{2}\mathbb{E}\left[\left\|\mathbf{y}_k - \mathbf{J}\mathbf{y}_k\right\|^2\right]$$

$$+ \frac{1+\rho_W}{1-\rho_W}\mathbb{E}\left[\left(\left\|\mathcal{P}_{\mathcal{Y}}\left(W\left(\mathbf{y}_k + \gamma_y U_{k+1}^{-\beta} \nabla_y F\left(\mathbf{x}_k, \mathbf{y}_k; \xi_k^y\right)\right)\right) - \mathcal{P}_{\mathcal{Y}}\left(W\mathbf{y}_k\right)\right\|\right)^2\right]$$

$$\leqslant \frac{1+\rho_W}{2}\mathbb{E}\left[\left\|\mathbf{y}_k - \mathbf{J}\mathbf{y}_k\right\|^2\right] + \frac{1+\rho_W}{1-\rho_W}\mathbb{E}\left[\left\|\gamma_y U_{k+1}^{-\beta} \nabla_y F\left(\mathbf{x}_k, \mathbf{y}_k; \xi_k^y\right)\right\|^2\right]$$

$$\leqslant \frac{1+\rho_W}{2}\mathbb{E}\left[\left\|\mathbf{y}_k - \mathbf{J}\mathbf{y}_k\right\|^2\right] + \frac{4\gamma_y^2\left(1+\zeta_u^2\right)}{(1-\rho_W)}\mathbb{E}\left[\bar{u}_{k+1}^{-2\beta} \left\|\nabla_y F\left(\mathbf{x}_k, \mathbf{y}_k; \xi_k^y\right)\right\|^2\right],$$

where we have used the non-expansiveness of projection operator. Then, we have

$$\sum_{k=0}^{K-1} \mathbb{E}\left[\|\mathbf{y}_k - \mathbf{1}\bar{y}_k\|^2\right]$$

$$\leqslant \frac{2}{1-\rho_W} \mathbb{E}\left[\|\mathbf{y}_0 - \mathbf{J}\mathbf{y}_0\|^2\right] + \frac{8\gamma_y^2\left(1+\zeta_u^2\right)}{(1-\rho_W)^2} \sum_{k=0}^{K-1} \mathbb{E}\left[\bar{u}_{k+1}^{-2\beta} \|\nabla_y F\left(\mathbf{x}_k,\mathbf{y}_k;\xi_k^y\right)\|^2\right].$$

Similar to the primal variable, we can bound the last term above, which completes the proof. $\square$

Next, we need to bound the term $S_3$ i.e., the optimality gap in dual variable. The intuition of the proof relies on the adaptive two time-scale protocol, that is, for given $\alpha$ and $\beta$, we try to find the threshold of the iterations $k_0$, after which the inner sub-problem can be well solved (faster) to ensure that the computation of outer sub-problem can be solved accurately (slower). In specific, we suppose that there is a constant $G$ such that $\bar{u}_k \leqslant G$ hold for $k = 0, 1, \cdots, k_0 - 1$, then the analysis is divided into two phases.

**Lemma 6** (First phase). *Suppose Assumption 1-4 hold. If $\bar{u}_k \leqslant G, k = 0, 1, \cdots, k_0 - 1$, then we have*

$$\sum_{k=0}^{k_0-1} \mathbb{E}\left[f\left(\bar{x}_k, y^*\left(\bar{x}_k\right)\right) - f\left(\bar{x}_k, \bar{y}_k\right)\right]$$

$$\leqslant \sum_{k=0}^{k_0-1} \mathbb{E}\left[E_{1,k}\right] + \frac{\gamma_x^2\kappa^2\left(1+\zeta_v^2\right)G^{2\beta}}{n\mu\gamma_y^2} \sum_{k=0}^{k_0-1} \mathbb{E}\left[\bar{v}_{k+1}^{-2\alpha} \|\nabla_x F\left(\mathbf{x}_k,\mathbf{y}_k;\xi_k^x\right)\|^2\right]$$

$$+ \frac{\gamma_y\left(1+\zeta_u^2\right)}{n} \sum_{k=0}^{k_0-1} \mathbb{E}\left[\bar{u}_{k+1}^{-\beta} \|\nabla_y F\left(\mathbf{x}_k,\mathbf{y}_k;\xi_k\right)\|^2\right] + \frac{4\kappa L}{n} \sum_{k=0}^{k_0-1} \mathbb{E}\left[\|\mathbf{x}_k - \mathbf{1}\bar{x}_k\|^2\right]$$

$$+ \frac{4}{\mu} \sum_{k=0}^{k_0-1} \mathbb{E}\left[\left\|\frac{\tilde{\boldsymbol{u}}_{k+1}^{-\beta}}{n\bar{u}_{k+1}^{-\beta}}\nabla_y F\left(\mathbf{x}_k,\mathbf{y}_k;\xi_k^y\right)\right\|^2\right] + C \sum_{k=0}^{k_0-1} \mathbb{E}\left[\sqrt{\frac{1}{n}\|\mathbf{y}_k - \mathbf{1}\bar{y}_k\|^2}\right],$$

(33)

*where*

$$E_{1,k} := \frac{1-3\mu\gamma_y\bar{u}_{k+1}^{-\beta}/4}{2\gamma_y\bar{u}_{k+1}^{-\beta}n}\|\mathbf{y}_k - \mathbf{1}y^*\left(\bar{x}_k\right)\|^2 - \frac{\|\mathbf{y}_{k+1} - \mathbf{1}y^*\left(\bar{x}_{k+1}\right)\|^2}{\left(2+\mu\gamma_y\bar{u}_{k+1}^{-\beta}\right)\gamma_y\bar{u}_{k+1}^{-\beta}n}.$$

(34)

*Proof.* Using Young's inequality with parameter $\lambda_k$, we get

$$\frac{1}{n}\|\mathbf{y}_{k+1} - \mathbf{1}\bar{y}^*\left(\bar{x}_{k+1}\right)\|^2$$

$$\leqslant \frac{(1+\lambda_k)}{n}\|\mathbf{y}_{k+1} - \mathbf{1}y^*\left(\bar{x}_k\right)\|^2 + \left(1+\frac{1}{\lambda_k}\right)\|y^*\left(\bar{x}_k\right) - y^*\left(\bar{x}_{k+1}\right)\|^2.$$

(35)

Recalling that $\mathbf{y}_{k+1} = \mathcal{P}_{\mathcal{Y}}\left(W\left(\mathbf{y}_k + \gamma_y U_{k+1}^{-\beta}\nabla_y F\left(\mathbf{x}_k,\mathbf{y}_k;\xi_k^y\right)\right)\right)$, we further define

$$\hat{\mathbf{y}}_{k+1} = W\left(\mathbf{y}_k + \gamma_y U_{k+1}^{-\beta}\nabla_y F\left(\mathbf{x}_k,\mathbf{y}_k;\xi_k^y\right)\right).$$

Then, for the first term on the RHS of (35), by the non-expansiveness property of projection operator $\mathcal{P}_{\mathcal{Y}}(\cdot)$ (c.f., Lemma 1 in (Nedic et al., 2010)), we have

$$\frac{1}{n}\|\mathbf{y}_{k+1} - \mathbf{1}y^*\left(\bar{x}_k\right)\|^2$$

$$\leqslant \frac{1}{n}\|\hat{\mathbf{y}}_{k+1} - \mathbf{1}y^*\left(\bar{x}_k\right)\|^2 - \frac{1}{n}\|\mathbf{y}_{k+1} - \hat{\mathbf{y}}_{k+1}\|^2$$

$$\leqslant \frac{1}{n}\|\mathbf{y}_k - \mathbf{1}y^*\left(\bar{x}_k\right)\|^2 + \frac{\gamma_y^2}{n}\left\|U_{k+1}^{-\beta}\nabla_y F\left(\mathbf{x}_k,\mathbf{y}_k;\xi_k^y\right)\right\|^2$$

$$- \frac{1}{n}\sum_{i=1}^n 2\left\langle\gamma_y\bar{u}_{k+1}^{-\beta}g_{i,k}^y, y_{i,k} - y^*\left(\bar{x}_k\right)\right\rangle - \frac{1}{n}\sum_{i=1}^n 2\left\langle\gamma_y\left(u_{i,k+1}^{-\beta} - \bar{u}_{k+1}^{-\beta}\right)g_{i,k}^y, y_{i,k} - y^*\left(\bar{x}_k\right)\right\rangle,$$

(36)

wherein the last inequality we have used the fact $\|W\|_2^2 \leqslant 1$. Then, multiplying by $1/\left(\gamma_y \bar{u}_{k+1}^{-\beta}\right)$ on both sides of (35) we get

$$\frac{1}{n\gamma_y \bar{u}_{k+1}^{-\beta}} \left\|\mathbf{y}_{k+1} - \mathbf{1}y^*\left(\bar{x}_k\right)\right\|^2$$

$$\leqslant \frac{1+\lambda_k}{\lambda_k \gamma_y \bar{u}_{k+1}^{-\beta}} \left\|\bar{y}^*\left(\bar{x}_k\right) - \bar{y}^*\left(\bar{x}_{k+1}\right)\right\|^2$$

$$+ (1+\lambda_k)\left(\frac{1}{n\gamma_y \bar{u}_{k+1}^{-\beta}} \left\|\mathbf{y}_k - \mathbf{1}y^*\left(\bar{x}_k\right)\right\|^2 + \frac{\gamma_y}{n\bar{u}_{k+1}^{-\beta}} \left\|U_{k+1}^{-\beta}\nabla_y F\left(\mathbf{x}_k, \mathbf{y}_k; \xi_k^y\right)\right\|^2\right)$$

$$- (1+\lambda_k)\left(\frac{1}{n}\sum_{i=1}^{n} 2\left\langle g_{i,k}^y, y_{i,k} - y^*\left(\bar{x}_k\right)\right\rangle - \frac{1}{n}\sum_{i=1}^{n} 2\left\langle\left(\frac{u_{i,k+1}^{-\beta} - \bar{u}_{k+1}^{-\beta}}{\bar{u}_{k+1}^{-\beta}}\right) g_{i,k}^y, y_{i,k} - y^*\left(\bar{x}_k\right)\right\rangle\right). \tag{37}$$

For the inner-product terms on the RHS, taking expectation on both sides, we have

$$\frac{1}{n}\sum_{i=1}^{n} \mathbb{E}\left[-2\left\langle g_{i,k}^y, y_{i,k} - y^*\left(\bar{x}_k\right)\right\rangle\right]$$

$$= \frac{1}{n}\sum_{i=1}^{n} \mathbb{E}\left[-2\left\langle \nabla_y f_i\left(\bar{x}_k, y_{i,k}\right), y_{i,k} - y^*\left(\bar{x}_k\right)\right\rangle\right]$$

$$+ \frac{1}{n}\sum_{i=1}^{n} \mathbb{E}\left[-2\left\langle \nabla_y f_i\left(x_{i,k}, y_{i,k}\right) - \nabla_y f_i\left(\bar{x}_k, y_{i,k}\right), y_{i,k} - y^*\left(\bar{x}_k\right)\right\rangle\right]$$

$$\leqslant \frac{1}{n}\sum_{i=1}^{n} \mathbb{E}\left[-2\left(f_i\left(\bar{x}_k, y^*\left(\bar{x}_k\right)\right) - f_i\left(\bar{x}_k, y_{i,k}\right)\right) - \mu\left\|y_{i,k} - y^*\left(\bar{x}_k\right)\right\|^2\right] \tag{38}$$

$$+ \frac{1}{n}\sum_{i=1}^{n} \mathbb{E}\left[\frac{8}{\mu}\left\|\nabla_y f_i\left(x_{i,k}, y_{i,k}\right) - \nabla_y f_i\left(\bar{x}_k, y_{i,k}\right)\right\|^2 + \frac{\mu}{8}\left\|y_{i,k} - \bar{y}^*\left(\bar{x}_k\right)\right\|^2\right]$$

$$\leqslant \mathbb{E}\left[-2\left(f\left(\bar{x}_k, y^*\left(\bar{x}_k\right)\right) - f\left(\bar{x}_k, \bar{y}_k\right)\right)\right] + \frac{1}{n}\sum_{i=1}^{n} \mathbb{E}\left[-2\left(f_i\left(\bar{x}_k, \bar{y}_k\right) - f_i\left(\bar{x}_k, y_{i,k}\right)\right)\right]$$

$$+ \frac{8\kappa L}{n}\sum_{i=1}^{n} \mathbb{E}\left[\left\|x_{i,k} - \bar{x}_k\right\|^2\right] - \frac{7\mu}{8n}\sum_{i=1}^{n} \mathbb{E}\left[\left\|y_{i,k} - y^*\left(\bar{x}_k\right)\right\|^2\right],$$

where we have used Young's inequality and strong-concavity of $f_i$, and

$$\frac{1}{n}\sum_{i=1}^{n} \mathbb{E}\left[-2\left\langle\left(\frac{u_{i,k+1}^{-\beta} - \bar{u}_{k+1}^{-\beta}}{\bar{u}_{k+1}^{-\beta}}\right) g_{i,k}^y, y_{i,k} - y^*\left(\bar{x}_k\right)\right\rangle\right]$$

$$\leqslant \frac{1}{n}\sum_{i=1}^{n} \mathbb{E}\left[\frac{8}{\mu}\left\|\left(\frac{u_{i,k+1}^{-\beta} - \bar{u}_{k+1}^{-\beta}}{\bar{u}_{k+1}^{-\beta}}\right) g_{i,k}^y\right\|^2 + \frac{\mu}{8}\left\|y_{i,k} - y^*\left(\bar{x}_k\right)\right\|^2\right]. \tag{39}$$

For the consensus error of dual variable on the objective function, using strong-concavity of $f_i$ and Jensen's inequality, we have

$$\frac{1}{n}\sum_{i=1}^{n} -2\left(f_i\left(\bar{x}_k, \bar{y}_k\right) - f_i\left(\bar{x}_k, y_{i,k}\right)\right)$$

$$\leqslant \frac{1}{n}\sum_{i=1}^{n} 2\left\langle \nabla_y f_i\left(\bar{x}_k, \bar{y}_k\right), y_{i,k} - \bar{y}_k\right\rangle - \frac{\mu}{n}\left\|\mathbf{y}_k - \mathbf{1}\bar{y}_k\right\|^2 \tag{40}$$

$$\leqslant 2C\frac{1}{n}\sum_{i=1}^{n} \left\|y_{i,k} - \bar{y}_k\right\| \leqslant 2C\sqrt{\frac{1}{n}\left\|\mathbf{y}_k - \mathbf{1}\bar{y}_k\right\|^2}.$$

Letting $\lambda_k = \mu\gamma_y \bar{u}_{k+1}^{-\beta}/2$, we get

$$
\begin{aligned}
&\mathbb{E}\left[f\left(\bar{x}_k, \bar{y}^*\left(\bar{x}_k\right)\right) - f\left(\bar{x}_k, \bar{y}_k\right)\right] \\
&\leqslant \mathbb{E}\left[\frac{1 - 3\mu\gamma_y \bar{u}_{k+1}^{-\beta}/4}{2\gamma_y \bar{u}_{k+1}^{-\beta} n}\left\|\mathbf{y}_k - \mathbf{1}y^*\left(\bar{x}_k\right)\right\|^2 - \frac{\left\|\mathbf{y}_{k+1} - \mathbf{1}y^*\left(\bar{x}_{k+1}\right)\right\|^2}{\left(2 + \mu\gamma_y \bar{u}_{k+1}^{-\beta}\right)\gamma_y \bar{u}_{k+1}^{-\beta} n}\right] \\
&\quad + \frac{\gamma_x^2 \kappa^2 \left(1 + \zeta_v^2\right) G^{2\beta}}{n\mu\gamma_y^2}\mathbb{E}\left[\bar{v}_{k+1}^{-2\alpha}\left\|\nabla_x F\left(\mathbf{x}_k, \mathbf{y}_k; \xi_k^x\right)\right\|^2\right] \\
&\quad + \frac{\gamma_y\left(1 + \zeta_u^2\right)}{n}\sum_{i=1}^n \mathbb{E}\left[\bar{u}_{k+1}^{-\beta}\left\|\nabla_y F\left(\mathbf{x}_k, \mathbf{y}_k; \xi_k\right)\right\|^2\right] + \frac{4\kappa L}{n}\mathbb{E}\left[\left\|\mathbf{x}_k - \mathbf{1}\bar{y}_k\right\|^2\right] \\
&\quad + \frac{4}{\mu}\mathbb{E}\left[\left\|\frac{\tilde{\boldsymbol{u}}_{k+1}^{-\beta}}{n\bar{u}_{k+1}^{-\beta}}\nabla_y F\left(\mathbf{x}_k, \mathbf{y}_k; \xi_k^y\right)\right\|^2\right] + C\mathbb{E}\left[\sqrt{\frac{1}{n}\left\|\mathbf{y}_k - \mathbf{1}\bar{y}_k\right\|^2}\right].
\end{aligned}
\tag{41}
$$

By the $\kappa$-smoothness of $y^*$, we have

$$
\begin{aligned}
&\left\|y^*\left(\bar{x}_{k+1}\right) - y^*\left(\bar{x}_k\right)\right\|^2 \\
&\leqslant \kappa^2\left\|\bar{x}_{k+1} - \bar{x}_k\right\|^2 \\
&= \kappa^2\left\|\gamma_x \bar{v}_{k+1}^{-\alpha}\frac{\mathbf{1}^T}{n}\nabla_x F\left(\mathbf{x}_k, \mathbf{y}_k; \xi_k\right) - \gamma_x\frac{\left(\tilde{\boldsymbol{v}}_{k+1}^{-\alpha}\right)^T}{n}\nabla_x F\left(\mathbf{x}_k, \mathbf{y}_k; \xi_k^x\right)\right\|^2 \\
&\leqslant \frac{2\gamma_x^2 \kappa^2\left(1 + \zeta_v^2\right)\bar{v}_{k+1}^{-2\alpha}}{n}\left\|\nabla_x F\left(\mathbf{x}_k, \mathbf{y}_k; \xi_k^x\right)\right\|^2.
\end{aligned}
\tag{42}
$$

Telescoping the obtained terms from 0 to $k_0 - 1$ and noticing that $\bar{u}_k \leqslant G$ for $k \leqslant k_0 - 1$ we complete the proof. $\qquad\square$

For the second phase, i.e., $k \geqslant k_0$, we have the following lemma.

**Lemma 7** (Second phase). *Suppose Assumption 1-4 hold. If $\bar{u}_k \leqslant G, k = 0, 1, \cdots, k_0 - 1$, then we have*

$$
\begin{aligned}
&\sum_{k=k_0}^{K-1}\mathbb{E}\left[f\left(\bar{x}_k, \bar{y}^*\left(\bar{x}_k\right)\right) - f\left(\bar{x}_k, \bar{y}_k\right)\right] \\
&\leqslant \sum_{k=k_0}^{K-1}\mathbb{E}\left[E_{1,k}\right] + \frac{8\gamma_x^2 \kappa^2\left(1 + \zeta_v^2\right)}{\mu\gamma_y^2 G^{2\alpha - 2\beta}}\sum_{k=k_0}^{K-1}\left\|\nabla_x f\left(\bar{x}_k, \bar{y}_k\right)\right\|^2 \\
&\quad + \left(\frac{8\gamma_x^2 \kappa^2 L^2\left(1 + \zeta_v^2\right)}{n\mu\gamma_y^2 G^{2\alpha - 2\beta}} + \frac{4\kappa L}{n}\right)\sum_{k=k_0}^{K-1}\mathbb{E}\left[\Delta_k\right] \\
&\quad + \frac{\gamma_y\left(1 + \zeta_u^2\right)}{n}\mathbb{E}\left[\bar{u}_{k+1}^{-\beta}\left\|\nabla_y F\left(\mathbf{x}_k, \mathbf{y}_k; \xi_k\right)\right\|^2\right] + C\sum_{k=k_0}^{K-1}\mathbb{E}\left[\sqrt{\frac{1}{n}\left\|\mathbf{y}_k - \mathbf{1}\bar{y}_k\right\|^2}\right] \\
&\quad + \frac{\gamma_x^2\left(1 + \zeta_v^2\right)}{\gamma_y \bar{v}_1^{\alpha - \beta}}\left(\kappa^2 + \frac{2\gamma_x^2\left(1 + \zeta_v^2\right)C^2\hat{L}^2}{\mu\gamma_y \bar{v}_1^{2\alpha - \beta}}\right)\sum_{k=k_0}^{K-1}\mathbb{E}\left[\frac{\bar{v}_{k+1}^{-\alpha}}{n}\left\|\nabla_x F\left(\mathbf{x}_k, \mathbf{y}_k; \xi_k^x\right)\right\|^2\right] \\
&\quad + \frac{4\gamma_x \kappa\left(1 + \zeta_v\right)C^2}{\mu\gamma_y \bar{v}_1^\alpha}\mathbb{E}\left[\bar{u}_K^\beta\right] + \frac{4}{\mu}\sum_{k=k_0}^{K-1}\mathbb{E}\left[\left\|\frac{\tilde{\boldsymbol{u}}_{k+1}^{-\beta}}{n\bar{u}_{k+1}^{-\beta}}\nabla_y F\left(\mathbf{x}_k, \mathbf{y}_k; \xi_k^y\right)\right\|^2\right].
\end{aligned}
\tag{43}
$$

*Proof.* Firstly, by the non-expansiveness of projection operator, we have

$$
\begin{aligned}
&\|y_{i,k+1} - y^*(\bar{x}_{k+1})\|^2 \\
&\leqslant \|\hat{y}_{i,k+1} - y^*(\bar{x}_{k+1})\|^2 - \|y_{i,k+1} - \hat{y}_{i,k+1}\|^2 \\
&= \|\hat{y}_{i,k+1} - y^*(\bar{x}_k)\|^2 + \|y^*(\bar{x}_{k+1}) - y^*(\bar{x}_k)\|^2 \\
&\quad - 2\langle \hat{y}_{i,k+1} - y^*(\bar{x}_k), y^*(\bar{x}_{k+1}) - y^*(\bar{x}_k)\rangle \\
&= \|\hat{y}_{i,k+1} - y^*(\bar{x}_k)\|^2 + \|y^*(\bar{x}_{k+1}) - y^*(\bar{x}_k)\|^2 \\
&\quad - 2(\hat{y}_{i,k+1} - y^*(\bar{x}_k))^T \nabla y^*(\bar{x}_k)(\bar{x}_{k+1} - \bar{x}_k)^T \\
&\quad - 2(\hat{y}_{i,k+1} - y^*(\bar{x}_k))^T \left(y^*(\bar{x}_{k+1}) - y^*(\bar{x}_k) - \nabla y^*(\bar{x}_k)(\bar{x}_{k+1} - \bar{x}_k)^T\right).
\end{aligned}
\tag{44}
$$

Then, for the first inner-product term on the RHS, letting $\nabla_x \tilde{F}_k = \nabla_x F(\mathbf{x}_k, \mathbf{y}_k; \xi_k) - \nabla_x F(\mathbf{x}_k, \mathbf{y}_k)$, we get

$$
\begin{aligned}
&- 2(\hat{y}_{i,k+1} - y^*(\bar{x}_k))^T \nabla y^*(\bar{x}_k)(\bar{x}_{k+1} - \bar{x}_k)^T \\
&= 2\gamma_x (\hat{y}_{i,k+1} - y^*(\bar{x}_k))^T \nabla y^*(\bar{x}_k)(\nabla_x F(\mathbf{x}_k, \mathbf{y}_k))^T \left(\frac{\mathbf{1}\bar{v}_{k+1}^{-\alpha}}{n} + \frac{\tilde{\boldsymbol{v}}_{k+1}^{-\alpha}}{n}\right) \\
&\quad + 2\gamma_x (\hat{y}_{i,k+1} - y^*(\bar{x}_k))^T \nabla y^*(\bar{x}_k)\left(\nabla_x \tilde{F}_k\right)^T \left(\frac{\mathbf{1}\bar{v}_{k+1}^{-\alpha}}{n} + \frac{\tilde{\boldsymbol{v}}_{k+1}^{-\alpha}}{n}\right) \\
&\leqslant 2\gamma_x \kappa \|\hat{y}_{i,k+1} - y^*(\bar{x}_k)\| \left\|(\nabla_x F(\mathbf{x}_k, \mathbf{y}_k))^T \left(\frac{\mathbf{1}\bar{v}_{k+1}^{-\alpha}}{n} + \frac{\tilde{\boldsymbol{v}}_{k+1}^{-\alpha}}{n}\right)\right\| \\
&\quad + 2\gamma_x (\hat{y}_{i,k+1} - y^*(\bar{x}_k))^T \nabla y^*(\bar{x}_k)\left(\nabla_x \tilde{F}_k\right)^T \left(\frac{\mathbf{1}\bar{v}_{k+1}^{-\alpha}}{n} + \frac{\tilde{\boldsymbol{v}}_{k+1}^{-\alpha}}{n}\right).
\end{aligned}
\tag{45}
$$

wherein the last inequality we have used the fact that $y^*$ is $\kappa$-Lipschitz. Then, using Young's inequality with parameter $\lambda_k$, we get

$$
\begin{aligned}
&- 2(\hat{y}_{i,k+1} - y^*(\bar{x}_k))^T \nabla y^*(\bar{x}_k)(\bar{x}_{k+1} - \bar{x}_k)^T \\
&\leqslant \lambda_k \|\hat{y}_{i,k+1} - y^*(\bar{x}_k)\|^2 \\
&\quad + \frac{2\gamma_x^2 \bar{v}_{k+1}^{-2\alpha} \kappa^2}{\lambda_k} \left(\left\|\frac{\mathbf{1}^T}{n}\nabla_x F(\mathbf{x}_k, \mathbf{y}_k)\right\|^2 + \left\|\frac{(\tilde{\boldsymbol{v}}_{k+1}^{-\alpha})^T}{n\bar{v}_{k+1}^{-\alpha}}\nabla_x F(\mathbf{x}_k, \mathbf{y}_k)\right\|^2\right) \\
&\quad + 2\gamma_x (\hat{y}_{i,k+1} - y^*(\bar{x}_k))^T \nabla y^*(\bar{x}_k)\left(\nabla_x \tilde{F}_k\right)^T \left(\frac{\mathbf{1}\bar{v}_{k+1}^{-\alpha}}{n} + \frac{\tilde{\boldsymbol{v}}_{k+1}^{-\alpha}}{n}\right).
\end{aligned}
\tag{46}
$$

For the second inner-product term on the RHS, noticing that $y^*$ is $\hat{L} = \kappa(1+\kappa)^2$ smooth given in Lemma 2, we have

$$
\begin{aligned}
&2(\hat{y}_{i,k+1} - y^*(\bar{x}_k))^T \left(y^*(\bar{x}_k) - y^*(\bar{x}_{k+1}) + \nabla y^*(\bar{x}_k)(\bar{x}_{k+1} - \bar{x}_k)^T\right) \\
&\leqslant 2\|\hat{y}_{i,k+1} - y^*(\bar{x}_k)\| \|y^*(\bar{x}_k) - y^*(\bar{x}_{k+1}) + \nabla y^*(\bar{x}_k)(\bar{x}_{k+1} - \bar{x}_k)\|^2 \\
&\leqslant 2\|\hat{y}_{i,k+1} - y^*(\bar{x}_k)\| \frac{\hat{L}}{2} \|\bar{x}_{k+1} - \bar{x}_k\|^2 \\
&\leqslant \gamma_x^2 \hat{L} \|\hat{y}_{i,k+1} - y^*(\bar{x}_k)\| \left\|\left(\frac{\bar{v}_{k+1}^{-\alpha}\mathbf{1}^T}{n} + \frac{(\tilde{\boldsymbol{v}}_{k+1}^{-\alpha})^T}{n}\right)\nabla_x F(\mathbf{x}_k, \mathbf{y}_k; \xi_k^x)\right\|^2 \\
&\leqslant \gamma_x^2 \hat{L} \|\hat{y}_{i,k+1} - y^*(\bar{x}_k)\| \frac{2\bar{v}_{k+1}^{-2\alpha}(1+\zeta_v^2)C}{n} \|\nabla_x F(\mathbf{x}_k, \mathbf{y}_k; \xi_k^x)\| \\
&\leqslant \tau \gamma_x^2 \bar{v}_{k+1}^{-2\alpha}(1+\zeta_v^2)C^2 \hat{L} \|\hat{y}_{i,k+1} - y^*(\bar{x}_k)\|^2 + \frac{\gamma_x^2 \bar{v}_{k+1}^{-2\alpha}(1+\zeta_v^2)\hat{L}}{\tau n} \|\nabla_x F(\mathbf{x}_k, \mathbf{y}_k; \xi_k^x)\|^2,
\end{aligned}
\tag{47}
$$

wherein the last inequality we have used Young's inequality with parameter $\tau$. Plugging the obtained inequalities into (44), we get

$$
\begin{aligned}
&\|y_{i,k+1} - y^* (\bar{x}_{k+1})\|^2 \\
&\leqslant \left(1 + \lambda_k + \tau \gamma_x^2 \bar{v}_{k+1}^{-2\alpha} \left(1 + \zeta_v^2\right) C^2 \hat{L}\right) \|\hat{y}_{i,k+1} - y^* (\bar{x}_k)\|^2 \\
&\quad + \frac{\gamma_x^2 \bar{v}_{k+1}^{-2\alpha} \left(1 + \zeta_v^2\right)}{n} \left(2\kappa^2 + \frac{\hat{L}}{\tau}\right) \|\nabla_x F (\mathbf{x}_k, \mathbf{y}_k; \xi_k)\|^2 \\
&\quad + \frac{2\gamma_x^2 \bar{v}_{k+1}^{-2\alpha} \kappa^2}{\lambda_k} \left(\left\|\frac{\mathbf{1}^T}{n} \nabla_x F (\mathbf{x}_k, \mathbf{y}_k)\right\|^2 + \left\|\frac{\left(\tilde{\boldsymbol{v}}_{k+1}^{-\alpha}\right)^T}{n \bar{v}_{k+1}^{-\alpha}} \nabla_x F (\mathbf{x}_k, \mathbf{y}_k)\right\|^2\right) \\
&\quad + 2\gamma_x \left(\hat{y}_{i,k+1} - y^* (\bar{x}_k)\right)^T \nabla y^* (\bar{x}_k) \left(\nabla_x \tilde{F}\right)^T \left(\frac{\mathbf{1}\bar{v}_{k+1}^{-\alpha}}{n} + \frac{\tilde{\boldsymbol{v}}_{k+1}^{-\alpha}}{n}\right).
\end{aligned}
\tag{48}
$$

Setting the parameters for Young's inequalities we used as follows,

$$
\lambda_k = \frac{\mu \gamma_y \bar{u}_{k+1}^{-\beta}}{4}, \quad \tau = \frac{\mu \gamma_y \bar{v}_0^{2\alpha - \beta}}{4\gamma_x^2 \left(1 + \zeta_v^2\right) C^2 \hat{L}},
\tag{49}
$$

then we get

$$
\begin{aligned}
&\|y_{i,k+1} - y^* (\bar{x}_{k+1})\|^2 \\
&\leqslant \left(1 + \frac{\mu \gamma_y \bar{u}_{k+1}^{-\beta}}{2}\right) \|\hat{y}_{i,k+1} - y^* (\bar{x}_k)\|^2 \\
&\quad + \frac{\gamma_x^2 \left(1 + \zeta_v^2\right)}{n} \left(2\kappa^2 + \frac{4\gamma_x^2 \left(1 + \zeta_v^2\right) C^2 \hat{L}^2}{\mu \gamma_y \bar{v}_0^{2\alpha - \beta}}\right) \bar{v}_{k+1}^{-2\alpha} \|\nabla_x F (\mathbf{x}_k, \mathbf{y}_k; \xi_k)\|^2 \\
&\quad + \frac{8\gamma_x^2 \bar{v}_{k+1}^{-2\alpha} \kappa^2}{\mu \gamma_y \bar{u}_{k+1}^{-\beta}} \left(\left\|\frac{\mathbf{1}^T}{n} \nabla_x F (\mathbf{x}_k, \mathbf{y}_k)\right\|^2 + \left\|\frac{\left(\tilde{\boldsymbol{v}}_{k+1}^{-\alpha}\right)^T}{n \bar{v}_{k+1}^{-\alpha}} \nabla_x F (\mathbf{x}_k, \mathbf{y}_k)\right\|^2\right) \\
&\quad + 2\gamma_x \left(\hat{y}_{i,k+1} - y^* (\bar{x}_k)\right)^T \nabla y^* (\bar{x}_k) \left(\nabla_x \tilde{F}_k\right)^T \left(\frac{\mathbf{1}\bar{v}_{k+1}^{-\alpha}}{n} + \frac{\tilde{\boldsymbol{v}}_{k+1}^{-\alpha}}{n}\right).
\end{aligned}
\tag{50}
$$

Recalling that

$$
\begin{aligned}
&\frac{1}{n} \sum_{i=1}^n \mathbb{E}\left[\frac{1}{\gamma_y \bar{u}_{k+1}^{-\beta}} \|\hat{y}_{i,k+1} - \bar{y}^* (\bar{x}_k)\|^2\right] \\
&\leqslant \frac{1}{n} \sum_{i=1}^n \mathbb{E}\left[\frac{1 - 3\mu \gamma_y \bar{u}_{k+1}^{-\beta}/4}{\gamma_y \bar{u}_{k+1}^{-\beta}} \|y_{i,k} - \bar{y}^* (\bar{x}_k)\|^2\right] + \frac{8\kappa L}{n} \mathbb{E}\left[\|\mathbf{x}_k - \mathbf{1}\bar{y}_k\|^2\right] \\
&\quad + \frac{2\gamma_y \left(1 + \zeta_u^2\right)}{n} \mathbb{E}\left[\bar{u}_{k+1}^{-\beta} \|\nabla_y F (\mathbf{x}_k, \mathbf{y}_k; \xi_k)\|^2\right] - \mathbb{E}\left[2 \left(f \left(\bar{x}_k, \bar{y}^* (\bar{x}_k)\right) - f \left(\bar{x}_k, \bar{y}_k\right)\right)\right] \\
&\quad + \frac{8}{\mu} \mathbb{E}\left[\left\|\frac{\tilde{\boldsymbol{u}}_{k+1}^{-\beta}}{n \bar{u}_{k+1}^{-\beta}} \nabla_y F (\mathbf{x}_k, \mathbf{y}_k; \xi_k^y)\right\|^2\right] + 2C \mathbb{E}\left[\sqrt{\frac{1}{n} \|\mathbf{y}_k - \mathbf{1}\bar{y}_k\|^2}\right],
\end{aligned}
$$

and multiplying by $\frac{2}{\left(2+\mu\gamma_y\bar{u}_{k+1}^{-\beta}\right)\gamma_y\bar{u}_{k+1}^{-\beta}}$ on both sides of (50), we obtain that

$$
\begin{aligned}
&\mathbb{E}\left[f\left(\bar{x}_k, \bar{y}^*\left(\bar{x}_k\right)\right) - f\left(\bar{x}_k, \bar{y}_k\right)\right] \\
&\leqslant \mathbb{E}\left[E_{1,k}\right] + \frac{\gamma_y\left(1+\zeta_u^2\right)}{n}\mathbb{E}\left[\bar{u}_{k+1}^{-\beta}\left\|\nabla_y F\left(\mathbf{x}_k, \mathbf{y}_k; \xi_k\right)\right\|^2\right] + \frac{4\kappa L}{n}\mathbb{E}\left[\left\|\mathbf{x}_k - \mathbf{1}\bar{y}_k\right\|^2\right] \\
&\quad + \frac{4}{\mu}\mathbb{E}\left[\left\|\frac{\tilde{\boldsymbol{u}}_{k+1}^{-\beta}}{n\bar{u}_{k+1}^{-\beta}}\nabla_y F\left(\mathbf{x}_k, \mathbf{y}_k; \xi_k^y\right)\right\|^2\right] + C\mathbb{E}\left[\sqrt{\frac{1}{n}\left\|\mathbf{y}_k - \mathbf{1}\bar{y}_k\right\|^2}\right] \\
&\quad + \underbrace{\mathbb{E}\left[\frac{4\gamma_x^2\bar{v}_{k+1}^{-2\alpha}\kappa^2}{\mu\gamma_y^2\bar{u}_{k+1}^{-2\beta}}\left(\left\|\frac{\mathbf{1}^T}{n}\nabla_x F\left(\mathbf{x}_k, \mathbf{y}_k\right)\right\|^2 + \left\|\frac{\left(\tilde{\boldsymbol{v}}_{k+1}^{-\alpha}\right)^T}{n\bar{v}_{k+1}^{-\alpha}}\nabla_x F\left(\mathbf{x}_k, \mathbf{y}_k\right)\right\|^2\right)\right]}_{\mathbb{E}[E_{2,k}]} \\
&\quad + \underbrace{\frac{\gamma_x^2\left(1+\zeta_v^2\right)}{n}\left(\kappa^2 + \frac{2\gamma_x^2\left(1+\zeta_v^2\right)C^2\hat{L}^2}{\mu\gamma_y\bar{v}_1^{2\alpha-\beta}}\right)\mathbb{E}\left[\frac{\bar{v}_{k+1}^{-2\alpha}}{\gamma_y\bar{u}_{k+1}^{-\beta}}\left\|\nabla_x F\left(\mathbf{x}_k, \mathbf{y}_k; \xi_k\right)\right\|^2\right]}_{\mathbb{E}[E_{3,k}]} \\
&\quad + \underbrace{\frac{1}{n}\sum_{i=1}^n\mathbb{E}\left[\frac{\gamma_x}{\gamma_y\bar{u}_{k+1}^{-\beta}}\left(\hat{y}_{i,k+1} - y^*\left(\bar{x}_k\right)\right)^T\nabla y^*\left(\bar{x}_k\right)\left(\nabla_x\tilde{F}_k\right)^T\left(\frac{\mathbf{1}\bar{v}_{k+1}^{-\alpha}}{n} + \frac{\tilde{\boldsymbol{v}}_{k+1}^{-\alpha}}{n}\right)\right]}_{\mathbb{E}[E_{4,k}]}.
\end{aligned}
\tag{51}
$$

Telescoping the terms from $t_0$ to $K-1$, we get

$$
\begin{aligned}
&\sum_{k=k_0}^{K-1}\mathbb{E}\left[f\left(\bar{x}_k, \bar{y}^*\left(\bar{x}_k\right)\right) - f\left(\bar{x}_k, \bar{y}_k\right)\right] \\
&\leqslant \sum_{k=k_0}^{K-1}\mathbb{E}\left[E_{1,k}\right] + \sum_{k=k_0}^{K-1}\mathbb{E}\left[E_{2,k}\right] + \sum_{k=k_0}^{K-1}\mathbb{E}\left[E_{3,k}\right] + \sum_{k=k_0}^{K-1}\mathbb{E}\left[E_{4,k}\right] \\
&\quad + \frac{\gamma_y\left(1+\zeta_u^2\right)}{n}\mathbb{E}\left[\bar{u}_{k+1}^{-\beta}\left\|\nabla_y F\left(\mathbf{x}_k, \mathbf{y}_k; \xi_k\right)\right\|^2\right] + \frac{4\kappa L}{n}\sum_{k=k_0}^{K-1}\mathbb{E}\left[\left\|\mathbf{x}_k - \mathbf{1}\bar{y}_k\right\|^2\right] \\
&\quad + \frac{4}{\mu}\mathbb{E}\left[\left\|\frac{\tilde{\boldsymbol{u}}_{k+1}^{-\beta}}{n\bar{u}_{k+1}^{-\beta}}\nabla_y F\left(\mathbf{x}_k, \mathbf{y}_k; \xi_k^y\right)\right\|^2\right] + C\sum_{k=k_0}^{K-1}\mathbb{E}\left[\sqrt{\frac{1}{n}\left\|\mathbf{y}_k - \mathbf{1}\bar{y}_k\right\|^2}\right].
\end{aligned}
\tag{52}
$$

Next we need to further bound the running sums of $\mathbb{E}\left[E_{2,k}\right]$, $\mathbb{E}\left[E_{3,k}\right]$ and $\mathbb{E}\left[E_{4,k}\right]$ respectively. For $\mathbb{E}\left[E_{2,k}\right]$, with the help of Assumption 2 and noticing that $\bar{u}_k \leqslant G, k = 0, 1, \cdots, k_0 - 1$, we get

$$
\begin{aligned}
&\sum_{k=k_0}^{K-1}\mathbb{E}\left[E_{2,k}\right] \\
&\leqslant \sum_{k=k_0}^{K-1}\mathbb{E}\left[\frac{4\gamma_x^2\bar{v}_{k+1}^{-2\alpha}\kappa^2}{\mu\gamma_y^2\bar{u}_{k+1}^{-2\beta}}\left(\left\|\frac{\mathbf{1}^T}{n}\nabla_x F\left(\mathbf{x}_k, \mathbf{y}_k\right)\right\|^2 + \left\|\frac{\left(\tilde{\boldsymbol{v}}_{k+1}^{-\alpha}\right)^T}{n\bar{v}_{k+1}^{-\alpha}}\nabla_x F\left(\mathbf{x}_k, \mathbf{y}_k\right)\right\|^2\right)\right] \\
&\leqslant \frac{8\gamma_x^2\kappa^2\left(1+\zeta_v^2\right)}{\mu\gamma_y^2 G^{2\alpha-2\beta}}\sum_{k=k_0}^{K-1}\mathbb{E}\left[\left\|\nabla_x f\left(\bar{x}_k, \bar{y}_k\right)\right\|^2 + \frac{L^2}{n}\Delta_k\right].
\end{aligned}
\tag{53}
$$

Then, for the term $\mathbb{E}\left[E_{3,k}\right]$, noticing that $\bar{u}_{k+1} \leqslant \bar{v}_{k+1}$ and $\bar{v}_{k+1} \geqslant \bar{v}_1$, we have

$$
\begin{aligned}
&\sum_{k=k_0}^{K-1} \mathbb{E}\left[E_{3,k}\right] \\
&\leqslant \sum_{k=k_0}^{K-1} \mathbb{E}\left[\frac{\gamma_x^2\left(1+\zeta_v^2\right)}{n\gamma_y}\left(\kappa^2 + \frac{2\gamma_x^2\left(1+\zeta_v^2\right)C^2\hat{L}^2}{\mu\gamma_y\bar{v}_1^{2\alpha-\beta}}\right)\frac{\bar{v}_{k+1}^{-2\alpha}}{\bar{u}_{k+1}^{-\beta}}\left\|\nabla_x F\left(\mathbf{x}_k,\mathbf{y}_k;\xi_k^x\right)\right\|^2\right] \\
&\leqslant \frac{\gamma_x^2\left(1+\zeta_v^2\right)}{\gamma_y\bar{v}_1^{\alpha-\beta}}\left(\kappa^2 + \frac{2\gamma_x^2\left(1+\zeta_v^2\right)C^2\hat{L}^2}{\mu\gamma_y\bar{v}_1^{2\alpha-\beta}}\right)\sum_{k=k_0}^{K-1}\mathbb{E}\left[\frac{\bar{v}_{k+1}^{-\alpha}}{n}\left\|\nabla_x F\left(\mathbf{x}_k,\mathbf{y}_k;\xi_k^x\right)\right\|^2\right].
\end{aligned}
\tag{54}
$$

For the term $E_{4,k}$, we denote

$$
e_k := \frac{\gamma_x}{\gamma_y\bar{u}_{k+1}^{-\beta}}\left(\frac{1}{n}\sum_{i=1}^{n}\left(\hat{y}_{i,k+1}-y^*\left(\bar{x}_k\right)\right)^T\right)\nabla y^*\left(\bar{x}_k\right)\left(\nabla_x\tilde{F}_k\right)^T\left(\frac{\mathbf{1}}{n}+\frac{\tilde{\boldsymbol{v}}_{k+1}^{-\alpha}}{n\bar{v}_{k+1}^{-\alpha}}\right),
$$

then we have

$$
\begin{aligned}
\left|e_k\right| &\leqslant \frac{\gamma_x\kappa}{\gamma_y\bar{u}_{k+1}^{-\beta}}\frac{1}{n}\sum_{i=1}^{n}\left\|\hat{y}_{i,k+1}-y^*\left(\bar{x}_k\right)\right\|\left\|\left(\nabla_x\tilde{F}_k\right)^T\left(\frac{\mathbf{1}}{n}+\frac{\tilde{\boldsymbol{v}}_{k+1}^{-\alpha}}{n\bar{v}_{k+1}^{-\alpha}}\right)\right\| \\
&\leqslant \frac{\gamma_x\kappa\left(1+\zeta_v\right)}{\gamma_y\sqrt{n}\bar{u}_{k+1}^{-\beta}}\left(\frac{1}{n}\sum_{i=1}^{n}\frac{1}{\mu}\left\|\nabla_y f\left(\bar{x}_k,\hat{y}_{i,k+1}\right)-\nabla_y f\left(\bar{x}_k,y^*\right)\right\|\right)\left\|\nabla_x\tilde{F}\right\| \\
&\leqslant \underbrace{\frac{2\gamma_x\kappa\left(1+\zeta_v\right)C^2\bar{u}_K^{\beta}}{\mu\gamma_y}}_{M},
\end{aligned}
\tag{55}
$$

where we have used the Lipschitz continuity of $y^*$ given in Lemma 2 and Assumption 3. Then, noticing that $\mathbb{E}\left[\nabla_x\tilde{F}_k\right]=0$, we obtain

$$
\begin{aligned}
\sum_{k=k_0}^{K-1}\mathbb{E}\left[E_{4,k}\right] &= \sum_{k=k_0}^{K-1}\mathbb{E}\left[e_k\bar{v}_{k+1}^{-\alpha}\right] \\
&= \mathbb{E}\left[e_{k_0}\bar{v}_{k_0+1}^{-\alpha}\right]+\underbrace{\sum_{k=k_0+1}^{K-1}\mathbb{E}\left[e_k\bar{v}_k^{-\alpha}\right]}_{0}+\sum_{k=k_0+1}^{K-1}\mathbb{E}\left[-e_k\underbrace{\left(\bar{v}_k^{-\alpha}-\bar{v}_{k+1}^{-\alpha}\right)}_{>0}\right] \\
&\leqslant \mathbb{E}\left[M\bar{v}_{k_0+1}^{-\alpha}\right]+\sum_{k=k_0+1}^{K-1}\mathbb{E}\left[M\left(\bar{v}_k^{-\alpha}-\bar{v}_{k+1}^{-\alpha}\right)\right] \\
&\leqslant 2\mathbb{E}\left[M\bar{v}_{k_0+1}^{-\alpha}\right] \leqslant \frac{4\gamma_x\kappa\left(1+\zeta_v\right)C^2}{\mu\gamma_y\bar{v}_1^{\alpha}}\mathbb{E}\left[\bar{u}_K^{\beta}\right].
\end{aligned}
\tag{56}
$$

Therefore, combining the obtained inequalities, we complete the proof. $\qquad\square$

Now, it remains to bound the term $E_{1,k}$.

**Lemma 8.** *Suppose Assumption 1-4 hold. Then, we have*

$$
\sum_{k=0}^{K-1}\mathbb{E}\left[E_{1,k}\right] \leqslant \frac{1}{2\gamma_y\bar{u}_1^{-\beta}n}\left\|\mathbf{y}_0-\mathbf{1}y^*\left(\bar{x}_0\right)\right\|^2 + \frac{2\left(4\beta C^2\right)^{2+\frac{1}{1-\beta}}}{\mu^{3+\frac{1}{1-\beta}}\gamma_y^{2+\frac{1}{1-\beta}}\bar{u}_1^{2-2\beta}}.
\tag{57}
$$

*Proof.* Recalling the definition of $E_{1,k}$ as given in (34), we have

$$\sum_{k=0}^{K-1} \mathbb{E}\left[\frac{1 - 3\mu\gamma_y\bar{u}_{k+1}^{-\beta}/4}{2\gamma_y\bar{u}_{k+1}^{-\beta}n}\left\|\mathbf{y}_k - \mathbf{1}y^*\left(\bar{x}_k\right)\right\|^2 - \frac{\left\|\mathbf{y}_{k+1} - \mathbf{1}y^*\left(\bar{x}_{k+1}\right)\right\|^2}{\left(2 + \mu\gamma_y\bar{u}_{k+1}^{-\beta}\right)\gamma_y\bar{u}_{k+1}^{-\beta}n}\right]$$

$$\leqslant \frac{1 - 3\mu\gamma_y\bar{u}_1^{-\beta}/4}{2\gamma_y\bar{u}_1^{-\beta}n}\left\|\mathbf{y}_0 - \mathbf{1}y^*\left(\bar{x}_0\right)\right\|^2$$

$$+ \sum_{k=1}^{K-1} \mathbb{E}\left[\left(\frac{1 - 3\mu\gamma_y\bar{u}_{k+1}^{-\beta}/4}{2\gamma_y\bar{u}_{k+1}^{-\beta}n} - \frac{1}{2n\gamma_y\bar{u}_k^{-\beta}\left(2 + \mu\gamma_y\bar{u}_k^{-\beta}\right)}\right)\left\|\mathbf{y}_k - \mathbf{1}y^*\left(\bar{x}_k\right)\right\|^2\right]$$

$$\leqslant \frac{1 - 3\mu\gamma_y\bar{u}_1^{-\beta}/4}{2\gamma_y\bar{u}_1^{-\beta}n}\left\|\mathbf{y}_0 - \mathbf{1}y^*\left(\bar{x}_0\right)\right\|^2$$

$$+ \sum_{k=1}^{K-1} \mathbb{E}\left[\left(\frac{1}{2\gamma_y\bar{u}_{k+1}^{-\beta}} - \frac{1}{4\gamma_y\bar{u}_k^{-\beta}} - \frac{\mu}{8} + \underbrace{\frac{\mu}{2\left(2 + \mu\gamma_y\bar{u}_k^{-\beta}\right)} - \frac{\mu}{2}}_{<0}\right)\frac{1}{n}\left\|\mathbf{y}_k - \mathbf{1}y^*\left(\bar{x}_k\right)\right\|^2\right]. \tag{58}$$

Next, we show that the term $\frac{1}{2\gamma_y\bar{u}_{k+1}^{-\beta}} - \frac{1}{2\gamma_y\bar{u}_k^{-\beta}} - \frac{\mu}{8}$ is positive for only a constant number of iterations. If the term is positive at iteration $k$, then we have

$$0 < \frac{\bar{u}_{k+1}^{\beta}}{2\gamma_y} - \frac{\bar{u}_k^{\beta}}{2\gamma_y} - \frac{\mu}{8}$$

$$\leqslant \bar{u}_k^{\beta}\frac{\left(1 + \left\|\nabla_y F\left(\mathbf{x}_k, \mathbf{y}_k; \xi_k^y\right)\right\|^2/n\bar{u}_k^{\beta}\right)^{\beta}}{2\gamma_y} - \frac{\bar{u}_k^{\beta}}{2\gamma_y} - \frac{\mu}{8}$$

$$\leqslant \bar{u}_k^{\beta}\frac{\left(1 + \beta\left\|\nabla_y F\left(\mathbf{x}_k, \mathbf{y}_k; \xi_k^y\right)\right\|^2/n\bar{u}_k\right)}{2\gamma_y} - \frac{\bar{u}_k^{\beta}}{2\gamma_y} - \frac{\mu}{8} \tag{59}$$

$$= \frac{\beta\left\|\nabla_y F\left(\mathbf{x}_k, \mathbf{y}_k; \xi_k^y\right)\right\|^2}{2\gamma_y n\bar{u}_k^{1-\beta}} - \frac{\mu}{8},$$

wherein the last inequality we used Bernoulli's inequality. Then we have the following two conditions,

$$\begin{cases}\frac{1}{n}\left\|\nabla_y F\left(\mathbf{x}_k, \mathbf{y}_k; \xi_k\right)\right\|^2 \geqslant \frac{\gamma_y\bar{u}_{k+1}^{1-\beta}}{4\beta} \geqslant \frac{\gamma_y\bar{u}_1^{1-\beta}}{4\beta}, \\ \frac{4\beta G^2}{\mu\gamma_y} \geqslant \frac{4\beta\left\|\nabla_y F\left(\mathbf{x}_k, \mathbf{y}_k; \xi_k^y\right)\right\|^2}{\mu\gamma_y n} \geqslant \bar{u}_{k+1}^{1-\beta},\end{cases} \tag{60}$$

which implies that we have at most

$$\left(\frac{4\beta C^2}{\mu\gamma_y}\right)^{\frac{1}{1-\beta}}\frac{4\beta}{\mu\gamma_y\bar{u}_1^{1-\beta}} \tag{61}$$

constant number of iterations when the term is positive. Furthermore, when the term is positive, by the inequality (59), we have

$$\left(\frac{1}{2\gamma_y\bar{u}_{k+1}^{-\beta}} - \frac{1}{2\gamma_y\bar{u}_k^{-\beta}} - \frac{\mu}{8}\right)\frac{1}{n}\left\|\mathbf{y}_k - \mathbf{1}y^*\left(\bar{x}_k\right)\right\|^2$$

$$\leqslant \frac{\beta\left\|\nabla_y F\left(\mathbf{x}_k, \mathbf{y}_k; \xi_k^y\right)\right\|^2}{2\gamma_y n\bar{u}_1^{1-\beta}}\frac{1}{n}\left\|\mathbf{y}_k - \mathbf{1}y^*\left(\bar{x}_k\right)\right\|^2$$

$$\leqslant \frac{\beta C^2}{2\mu^2\gamma_y\bar{u}_1^{1-\beta}}\frac{1}{n}\sum_{i=1}^{n}\left\|\nabla_y f_i\left(\bar{x}_k, y_{i,k}\right) - \nabla_y f_i\left(\bar{x}_k, y^*\right)\right\|^2 \tag{62}$$

$$\leqslant \frac{2\beta C^4}{\mu^2\gamma_y\bar{u}_1^{1-\beta}},$$

where we have used the concavity of $f_i$ in $y$ and Assumption 3. Then, we have

$$\sum_{k=1}^{K-1} \mathbb{E}\left[\left(\frac{1}{2\gamma_y \bar{u}_{k+1}^{-\beta}} - \frac{1}{2\gamma_y \bar{u}_k^{-\beta}} - \frac{\mu}{8}\right) \frac{1}{n} \left\|\mathbf{y}_k - \mathbf{1}y^*\left(\bar{x}_k\right)\right\|^2\right]$$

$$\leqslant \frac{2\beta C^4}{\mu^2 \gamma_y \bar{u}_1^{1-\beta}} \left(\frac{4\beta C^2}{\mu \gamma_y}\right)^{\frac{1}{1-\beta}} \frac{4\beta}{\mu \gamma_y \bar{u}_1^{1-\beta}} \tag{63}$$

$$\leqslant \frac{2\left(4\beta C^2\right)^{2+\frac{1}{1-\beta}}}{\mu^{3+\frac{1}{1-\beta}} \gamma_y^{2+\frac{1}{1-\beta}} \bar{u}_1^{2-2\beta}},$$

which completes the proof. $\qquad\square$

Next, we show in the following lemma that the inconsistency terms, as described in (5), exhibit asymptotic convergence for the proposed D-AdaST algorithm.

**Lemma 9** (Convergence of inconsistency terms)**.** *Suppose Assumption 1-4 hold. For the proposed D-AdaST in Algorithm 1, we have*

$$\frac{1}{K} \sum_{k=0}^{K-1} \mathbb{E}\left[\left\|\frac{\left(\tilde{\boldsymbol{v}}_{k+1}^{-\alpha}\right)^T}{n\bar{v}_{k+1}^{-\alpha}} \nabla_x F\left(\mathbf{x}_k, \mathbf{y}_k; \xi_k^x\right)\right\|^2\right] \leqslant \sqrt{\frac{1}{n^{1-\alpha}} \left(\frac{4\rho_W}{\left(1-\rho_W\right)^2}\right)^\alpha \frac{\left(1+\zeta_v\right)\zeta_v C^{2-\alpha}}{\left(1-\alpha\right)K^\alpha}},$$

$$\tag{64}$$

*and*

$$\frac{1}{K} \sum_{k=0}^{K-1} \mathbb{E}\left[\left\|\frac{\left(\tilde{\boldsymbol{u}}_{k+1}^{-\beta}\right)^T}{n\bar{u}_{k+1}^{-\beta}} \nabla_y F\left(\mathbf{x}_k, \mathbf{y}_k; \xi_k^y\right)\right\|^2\right] \leqslant \sqrt{\frac{1}{n^{1-\beta}} \left(\frac{4\rho_W}{\left(1-\rho_W\right)^2}\right)^\beta \frac{\left(1+\zeta_u\right)\zeta_u C^{2-\beta}}{\left(1-\beta\right)K^\beta}}.$$

$$\tag{65}$$

*Proof.* By the definition of $v_{i,k}$ in (3), we have

$$\mathbb{E}\left[\left\|\frac{\left(\tilde{\boldsymbol{v}}_{k+1}^{-\alpha}\right)^T}{n\bar{v}_{k+1}^{-\alpha}} \nabla_x F\left(\mathbf{x}_k, \mathbf{y}_k; \xi_k^x\right)\right\|^2\right]$$

$$\leqslant \mathbb{E}\left[\frac{1}{n^2} \sum_{i=1}^n \left(\bar{v}_{k+1}^\alpha - v_{i,k+1}^\alpha\right)^2 \frac{\left\|g_{i,k}^x\right\|^2}{v_{i,k+1}^{2\alpha}}\right] \tag{66}$$

$$\leqslant \mathbb{E}\left[\frac{1}{n^2} \sum_{i=1}^n \left(\bar{v}_{k+1}^\alpha - v_{i,k+1}^\alpha\right)^2 \frac{\bar{v}_{k+1}^\alpha}{v_{i,k+1}^{2\alpha}} \frac{\left\|g_{i,k}^x\right\|^2}{\bar{v}_{k+1}^\alpha}\right].$$

Noticing that $\frac{\left|\bar{v}^{\alpha}_{k+1}-v^{\alpha}_{i,k+1}\right|}{v^{\alpha}_{i,k+1}} \leqslant \zeta_v$, we have

$$
\mathbb{E}\left[\left\|\frac{\left(\tilde{\boldsymbol{v}}^{-\alpha}_{k+1}\right)^T}{n\bar{v}^{-\alpha}_{k+1}}\nabla_x F\left(\mathbf{x}_k,\mathbf{y}_k;\xi^x_k\right)\right\|^2\right]
$$

$$
\leqslant \mathbb{E}\left[\frac{1}{n^2}\sum_{i=1}^{n}\left(\bar{v}^{\alpha}_{k+1}-v^{\alpha}_{i,k+1}\right)^2\left(\frac{\bar{v}^{\alpha}_{k+1}-v^{\alpha}_{i,k+1}}{v^{2\alpha}_{i,k+1}}+\frac{1}{v^{\alpha}_{i,k+1}}\right)\frac{\left\|g^x_{i,k}\right\|^2}{\bar{v}^{\alpha}_{k+1}}\right]
$$

$$
\leqslant \mathbb{E}\left[\frac{1}{n^2}\sum_{i=1}^{n}\frac{\left(\bar{v}^{\alpha}_{k+1}-v^{\alpha}_{i,k+1}\right)^2}{v^{2\alpha}_{i,k+1}}\left|\bar{v}^{\alpha}_{k+1}-v^{\alpha}_{i,k+1}\right|\frac{\left\|g^x_{i,k}\right\|^2}{\bar{v}^{\alpha}_{k+1}}\right] \tag{67}
$$

$$
+\mathbb{E}\left[\frac{1}{n^2}\sum_{i=1}^{n}\frac{\left|\bar{v}^{\alpha}_{k+1}-v^{\alpha}_{i,k+1}\right|}{v^{\alpha}_{i,k+1}}\left|\bar{v}^{\alpha}_{k+1}-v^{\alpha}_{i,k+1}\right|\frac{\left\|g^x_{i,k}\right\|^2}{\bar{v}^{\alpha}_{k+1}}\right]
$$

$$
\leqslant (1+\zeta_v)\zeta_v\mathbb{E}\left[\frac{1}{n}\sum_{i=1}^{n}\left|\bar{v}^{\alpha}_{k+1}-v^{\alpha}_{i,k+1}\right|\frac{1}{n}\sum_{i=1}^{n}\frac{\left\|g^x_{i,k}\right\|^2}{\bar{v}^{\alpha}_{k+1}}\right].
$$

By Lemma 4, we get

$$
\frac{1}{K}\sum_{k=0}^{K-1}\mathbb{E}\left[\left\|\frac{\left(\tilde{\boldsymbol{v}}^{-\alpha}_{k+1}\right)^T}{n\bar{v}^{-\alpha}_{k+1}}\nabla_x F\left(\mathbf{x}_k,\mathbf{y}_k;\xi^x_k\right)\right\|^2\right]
$$

$$
\leqslant (1+\zeta_v)\zeta_v\mathbb{E}\left[\frac{1}{n}\sum_{i=1}^{n}\left|\bar{v}^{\alpha}_{k+1}-v^{\alpha}_{i,k+1}\right|\frac{1}{K}\sum_{k=0}^{K-1}\frac{\frac{1}{n}\sum_{i=1}^{n}\left\|g^x_{i,k}\right\|^2}{\bar{v}^{\alpha}_{k+1}}\right] \tag{68}
$$

$$
\leqslant (1+\zeta_v)\zeta_v\mathbb{E}\left[\frac{1}{n}\sum_{i=1}^{n}\left|\bar{v}^{\alpha}_{k+1}-v^{\alpha}_{i,k+1}\right|\right]\frac{C^{2-2\alpha}}{(1-\alpha)K^{\alpha}}
$$

$$
\leqslant (1+\zeta_v)\zeta_v\sqrt{\frac{1}{n}\mathbb{E}\left[\left\|\boldsymbol{v}_{k+1}-\mathbf{1}\bar{v}_{k+1}\right\|^{2\alpha}\right]}\frac{C^{2-2\alpha}}{(1-\alpha)K^{\alpha}}.
$$

Next, for the term of inconsistency of the stepsize $\|\boldsymbol{v}_k-\mathbf{1}\bar{v}_k\|^2$, we consider two cases due to the max operator we used. At iteration $k$, for the case $\mathbf{m}^x_k \geqslant \mathbf{m}^y_k$ with $\|\mathbf{m}^x_0-\mathbf{1}\bar{m}^x_0\|^2=0$, we have

$$
\mathbb{E}\left[\|\boldsymbol{v}_{k+1}-\mathbf{1}\bar{v}_{k+1}\|^2\right] = \mathbb{E}\left[\|\mathbf{m}^x_{k+1}-\mathbf{1}\bar{m}^x_{k+1}\|^2\right]
$$

$$
= \mathbb{E}\left[\|(W-\mathbf{J})(\mathbf{m}^x_k-\mathbf{1}\bar{m}^x_k)+\eta_k(W-\mathbf{J})\boldsymbol{h}^x_k\|^2\right]
$$

$$
\leqslant \frac{1+\rho_W}{2}\mathbb{E}\left[\|\mathbf{m}^x_k-\mathbf{1}\bar{m}^x_k\|^2\right]+\frac{(1+\rho_W)\rho_W}{1-\rho_W}\mathbb{E}\left[\|\boldsymbol{h}^x_k\|^2\right] \tag{69}
$$

$$
\leqslant \left(\frac{1+\rho_W}{2}\right)^k\mathbb{E}\left[\|\mathbf{m}^x_0-\mathbf{1}\bar{m}^x_0\|^2\right]+\frac{nC^2(1+\rho_W)\rho_W}{1-\rho_W}\sum_{t=0}^{k}\left(\frac{1+\rho_W}{2}\right)^{k-t}
$$

$$
\leqslant \frac{2nC^2(1+\rho_W)\rho_W}{(1-\rho_W)^2}.
$$

For the case $\mathbf{m}^x_k < \mathbf{m}^y_k$, with $\|\mathbf{m}^y_0-\mathbf{1}\bar{m}^y_0\|^2=0$,

$$
\mathbb{E}\left[\|\boldsymbol{v}_{k+1}-\mathbf{1}\bar{v}_{k+1}\|^2\right] = \mathbb{E}\left[\|\mathbf{m}^y_{k+1}-\mathbf{1}\bar{m}^y_{k+1}\|^2\right] \leqslant \frac{2nC^2(1+\rho_W)\rho_W}{(1-\rho_W)^2}, \tag{70}
$$

Combining these two cases, and using Lemma 4 and the fact $\|v_k^\alpha - \mathbf{1}\bar{v}_k^\alpha\|^2 \leqslant \|v_k - \mathbf{1}\bar{v}_k\|^{2\alpha}$ for $\alpha \in (0,1)$, we obtain the result for primal decision variable. Following the same proof, we can also derive the result for dual decision variable. We thus complete the proof. $\qquad\square$

We further give the following lemma to show that the inconsistency of stepsize remains uniformly bounded for the vanilla D-TiAda algorithm as given in (2).

**Lemma 10** (Inconsistency for D-TiAda). *Suppose Assumption 1-4 hold. Then, for D-TiAda, we have*

$$
\frac{1}{K}\sum_{k=0}^{K-1}\mathbb{E}\left[\left\|\frac{\left(\tilde{\boldsymbol{v}}_{k+1}^{-\alpha}\right)^T}{n\bar{v}_{k+1}^{-\alpha}}\nabla_x F\left(\mathbf{x}_k,\mathbf{y}_k;\xi_k^x\right)\right\|^2\right] \leqslant \zeta_v^2 C^2,
$$

$$
\frac{1}{K}\sum_{k=0}^{K-1}\mathbb{E}\left[\left\|\frac{\left(\tilde{\boldsymbol{u}}_{k+1}^{-\beta}\right)^T}{n\bar{u}_{k+1}^{-\beta}}\nabla_y F\left(\mathbf{x}_k,\mathbf{y}_k;\xi_k^y\right)\right\|^2\right] \leqslant \zeta_u^2 C^2.
\tag{71}
$$

*Proof.* By the definition of inconsistency of stepsizes in (8) and Assumption 3 on bounded gradient, we immediately get the result. $\qquad\square$

## B.3 Proof of Theorem 1

*Proof of Theorem 1.* Consider a complete graph with 3 nodes where the functions corresponding to the nodes are as follows:

$$
f_1(x,y) = -\frac{1}{2}y^2 + xy - \frac{1}{2}x^2,
$$

$$
f_2(x,y) = f_3(x,y) = -\frac{1}{2}y^2 - (1 + \frac{1}{a} + \frac{1}{b})xy - \frac{1}{2}x^2,
$$

where $a = 2^{\frac{-1}{2\alpha-1}}$ and $b = 2^{\frac{-1}{2\beta-1}}$.

Notice that the only stationary point of $f(x,y) = (f_1(x,y) + f_2(x,y) + f_3(x,y))/3$ is $(0,0)$. We denote $g_{i,k}^x = \nabla_x f_i(x_k,y_k)$ and $g_{i,k}^y = \nabla_y f_i(x_k,y_k)$.

Now we consider points initialized in line

$$
y = -\frac{1+a}{a+\frac{a}{b}}x,
\tag{72}
$$

where we have

$$
g_{1,0}^x = y_0 - x_0 = -\frac{2ab+a+b}{ab+a}x_0
$$

$$
g_{2,0}^x = g_{3,0}^x = -\left(1 + \frac{1}{b} + \frac{1}{a}\right)y_0 - x_0 = \frac{2ab+a+b}{a^2(b+1)}x_0
$$

$$
g_{1,0}^y = x_0 - y_0 = \frac{2ab+a+b}{ab+a}x_0
$$

$$
g_{2,0}^y = g_{2,0}^y = -\frac{2ab+a+b}{ab(b+1)}x_0.
$$

Note that by our assumptions of the range of $\alpha$ and $\beta$, we have $a < b$. Thus, we have

$$
|g_{1,0}^x| = |g_{1,0}^y| \quad \text{and} \quad |g_{2,0}^x| > |g_{2,0}^y|,
$$

which means $g_{2,0}^x$ would be chosen in the maximum operator in the denominator of TiAda stepsize for $x$. Therefore, after one step, we have

$$x_1 = x_0 - \eta^x \underbrace{\left( \frac{g_{1,0}^x}{\left(|g_{1,0}^x|^2\right)^\alpha} + \frac{g_{2,0}^x}{\left(|g_{2,0}^x|^2\right)^\alpha} + \frac{g_{3,0}^x}{\left(|g_{3,0}^x|^2\right)^\alpha} \right)}_{=0}$$

$$y_1 = y_0 - \eta^y \underbrace{\left( \frac{g_{1,0}^y}{\left(|g_{1,0}^y|^2\right)^\beta} + \frac{g_{2,0}^y}{\left(|g_{2,0}^y|^2\right)^\beta} + \frac{g_{3,0}^y}{\left(|g_{3,0}^y|^2\right)^\beta} \right)}_{=0}.$$

Next, we will use induction to show that $x$ and $y$ will stay in $x_0$ and $y_0$ for any iteration. Assuming for all iterations $k$ in $1, \ldots, t$, $x_k = x_0$ and $y_k = y_0$, then we have in next step

$$x_{t+1} = x_t - \eta^x \left( \frac{g_{1,0}^x}{\left(t \cdot |g_{1,0}^x|^2\right)^\alpha} + \frac{g_{2,0}^x}{\left(t \cdot |g_{2,0}^x|^2\right)^\alpha} + \frac{g_{3,0}^x}{\left(t \cdot |g_{3,0}^x|^2\right)^\alpha} \right).$$

Note that $g_{1,0}^x = -a \cdot g_{2,0}^x$. Then, we get

$$x_{t+1} = x_t - \eta^x \left( \frac{-p \cdot g_{2,0}^x}{t^\alpha \cdot a^{2\alpha} \cdot |g_{2,0}^x|^{2\alpha}} + \frac{2 g_{2,0}^x}{t^\alpha \cdot |g_{2,0}^x|^{2\alpha}} \right)$$

$$= x_t - \frac{g_{2,0}^x}{t^\alpha \cdot |g_{2,0}^x|^{2\alpha}} \underbrace{\left( 2 - a^{1-2\alpha} \right)}_{=0 \text{ (by definition of } a)}$$

$$= x_t.$$

Similarly, we can show that $y_{t+1} = y_t$. Therefore all iterates will stay at $(x_0, y_0)$ if initialized at line $y = -\frac{ab+b}{ab+a}x$, which implies that the initial gradient norm can be arbitrarily large by picking $x_0$ to be large. $\qquad\square$

## B.4   Proof of Theorem 2 and Corollary 1

*Proof of Theorem 2.*   Combining the results obtained in Lemma 6, 7 and 8, we get

$$\sum_{k=0}^{K-1} \mathbb{E}\left[ f\left(\bar{x}_k, y^*\left(\bar{x}_k\right)\right) - f\left(\bar{x}_k, \bar{y}_k\right) \right]$$

$$= \sum_{k=0}^{k_0-1} \mathbb{E}\left[ f\left(\bar{x}_k, y^*\left(\bar{x}_k\right)\right) - f\left(\bar{x}_k, \bar{y}_k\right) \right] + \sum_{k=k_0}^{K-1} \mathbb{E}\left[ f\left(\bar{x}_k, y^*\left(\bar{x}_k\right)\right) - f\left(\bar{x}_k, \bar{y}_k\right) \right]$$

$$\leqslant \frac{1}{2\gamma_y \bar{u}_1^{-\beta} n} \mathbb{E}\left[ \|\mathbf{y}_0 - \mathbf{1}y^*\left(\bar{x}_0\right)\|^2 \right] + \frac{2\left(4\beta C^2\right)^{2+\frac{1}{1-\beta}}}{\mu^{3+\frac{1}{1-\beta}} \gamma_y^{2+\frac{1}{1-\beta}} \bar{u}_1^{2-2\beta}}$$

$$+ \frac{2\gamma_x^2 \kappa^2 \left(1 + \zeta_v^2\right) G^{2\beta}}{n\mu\gamma_y^2} \sum_{k=0}^{k_0-1} \mathbb{E}\left[ \bar{v}_{k+1}^{-2\alpha} \|\nabla_x F\left(\mathbf{x}_k, \mathbf{y}_k; \xi_k^x\right)\|^2 \right]$$

$$+ \frac{\gamma_y \left(1 + \zeta_u^2\right)}{n} \sum_{k=0}^{K-1} \mathbb{E}\left[ \bar{u}_{k+1}^{-\beta} \|\nabla_y F\left(\mathbf{x}_k, \mathbf{y}_k; \xi_k\right)\|^2 \right] + C \sum_{k=0}^{K-1} \mathbb{E}\left[ \sqrt{\frac{1}{n} \|\mathbf{y}_k - \mathbf{1}\bar{y}_k\|^2} \right] \qquad (73)$$

$$+ \frac{4}{\mu} \sum_{k=0}^{K-1} \mathbb{E}\left[ \left\| \frac{\tilde{\boldsymbol{u}}_{k+1}^{-\beta}}{n\bar{u}_{k+1}^{-\beta}} \nabla_y F\left(\mathbf{x}_k, \mathbf{y}_k; \xi_k^y\right) \right\|^2 \right] + \frac{8\gamma_x^2 \kappa^2 \left(1 + \zeta_v^2\right)}{\mu\gamma_y^2 G^{2\alpha-2\beta}} \sum_{k=k_0}^{K-1} \|\nabla_x f\left(\bar{x}_k, \bar{y}_k\right)\|^2$$

$$+ \left( \frac{8\gamma_x^2 \kappa^2 L^2 \left(1 + \zeta_v^2\right)}{n\mu\gamma_y^2 G^{2\alpha-2\beta}} + \frac{4\kappa L}{n} \right) \sum_{k=0}^{K-1} \mathbb{E}\left[ \Delta_k \right] + \frac{4\gamma_x \kappa \left(1 + \zeta_v\right) C^2}{\mu\gamma_y \bar{v}_1^\alpha} \mathbb{E}\left[ \bar{u}_K^\beta \right]$$

$$+ \frac{\gamma_x^2 \left(1 + \zeta_v^2\right)}{\gamma_y \bar{v}_1^{\alpha-\beta}} \left( \kappa^2 + \frac{2\gamma_x^2 \left(1 + \zeta_v^2\right) C^2 \hat{L}^2}{\mu\gamma_y \bar{v}_1^{2\alpha-\beta}} \right) \sum_{k=k_0}^{K-1} \mathbb{E}\left[ \frac{\bar{v}_{k+1}^{-\alpha}}{n} \|\nabla_x F\left(\mathbf{x}_k, \mathbf{y}_k; \xi_k^x\right)\|^2 \right].$$

Letting the separation point between the two phases discussed in Lemma 6 and 7 satisfy

$$G = \left( \frac{16 \left(1 + \zeta_v^2\right) \gamma_x^2 \kappa^4}{\gamma_y^2} \right)^{\frac{1}{2\alpha - 2\beta}}, \tag{74}$$

then, plugging above inequality into (18), with the help of Lemma 4-8 and Lemma 9, we get

$$
\begin{aligned}
\frac{1}{K} &\sum_{k=0}^{K-1} \mathbb{E}\left[ \left\| \nabla \Phi \left( \bar{x}_k \right) \right\|^2 \right] \\
\leqslant{}& E_0 + E_G + E_W + \frac{8 C^{2\alpha} \left( \Phi^{\max} - \Phi^* \right)}{\gamma_x K^{1-\alpha}} \\
&+ \frac{32 \gamma_x \kappa^3 \left(1 + \zeta_v\right) C^{2+2\beta}}{\gamma_y \bar{v}_1^\alpha K^{1-\beta}} + \frac{8\kappa L \gamma_y \left(1 + \zeta_u^2\right) C^{2-2\beta}}{(1-\beta) K^\beta} \\
&+ \left( \gamma_x L_\Phi + \frac{\kappa^3 L \gamma_x^2}{\gamma_y \bar{v}_1^{\alpha-\beta}} + \frac{2 \gamma_x^4 \kappa^2 \left(1 + \zeta_v^2\right) C^2 \hat{L}^2}{\gamma_y^2 \bar{v}_1^{3\alpha-2\beta}} \right) \frac{8 \left(1 + \zeta_v^2\right) C^{2-2\alpha}}{(1-\alpha) K^\alpha} \\
&+ \sqrt{ \frac{1}{n^{1-\alpha}} \left( \frac{4 \rho_W}{(1-\rho_W)^2} \right)^\alpha \frac{16 \left(1 + \zeta_v\right) \zeta_v C^{2-\alpha}}{(1-\alpha) K^\alpha} } \\
&+ \sqrt{ \frac{1}{n^{1-\beta}} \left( \frac{4 \rho_W}{(1-\rho_W)^2} \right)^\beta \frac{32 \kappa^2 \left(1 + \zeta_u\right) \zeta_u C^{2-\beta}}{(1-\beta) K^\beta} } \\
&+ 8 \kappa L C \sqrt{ \frac{8 \rho_W \gamma_y^2 \left(1 + \zeta_u^2\right)}{(1-\rho_W)^2} \left( \frac{C^{2-4\beta}}{(1-2\beta) K^{2\beta}} \mathbb{I}_{\beta<1/2} + \frac{1 + \log u_K - \log v_1}{K \bar{u}_1^{2\beta-1}} \mathbb{I}_{\beta \geqslant 1/2} \right) },
\end{aligned}
\tag{75}
$$

where $\hat{L} = \kappa \left(1 + \kappa\right)^2$, $L_\Phi = L \left(1 + \kappa\right)$, and

$$E_0 := \frac{4\kappa L}{K \gamma_y \bar{u}_1^{-\beta} n} \mathbb{E}\left[ \left\| \mathbf{y}_0 - \mathbf{1} y^* \left(\bar{x}_0\right) \right\|^2 \right] + \frac{16 \kappa^2 \left(4 \beta C^2\right)^{2+\frac{1}{1-\beta}}}{K \mu^{2+\frac{1}{1-\beta}} \gamma_y^{2+\frac{1}{1-\beta}} \bar{u}_1^{2-2\beta}},$$

$$E_G := \frac{16 \gamma_x^2 \kappa^4 \left(1 + \zeta_v^2\right) G^{2\beta}}{\gamma_y^2} \left( \frac{C^{2-4\alpha}}{(1-2\alpha) K^{2\alpha}} \mathbb{I}_{\alpha<1/2} + \frac{1 + \log v_K - \log v_1}{K \bar{v}_1^{2\alpha-1}} \mathbb{I}_{\alpha \geqslant 1/2} \right),$$

$$
\begin{aligned}
E_W :={}& \frac{32 \left(8 \kappa L + 3 L^2\right) \rho_W \gamma_x^2 \left(1 + \zeta_v^2\right)}{(1-\rho_W)^2} \left( \frac{C^{2-4\alpha}}{(1-2\alpha) K^{2\alpha}} \mathbb{I}_{\alpha<1/2} + \frac{1 + \log v_K - \log v_1}{K \bar{v}_1^{2\alpha-1}} \mathbb{I}_{\alpha \geqslant 1/2} \right) \\
&+ \frac{32 \left(8 \kappa L + 3 L^2\right) \rho_W \gamma_y^2 \left(1 + \zeta_u^2\right)}{(1-\rho_W)^2} \left( \frac{C^{2-4\beta}}{(1-2\beta) K^{2\beta}} \mathbb{I}_{\beta<1/2} + \frac{1 + \log u_K - \log v_1}{K \bar{u}_1^{2\beta-1}} \mathbb{I}_{\beta \geqslant 1/2} \right).
\end{aligned}
$$

Letting the total iteration $K$ satisfy the conditions given in (12) such that the terms $E_0$, $E_G$ and $E_W$ are dominated by the others, we thus complete the proof. $\qquad \square$

*Proof of Corollary 1.* With the help of Lemma 10, we can directly adapt the proof of Theorem 2 to get the result in (14). $\qquad \square$

## B.5 Extend the proof to coordinate-wise stepsize

In this subsection, we show how to extend our convergence analysis of D-AdaST to the coordinate-wise adaptive stepsize (Zhou et al., 2018) variant. We first present this variant in Algorithm 2, which can be rewritten in a compact form with the Hadamard product denoted by $\odot$.

**Algorithm 2 D-AdaST with coordinate-wise adaptive stepsize**

**Initialization:** $x_{i,0} \in \mathbb{R}^p$, $y_{i,0} \in \mathcal{Y}$, buffers $m_{i,0}^x, m_{i,0}^y > 0$, stepsizes $\gamma_x, \gamma_y > 0$ and $0 < \beta < \alpha < 1$.

1: **for** iteration $k = 0, 1, \cdots$, each node $i \in [n]$, **do**

2:      Sample i.i.d $\xi_{i,k}^x$ and $\xi_{i,k}^y$, compute:

$$g_{i,k}^x = \nabla_x f_i \left( x_{i,k}, y_{i,k}; \xi_{i,k}^x \right), \ g_{i,k}^y = \nabla_y f_i \left( x_{i,k}, y_{i,k}; \xi_{i,k}^y \right).$$

3:      Accumulate the gradient with Hadamard product:

$$m_{i,k+1}^x = m_{i,k}^x + g_{i,k}^x \odot g_{i,k}^x, \ m_{i,k+1}^y = m_{i,k}^y + g_{i,k}^y \odot g_{i,k}^y$$

4:      Compute the ratio:

$$\psi_{i,k+1} = \left\| m_{i,k+1}^x \right\|^{2\alpha} / \max \left\{ \left\| m_{i,k+1}^x \right\|^{2\alpha}, \left\| m_{i,k+1}^y \right\|^{2\alpha} \right\} \leqslant 1.$$

5:      Update primal and dual variables locally:

$$x_{i,k+1} = x_{i,k} - \gamma_x \psi_{i,k+1} \left( m_{i,k+1}^x \right)^{-\alpha} \odot g_{i,k}^x,$$

$$y_{i,k+1} = y_{i,k} + \gamma_y \left( m_{i,k+1}^y \right)^{-\beta} \odot g_{i,k}^y.$$

6:      Communicate parameters with neighbors:

$$\left\{ m_{i,k+1}^x, m_{i,k+1}^y, x_{i,k+1}, y_{i,k+1} \right\} \leftarrow \sum_{j \in \mathcal{N}_i} W_{i,j} \left\{ m_{j,k+1}^x, m_{j,k+1}^y, x_{j,k+1}, y_{j,k+1} \right\}.$$

7:      Projection of dual variable on to set $\mathcal{Y}$: $y_{i,k+1} \leftarrow \mathcal{P}_{\mathcal{Y}} \left( y_{i,k+1} \right)$.

8: **end for**

$$\mathbf{m}_{k+1}^x = W \left( \mathbf{m}_k^x + \mathbf{h}_k^x \right), \tag{76a}$$

$$\mathbf{m}_{k+1}^y = W \left( \mathbf{m}_k^y + \mathbf{h}_k^y \right), \tag{76b}$$

$$\mathbf{x}_{k+1} = W \left( \mathbf{x}_k - \gamma_x V_{k+1}^{-\alpha} \odot \nabla_x F \left( \mathbf{x}_k, \mathbf{y}_k; \xi_k^x \right) \right), \tag{76c}$$

$$\mathbf{y}_{k+1} = \mathcal{P}_{\mathcal{Y}} \left( W \left( \mathbf{y}_k + \gamma_y U_{k+1}^{-\beta} \odot \nabla_y F \left( \mathbf{x}_k, \mathbf{y}_k; \xi_k^y \right) \right) \right), \tag{76d}$$

where

$$\boldsymbol{h}_k^x = \left[ \cdots, g_{i,k}^x \odot g_{i,k}^x, \cdots \right]^T \in \mathbb{R}^{n \times p}, \ \boldsymbol{h}_k^y = \left[ \cdots, g_{i,k}^y \odot g_{i,k}^y, \cdots \right]^T \in \mathbb{R}^{n \times d},$$

and the matrices $U_k^\alpha$ and $V_k^\beta$ are redefined as follows:

$$V_k^{-\alpha} = \left[ \cdots, v_{i,k}^{-\alpha}, \cdots \right]^T, \ [v_{i,k}]_j = \max \left\{ \left[ m_{i,k}^x \right]_j, \left[ m_{i,k}^y \right]_j \right\}, \ j \in [p],$$

$$U_k^{-\beta} = \left[ \cdots, u_{i,k}^{-\beta}, \cdots \right]^T, \ [u_{i,k}]_j = \left[ m_{i,k}^y \right]_j, \ j \in [d], \tag{77}$$

where $[\cdot]_j$ denotes the $j$-th element of a vector.

Recalling the definitions of inconsistency of stepsize in (8), we give the following notations:

$$\tilde{V}_k = V_k - \bar{v}_k \mathbf{1} \mathbf{1}_p^T, \ \bar{v}_k = \frac{1}{np} \sum_{i=1}^n \sum_j^p V_{ij}, \ \bar{v}_{i,k} = \frac{1}{p} \sum_j^p V_{ij}, \ \bar{v}_{j,k} = \frac{1}{n} \sum_{i=1}^n V_{ij},$$

$$\tilde{U}_k = U_k - \bar{u}_k \mathbf{1} \mathbf{1}_p^T, \ \bar{u}_k = \frac{1}{nd} \sum_{i=1}^n \sum_j^d U_{ij}, \ \bar{u}_{i,k} = \frac{1}{d} \sum_j^d U_{ij}, \ \bar{u}_{j,k} = \frac{1}{n} \sum_{i=1}^n U_{ij}, \tag{78}$$

and

$$\zeta_V^2 = \sup_{k \geqslant 0}\left\{\frac{\left\|V_k^{-\alpha} - \bar{v}_k^{-\alpha}\mathbf{1}\mathbf{1}_p^T\right\|^2}{np\left(\bar{v}_k^{-\alpha}\right)^2}\right\}, \quad \hat{\zeta}_v^2 = \sup_{k \geqslant 0}\left\{\frac{\left\|V_k^{-\alpha} - (V_k\mathbf{J}_p)^{-\alpha}\right\|^2}{np\left(\bar{v}_k^{-\alpha}\right)^2}\right\},$$

$$\zeta_U^2 = \sup_{k \geqslant 0}\left\{\frac{\left\|U_k^{-\beta} - \bar{u}_k^{-\beta}\mathbf{1}\mathbf{1}_d^T\right\|^2}{nd\left(\bar{u}_k^{-\beta}\right)^2}\right\}, \quad \hat{\zeta}_u^2 = \sup_{k \geqslant 0}\left\{\frac{\left\|U_k^{-\beta} - (U_k\mathbf{J}_d)^{-\beta}\right\|^2}{nd\left(\bar{u}_k^{-\beta}\right)^2}\right\}.$$

Building upon the established definitions of coordinate-wise stepsize inconsistency, the subsequent lemma is presented to show the non-convergence of the inconsistency term compared to Lemma 9.

**Lemma 11** (Inconsistency, coordinate-wise)**.** *Suppose Assumption 1-4 hold. For the proposed D-AdaST algorithm, we have*

$$\frac{1}{K}\sum_{k=0}^{K-1}\mathbb{E}\left[\left\|\frac{\mathbf{1}^T}{n\bar{v}_{k+1}^{-\alpha}}\tilde{V}_{k+1}^{-\alpha}\odot\nabla_x F\left(\mathbf{x}_k,\mathbf{y}_k;\xi_k^x\right)\right\|^2\right]$$

$$\leqslant 2\left(1+\zeta_v\right)\zeta_v\sqrt{\frac{1}{n^{1-\alpha}}\left(\frac{4C^2\rho_W}{(1-\rho_W)^2}\right)^\alpha\frac{C^{2-2\alpha}}{(1-\alpha)K^\alpha}} + 2np\hat{\zeta}_v^2 C^2 \tag{79}$$

*and*

$$\frac{1}{K}\sum_{k=0}^{K-1}\mathbb{E}\left[\left\|\frac{\mathbf{1}^T}{n\bar{u}_{k+1}^{-\beta}}\tilde{U}_{k+1}^{-\beta}\odot\nabla_y F\left(\mathbf{x}_k,\mathbf{y}_k;\xi_k^y\right)\right\|^2\right]$$

$$\leqslant 2\left(1+\zeta_u\right)\zeta_u\sqrt{\frac{1}{n^{1-\beta}}\left(\frac{4C^2\rho_W}{(1-\rho_W)^2}\right)^\beta\frac{C^{2-2\beta}}{(1-\beta)K^\beta}} + 2nd\hat{\zeta}_u^2 C^2. \tag{80}$$

*In contrast, for D-TiAda, we have*

$$\frac{1}{K}\sum_{k=0}^{K-1}\mathbb{E}\left[\left\|\frac{\mathbf{1}^T}{n\bar{v}_{k+1}^{-\alpha}}\tilde{V}_{k+1}^{-\alpha}\odot\nabla_x F\left(\mathbf{x}_k,\mathbf{y}_k;\xi_k^x\right)\right\|^2\right] \leqslant p\zeta_V^2 C^2,$$

$$\frac{1}{K}\sum_{k=0}^{K-1}\mathbb{E}\left[\left\|\frac{\mathbf{1}^T}{n\bar{u}_{k+1}^{-\beta}}\tilde{U}_{k+1}^{-\beta}\odot\nabla_y F\left(\mathbf{x}_k,\mathbf{y}_k;\xi_k^y\right)\right\|^2\right] \leqslant d\zeta_U^2 C^2. \tag{81}$$

*Proof.* For the coordinate-wise adaptive stepsize, with the definitions of Frobenius norm and Hadamard product, we have

$$\mathbb{E}\left[\left\|\frac{\mathbf{1}^T}{n\bar{v}_{k+1}^{-\alpha}}\tilde{V}_{k+1}^{-\alpha}\odot\nabla_x F\left(\mathbf{x}_k,\mathbf{y}_k;\xi_k^x\right)\right\|^2\right]$$

$$= \mathbb{E}\left[\left\|\frac{\mathbf{1}^T}{n\bar{v}_{k+1}^{-\alpha}}\left(V_{k+1}^{-\alpha} - (V_{k+1}\mathbf{J})^{-\alpha} + (V_{k+1}\mathbf{J})^{-\alpha} - \bar{v}_{k+1}^{-\alpha}\mathbf{1}\mathbf{1}_p^T\right)\odot\nabla_x F\left(\mathbf{x}_k,\mathbf{y}_k;\xi_k^x\right)\right\|^2\right]$$

$$\leqslant 2\mathbb{E}\left[\left\|\frac{\mathbf{1}^T}{n\bar{v}_{k+1}^{-\alpha}}\left((V_{k+1}\mathbf{J})^{-\alpha} - \bar{v}_{k+1}^{-\alpha}\mathbf{1}\mathbf{1}_p^T\right)\odot\nabla_x F\left(\mathbf{x}_k,\mathbf{y}_k;\xi_k^x\right)\right\|^2\right] \tag{82}$$

$$+ 2\mathbb{E}\left[\left\|\frac{\mathbf{1}^T}{n\bar{v}_{k+1}^{-\alpha}}\left(V_{k+1}^{-\alpha} - (V_{k+1}\mathbf{J})^{-\alpha}\right)\odot\nabla_x F\left(\mathbf{x}_k,\mathbf{y}_k;\xi_k^x\right)\right\|^2\right].$$

For the first term on the RHS, according to the definitions given in (78), we have

$$
\mathbb{E}\left[\left\|\frac{\mathbf{1}^T}{n\bar{v}_{k+1}^{-\alpha}}\left((V_{k+1}\mathbf{J})^{-\alpha}-\bar{v}_{k+1}^{-\alpha}\mathbf{1}\mathbf{1}_p^T\right)\odot\nabla_x F\left(\mathbf{x}_k,\mathbf{y}_k;\xi_k^x\right)\right\|^2\right]
$$
$$
\leqslant\mathbb{E}\left[\frac{1}{n^2\bar{v}_{k+1}^{-2\alpha}}\sum_{i=1}^n\left(\bar{v}_{i,k+1}^\alpha-\bar{v}_{k+1}^\alpha\right)^2\left\|\nabla_x f_i\left(x_{i,k},y_{i,k};\xi_{i,k}^x\right)\right\|^2\right].
$$

(83)

Then, for the second part, we have

$$
\mathbb{E}\left[\left\|\frac{\mathbf{1}^T}{n\bar{v}_{k+1}^{-\alpha}}\left(V_{k+1}^{-\alpha}-(V_{k+1}\mathbf{J})^{-\alpha}\right)\odot\nabla_x F\left(\mathbf{x}_k,\mathbf{y}_k;\xi_k^x\right)\right\|^2\right]
$$
$$
\leqslant\frac{1}{n}\mathbb{E}\left[\left\|\frac{V_{k+1}^{-\alpha}-(V_{k+1}\mathbf{J})^{-\alpha}}{\bar{v}_{k+1}^{-\alpha}}\right\|^2\|\nabla_x F\left(\mathbf{x}_k,\mathbf{y}_k;\xi_k^x\right)\|^2\right]
$$
$$
\leqslant p\hat{\zeta}_v^2\mathbb{E}\left[\|\nabla_x F\left(\mathbf{x}_k,\mathbf{y}_k;\xi_k^x\right)\|^2\right].
$$

(84)

where the term $\hat{\zeta}_v^2$ is not guaranteed to be convergent because the stepsizes between the different dimensions of each node are not consistent. Then, similar to the proof of Lemma 9, we can obtain the result presented in (79).

Next, noticing that for D-TiAda,

$$
\mathbb{E}\left[\left\|\frac{\mathbf{1}^T}{n\bar{v}_{k+1}^{-\alpha}}\tilde{V}_{k+1}^{-\alpha}\odot\nabla_x F\left(\mathbf{x}_k,\mathbf{y}_k;\xi_k^x\right)\right\|^2\right]\leqslant\frac{1}{n}\mathbb{E}\left[\left\|\frac{\tilde{V}_{k+1}^{-\alpha}}{\bar{v}_{k+1}^{-\alpha}}\right\|^2\|\nabla_x F\left(\mathbf{x}_k,\mathbf{y}_k;\xi_k^x\right)\|^2\right]\leqslant p\zeta_V^2 C^2,
$$

(85)

and using Lemma 9, we complete the proof. □

**Theorem 3.** *Suppose Assumption 1-4 hold. Let $0<\beta<\alpha<1$ and the total iteration satisfy*

$$
K=\Omega\left(\max\left\{\left(\frac{\gamma_x^2\kappa^4}{\gamma_y^2}\right)^{\frac{1}{\alpha-\beta}},\quad\left(\frac{1}{(1-\rho_W)^2}\right)^{\max\left\{\frac{1}{\alpha},\frac{1}{\beta}\right\}}\right\}\right).
$$

*to ensure time-scale separation and quasi-independence of network. For D-AdaST with coordinate-wise adaptive stepsize, we have*

$$
\frac{1}{K}\sum_{k=0}^{K-1}\mathbb{E}\left[\|\nabla\Phi\left(\bar{x}_k\right)\|^2\right]
$$
$$
=\tilde{\mathcal{O}}\left(\frac{1}{K^{1-\alpha}}+\frac{1}{(1-\rho_W)^\alpha K^\alpha}+\frac{1}{K^{1-\beta}}+\frac{1}{(1-\rho_W) K^\beta}\right)+\mathcal{O}\left(n\left(p\hat{\zeta}_v^2+\kappa^2 d\hat{\zeta}_u^2\right)C^2\right).
$$

(86)

*Proof.* With the help of Lemma 11 and the obtained result (75) in the proof of Theorem 2, we can derive the convergence results for D-AdaST with coordinate-wise adaptive stepsize. □

**Remark 6.** *In Theorem 3, we show that the coordinate-wise variant of D-AdaST exhibits a steady-state error in its upper bound. This error depends on the number of nodes and the dimension of the problem, which stems from the stepsize inconsistency in each dimension of the local decision variables for each node (c.f., Line 3 of Algorithm 2).*

