# OpenReview forum: "Achieving Near-Optimal Convergence for Distributed Minimax Optimization with Adaptive Stepsizes"
_NeurIPS.cc/2024/Conference — NeurIPS 2024 poster_

### Official Review · Reviewer_KXvx · 2024-07-10

**Soundness:** 3
**Presentation:** 3
**Contribution:** 3
**Rating:** 6
**Confidence:** 3

**Summary:**

The paper presents D-AdaST, a distributed adaptive minimax method designed to address non-convergence issues in nonconvex-strongly-concave (NC-SC) minimax problems caused by inconsistent locally computed adaptive stepsizes in distributed settings. The method incorporates a stepsize tracking mechanism, which ensures consistency across local stepsizes and maintains time-scale separation, achieving near-optimal convergence rates comparable to centralized counterparts.

**Strengths:**

The paper addresses a significant challenge in distributed minimax optimization by introducing a method that achieves near-optimal convergence rates. The theoretical contributions are well-founded, and the experimental validation is comprehensive.

**Weaknesses:**

Please consider my questions for this part.

**Questions:**

1- How does D-AdaST compare to other existing adaptive minimax methods in terms of computational time efficiency?

2-The current analysis focuses on nonconvex-strongly-concave minimax problems. How would the theoretical framework extend to other problem classes, such as nonconvex-nonconcave or convex-concave problems? What modifications or additional assumptions would be necessary?

3-In a distributed setting, communication delays and asynchrony can pose significant challenges. How does the theoretical analysis account for these factors? Are there any bounds or guarantees provided for performance in asynchronous or delayed communication settings?

**Limitations:**

They discussed the limitations of their work in terms of assumptions and main results.

---

> ### Author Rebuttal · Authors · 2024-08-06
>
> Thank you for your valuable comments and suggestions. Please see below for a detailed point-by-point response.
>
> >__Q1:__ How does D-AdaST compare to other existing adaptive minimax methods in terms of computational time efficiency?
>
> __Response:__ We note that, as discussed in the Introduction section, there is limited literature on the distributed adaptive minimax method; instead, the proposed D-AdaST is the first distributed adaptive method that achieves near-optimal convergence without knowledge of problem-dependent parameters for nonconvex minimax problems. The resulting *sample complexity* $\tilde{\mathcal{O}} \left( \epsilon ^{-(4+\delta)} \right)$ with arbitrarily small $\delta>0$ represents the number of stochastic gradient computation needed to find an $\epsilon$-stationary point, which indicates the theoretical computational complexity and is near-optimal compared to the existing lower bound $\varOmega \left( \epsilon ^{- 4} \right)$ for the centralized nonconvex minimax optimization [6]. Experimentally, we compared the convergence performance of D-AdaST with the distributed variants of AdaGrad, TiAda, and NeAda in terms of the number of *gradient calls* (see Figure 3 and 4), which properly represents the computational efficiency of the algorithms when all experiments are run on the same server with the same batch size. We believe that these results demonstrate the superiority of the proposed D-AdaST algorithm regarding computational efficiency.
>
> >__Q2:__ The current analysis focuses on nonconvex-strongly-concave minimax problems. How would the theoretical framework extend to other problem classes, such as nonconvex-nonconcave or convex-concave problems? What modifications or additional assumptions would be necessary?
>
> __Response:__ Given the limited existing work on distributed adaptive minimax methods, we believe that it is an interesting and important direction for future work to study other problem settings, such as nonconvex-nonconcave or convex-concave objective functions. For the convex-concave setting, we expect that the convergence analysis techniques from centralized methods [14] can be used to obtain results in our decentralized setting. Going beyond the convex-concave assumption poses a major challenge in the sense that there is no well-formulated suboptimality measure for general nonconvex-nonconcave problems. Specifically, for a general nonconvex-nonconcave minimax problem, a saddle point may not exist even in a centralized setting [15], and determining its existence is known to be NP-hard. Therefore, making additional assumptions to ensure better properties of the inner problem is necessary,  such as employing the Polyak-Lojasiewicz condition on $y$  to obtain global convergence for the nonconvex-nonconcave  problem [16].
>
> >__Q3:__ In a distributed setting, communication delays and asynchrony can pose significant challenges. How does the theoretical analysis account for these factors? Are there any bounds or guarantees provided for performance in asynchronous or delayed communication settings?
>
> __Response:__ We believe that it is an independently interesting and important problem to consider the communication delays and asynchronous communication protocols in distribution minimax optimization, which requires major adjustments to the communication model. Particularly, we notice that in [17], by constructing an augmented topology graph (c.f., Fig. 2), the delayed and asynchronous system is reduced to a synchronous augmented one with no delays by adding virtual nodes to the graph. They obtained linear and sublinear convergence for strongly convex and non-convex minimization, respectively, under the assumptions of bounded delay and asynchronous model (see Assumption 6 in [17]). This motivates us to potentially extend this technique for modeling communication to our setting in future work.

---

> > ### Comment · Reviewer_KXvx · 2024-08-09
> >
> > Appreciate authors for their response and clarification. I am satisfied with their response and maintain my score.

---

> > > ### Author Response · Authors · 2024-08-10
> > > **Thanks to Reviewer KXvx**
> > >
> > > We appreciate the reviewer's acknowledgement. Thank you for taking the time to carefully review our paper and respond to our clarifications.

---

### Official Review · Reviewer_Nhkx · 2024-07-11

**Soundness:** 3
**Presentation:** 3
**Contribution:** 3
**Rating:** 6
**Confidence:** 3

**Summary:**

This paper proposed a decentralized stochastic first-order method for nonconvex minimax optimization. For nonconvex-strongly-concave setting, the proposed D-AdaST has the convergence rate of $O(\epsilon^{-4+\delta})$ for any small $\delta>0$.

**Strengths:**

see questions

**Weaknesses:**

see questions

**Questions:**

The proposed method does not require the knowledge of the parameters $L$, $\mu$ and total iterations number, which is nice in  empirical.

I have some questions as follows:

1.	This paper states D-AdaST achieves the near-optimal convergence rate, while the definition of optimality is unclear, e.g., problem setting and algorithm class. I’m not sure whether this statement is correct. Specifically, Assumption 2 requires the second-order Lipschitz continuous, while the existing lower for solving nonconvex-strongly-concave minimax problem by first-order methods only considers the first-order Lipschitz continuous. The author should clarify how to fill this gap and explain how to achieve the lower bound in the setting of D-AdaST more clearly.
2.	Assumption 3 introduce the upper bound $C$ for stochastic gradient, which should be included in the final convergence rate.
3.	Can you provide some explanation why the small $\delta$ cannot be avoided in result? What happens if we try to take $\delta\to0$?
4.	The section of introduction describes the problem setting with constraints on both $x$ and $y$, while the algorithm design and convergence analysis looks only consider the constraint on $y$.
5.	Is it possible to increase the batch-size of stochastic gradient to reduce the total communication rounds? Typically, the communication rounds in decentralized optimization can be smaller than the computation rounds, e.g., Chen et al., 2024.

---

> ### Author Rebuttal · Authors · 2024-08-06
>
> Thank you for your valuable comments and suggestions. Please see below for a detailed point-by-point response.
>
> >__Q1:__  This paper states D-AdaST achieves the near-optimal convergence rate, while the definition of optimality is unclear, e.g., problem setting and algorithm class. I’m not sure whether this statement is correct. Specifically, Assumption 2 requires the second-order Lipschitz continuous, while the existing lower for solving nonconvex-strongly-concave minimax problems by first-order methods only considers the first-order Lipschitz continuous. The author should clarify how to fill this gap and explain how to achieve the lower bound in the setting of D-AdaST more clearly.
>
> __Response:__ We will further clarify the definition of optimality of the convergence in the revision to avoid possible confusion. Specifically, we followed the definitions in [6], where they derived a complexity lower bound $\varOmega \left( \epsilon ^{-4} \right)$ for a class of smooth nonconvex-strongly-concave (NC-SC) functions (c.f., Definition 1) and the first-order algorithms that satisfy the zero-respecting condition using only (historical) gradient information (c.f., Definition 4). Under this problem setting and algorithm class, they proved that the dependency on $\epsilon$ is not improvable. Therefore, we claimed that our obtained convergence rate $\tilde{\mathcal{O}} \left( \epsilon ^{-\left( 4+\delta \right)} \right)$ is near-optimal in terms of $\epsilon$ in the sense that the parameter $\delta$ can be arbitrarily small so as to achieve the lower bound. It should be noted that the parameter $\delta$ can not be 0 due to its key role in ensuring time-scale separation (refer to the response to Q3 and Remark 4 for a more detailed discussion). We note that, to our knowledge, there is no existing parameter-agnostic method that achieves the optimal convergence rate for NC-SC problems as considered in this work, even in a centralized setting.
>
> Regarding the assumption on second-order Lipschitz continuous for $y$, we note that it is essential for achieving the (near) optimal convergence rate $\tilde{\mathcal{O}} \left( \epsilon ^{- 4} \right)$ for NC-SC problems [7, 8]. Specifically, together with Assumptions 1 and 2, we can show that $y^*\left( \cdot \right)$ is smooth (c.f., Lemma 2). Nevertheless, without this assumption, [9] only shows a worse complexity of $\tilde{\mathcal{O}} \left( \epsilon ^{- 5} \right)$ without large batch size (c.f., Remark 4.6).
>
> >__Q2:__ Assumption 3 introduce the upper bound $C$ for stochastic gradient, which should be included in the final convergence rate.
>
> __Response:__ We will provide a more detailed convergence result including dependence on $C$ in the revision. The explicit convergence result can be found in (75) in the Appendix, which shows the dependence on $C$ and other constant parameters.
>
> >__Q3:__ Can you provide some explanation why the small $\delta$ cannot be avoided in result? What happens if we try to take $\delta \rightarrow 0$.
>
> __Response:__ We note that, for the NC-SC minimax problem, it is necessary to have a time-scale separation in stepsizes between the minimization and maximization processes to ensure the convergence of gradient-descent-ascent-based algorithms [10]. As shown in Theorem 2, the transient time required to reach this state is related to $\left( \cdot \right) ^{\frac{1}{\alpha -\beta}}$, while we need $\alpha -\beta =0$ to reach the convergence rate of $\tilde{\mathcal{O}} \left( \epsilon ^{-4} \right)$ (c.f., Eq. (13)). Therefore, by setting $\alpha -\beta =\mathcal{O} \left( \delta \right), \delta >0$, we found that the smaller $\delta$, the faster the convergence and the longer the transition time (c.f., Figure 9 in Appendix A.3). Particularly, the transition time becomes infinite when $\delta$ approaches 0, indicating that the algorithm does not converge. Therefore, the parameter $\delta$ plays a key role in achieving adaptive time-scale separation as well as balancing the convergence speed with the transient process. Please refer to Remark 4 for a more detailed discussion.
>
> >__Q4:__ The section of introduction describes the problem setting with constraints on both $x$ and $y$, while the algorithm design and convergence analysis looks only consider the constraint on $y$.
>
> __Response:__ In the introduction, we initially introduced a general minimax problem with $x\in \mathcal{X} \subseteq \mathbb{R} ^p$ and $y\in \mathcal{Y} \subseteq \mathbb{R} ^d$, which subsumes both scenarios of the constrained and unconstrained problem, as $\mathcal{X}$ and $\mathcal{Y}$ can represent the full Euclidean space with appropriate dimensions. For the specific problem (1) considered in this work, we set $\mathcal{X} =\mathbb{R} ^p$ and  $\mathcal{Y} \subset \mathbb{R} ^d$ is a closed and convex set, which is widely considered in centralized/distributed NC-SC minimax problems [10, 11].
>
> >__Q5:__ Is it possible to increase the batch-size of stochastic gradient to reduce the total communication rounds? Typically, the communication rounds in decentralized optimization can be smaller than the computation rounds, e.g., Chen et al., 2024.
>
> __Response:__ We agree with the reviewer that the complexity of communication can be further reduced by incorporating certain techniques that amount to increasing the batch size of the stochastic gradient, such as variance reduction [12] and multiple local updates used in federated learning [13]. As mentioned by the reviewer, Chen et al. [12] used larger batch-size or full gradient evaluations (c.f., step 15 in Algorithm 2), together with the FastMix protocol, to reduce the computation and communication complexity. However, it should be noted that the resulting complexity is obtained under a stronger assumption of the average smoothness (c.f., Assumption 2.2 in [12]). Instead, we obtained a near-optimal rate under a weaker assumption of smoothness (c.f, Assumption 2), which is commonly used in machine learning tasks [6].

---

> > ### Comment · Reviewer_Nhkx · 2024-08-07
> >
> > Thanks for your detailed rebuttal. I am satisfied with most of your response, while there are still some questions should be addressed.
> >
> > My main consideration is the applicability of the lower bound provided by Li et al. [6]. Noticing that their Definition 1 does not include second-order Lipschitz continuity. The lower bound of $\Omega(\epsilon^{-4})$ in their Theorem 2 is based on the function $f^{{\rm nc}-{\rm sc}-{\rm sg}}$. The proof of this theorem only verifies its gradient is Lipschitz continuous. However, the Lipschitz continuity of its Hessian is unclear. Therefore, we should check if the construction of Li et al. [6] holds the second-order Lipschitz continuity shown in Assumption 2. If the author can address this issue, I will increase my overall rating. Otherwise, the statement on the optimality should be avoided in revision (I will still keep borderline accept).
> >
> > **Some minor comments:**
> >
> > For Q4, I recommend to unify the presentation of constrained/unconstrained setting on the variable.
> >
> > For Q5, I think reduce the communication complexity is possible even if there is no average smoothness assumption. At least, this can be address in nonconvex minimization, e.g.,
> >
> > [18] Yucheng Lu and Christopher De Sa. Optimal Complexity in Decentralized Training. ICML 2021.

---

> > > ### Author Response · Authors · 2024-08-09
> > > **Response to Reviewer Nhkx**
> > >
> > > Thank you for your prompt response and valuable comments. We have carefully considered your remaining concerns and have responded in detail as follows:
> > >
> > > >__Q6:__ My main consideration is the applicability of the lower bound provided by Li et al. [6]. Noticing that their Definition 1 does not include second-order Lipschitz continuity. The lower bound of $\varOmega \left( \epsilon ^{-4} \right)$ in their Theorem 2 is based on the function $f^{\text{sc-nc-sg}}$. The proof of this theorem only verifies its gradient is Lipschitz continuous. However, the Lipschitz continuity of its Hessian is unclear. Therefore, we should check if the construction of Li et al. [6] holds the second-order Lipschitz continuity shown in Assumption 2. If the author can address this issue, I will increase my overall rating. Otherwise, the statement on the optimality should be avoided in revision (I will still keep borderline accept).
> > >
> > > __Response__ We have carefully checked the construction of the hard examples for obtaining the complexity lower bound in [6] and verified that they satisfy second-order Lipschitz continuity on $y$ as assumed in Assumption 2. Specifically, we recall the hard instance in the stochastic setting in [6] (c.f., Eqs. (10) and (11)) as follows:
> > > $$
> > > \bar{f}^{\text{nc-sc-sg}}\left( \boldsymbol{x},\boldsymbol{z};\bar{\boldsymbol{y}} \right) =-\Psi \left( 1 \right) \Phi \left( x_1 \right) +\sum_{i=2}^T{\left[ \Psi \left( -z_i \right) \Phi \left( -x_i \right) -\Psi \left( z_i \right) \varPhi \left( x_i \right) \right]}+\sum_{i=1}^{T-1}{\left[ c_1x_{i}^{2}+c_2z_{i+1}^{2} \right]}+\sum_{i=1}^{T-1}{h^{\text{sg}}\left( x_i,z_{i+1};\bar{\boldsymbol{y}}^{\left( i \right)} \right)},
> > > $$
> > >
> > > with
> > >
> > > $$
> > > h^{\text{sg}}\left( x,z;\boldsymbol{y} \right) =\frac{C}{n}\left[ -\frac{1}{2}\boldsymbol{y}^{\text{T}}\left( \frac{1}{n^2}I_n+A \right) \boldsymbol{y}+\boldsymbol{b}_{x,z}^{\text{T}}\boldsymbol{y} \right]
> > > $$
> > >
> > > and $\boldsymbol{b}_{x,z}=x\boldsymbol{e}_1-\frac{1}{2}z\boldsymbol{e}_n$, where $\boldsymbol{x}$ and $\boldsymbol{z}$ are variables to minimize, $\boldsymbol{y}$ is the variable to maximize, $\Psi \left( \cdot \right)$ and $\Phi \left( \cdot \right)$ are component functions, $c_1$, $c_2$ and $C$ are constants, $A$ is a positive semi-definite matrix (refer to Section 4 in [6] for detailed definitions).
> > >
> > > It should be noted that the assumption of second-order Lipschitz continuity of this work is only required to hold for $y$ (c.f., Eq. (11) in Assumption 2), and the terms related to $y$ in the objective function $\bar{f}^{\text{nc-sc-sg}}$ are only in $h^{\text{sg}}$, which is twice differentiable. It is not difficult to verify that $\nabla _{x,z;\boldsymbol{y}}^{2}h^{\mathrm{sg}}=-\left( \frac{1}{n^2}I_n+A \right)$ and $\nabla _{\boldsymbol{yy}}^{2}h^{\mathrm{sg}}=\mathrm{diag}\left\\{ 1,0,\cdots,0,-1/2 \right\\}$ are constant matrices and thus both are Lipschitz continuous. This indicates that $\bar{f}^{\text{nc-sc-sg}}$ is second-order Lipschitz continuous for $y$, satisfying Eq. (11) in Assumption 2. Therefore, we believe that the lower bound given by Li et al. [6] is applicable to the function class considered in our work. Combined with the previous discussion, we will carefully revise our statement about "near-optimal" to avoid any possible confusion.
> > >
> > > >__Q7:__ For Q4, I recommend to unify the presentation of constrained/unconstrained setting on the variable.
> > >
> > > __Response__ Thank you for your suggestion. We will do a proper revision of the presentation of the decision variable domains.
> > >
> > > >__Q8:__ For Q5, I think reduce the communication complexity is possible even if there is no average smoothness assumption. At least, this can be address in nonconvex minimization, e.g.,
> > > > * [18] Yucheng Lu and Christopher De Sa. Optimal Complexity in Decentralized Training. ICML 2021.
> > >
> > > __Response__ We agree with the reviewer that reducing the communication complexity is possible even without relying on the average smoothness assumption. As mentioned by the reviewer, without this assumption, Lu et al. [18] combined local updates with the Factorized Consensus method (c.f., Lemma 1 in [18]) and Accelerated Gossip protocol (c.f., Algorithm 3), yielding reduced communication complexity. However, in order to obtain a computational complexity of $\varOmega \left( \epsilon ^{-3} \right)$ as in [12], this assumption is required in most of the literature based on the variance reduction methods. Otherwise, without this assumption, the sampling complexity obtained by Lu et al. [18] matches another lower bound $\varOmega \left( \epsilon ^{-4} \right)$. Having noted the characteristics and limitations of these methods, we believe they are all worthy of further investigation to improve the communication efficiency of distributed algorithms, with or without this average smoothness assumption.

---

> > > > ### Comment · Reviewer_Nhkx · 2024-08-09
> > > >
> > > > Thanks for your very detailed response. The response to Q7 and Q8 should be involved into the revision.
> > > >
> > > > Since my questions have been well addressed, I will raise my overall rating.

---

> > > > > ### Author Response · Authors · 2024-08-09
> > > > > **Thanks to Reviewer Nhkx**
> > > > >
> > > > > We are grateful for the reviewer's engagement and recognition.  Thanks for spending your time carefully reviewing our paper and responding to our clarifications.
> > > > >
> > > > > We will incorporate the necessary revisions, taking into account all the points we have discussed. We believe that these changes will enhance the quality of our paper.

---

### Official Review · Reviewer_vU1y · 2024-07-13

**Soundness:** 3
**Presentation:** 3
**Contribution:** 3
**Rating:** 6
**Confidence:** 2

**Summary:**

The authors introduced a new distributed adaptive minimax method, D-AdaST, to address the issue of non-convergence in nonconvex-strongly-concave minimax problems caused by inconsistencies in locally computed adaptive stepsizes. D-AdaST employs an efficient adaptive stepsize tracking protocol that ensures time-scale separation and consistency of stepsizes among nodes, thereby eliminating steady-state errors. Extensive experiments on both real-world and synthetic datasets validate their theoretical findings across various scenarios.

**Strengths:**

Originality and Significance:
The paper introduces a novel Distributed Adaptive Minimax Optimization Algorithm with a Stepsize Tracking Protocol. This approach is significant in addressing the non-convergence issues that arise when transitioning adaptive single-machine algorithms to a distributed environment. The authors provide a thorough theoretical analysis, which could facilitate the development of similar methods or modifications to existing single-machine algorithms.

Quality  and Clarity:
The paper is well-structured and clear, with thoroughly explained assumptions and convergence rates, making it easy to understand.

**Weaknesses:**

The experimental section has some deficiencies, as it only includes a GAN experiment on the CIFAR-10 dataset. I believe it would be beneficial to supplement the paper with results from additional datasets or other minimax optimization problems to provide a more comprehensive evaluation.

**Questions:**

Please see weakness.

**Limitations:**

The paper employs general assumptions for distributed minimax optimization problems; therefore, it is unnecessary to discuss the limitations.

---

> ### Author Rebuttal · Authors · 2024-08-06
>
> >__W1:__ The experimental section has some deficiencies, as it only includes a GAN experiment on the CIFAR-10 dataset. I believe it would be beneficial to supplement the paper with results from additional datasets or other minimax optimization problems to provide a more comprehensive evaluation.
>
> __Response:__ Thank you for your comments. We provide additional experiments of training GANs on a more complicated dataset CIFAR-100 to further illustrate the effectiveness of the proposed D-AdaST, as shown in the __attached PDF file in General Response__. We use the entire training set of CIFAR-100 with coarse labels (20 classes) to train GANs over networks, where each node is assigned four distinct classes of labeled samples. Due to time constraints, we ran one experiment with the same settings as in Figure 4(a). The revision may include other complementary experiments on additional datasets. It can be observed that D-AdaST outperforms the others in terms of the inception score. Together with other experimental results in the paper, we believe that we have demonstrated the effectiveness of the proposed D-AdaST method and its potential for further real-world applications.

---

> > ### Comment · Reviewer_vU1y · 2024-08-13
> >
> > Thank you for the additional information regarding the experiment. I will accordingly adjust my score.

---

> > > ### Author Response · Authors · 2024-08-14
> > > **Thanks to Reviewer vU1y**
> > >
> > > Thanks for the thoughtful acknowledgement and consideration of our responses. We sincerely appreciate the reviewer's valuable time in reviewing our paper.

---

### Official Review · Reviewer_rQb7 · 2024-07-13

**Soundness:** 3
**Presentation:** 3
**Contribution:** 2
**Rating:** 6
**Confidence:** 2

**Summary:**

The paper introduces a method for distributed minimax optimization, for the scenario that various "agents" each hold part of a data set locally, and aim to coordinate to find the minimax solution of some criterion. Here, the criterion consists of the average of local cost functions, which are assumed to be smooth but non-convex-strongly-concave. Each agent is assumed to able to compute a stochastic gradient at each iteration, which can be communicated across the neighboring agents, who collectively communicate according to a given, known graph structure.

In the nondistributed setting, known gradient-based methods depend on hyper-parameter tuning, in particular in choosing the stepsizes. When such a method is properly adaptive, it finds the right stepsize without apriori knowledge based on continuous adjustment per iteration.

The authors construct counterexamples showing that directly applying adaptive methods designed for centralized problems results in non-convergence in distributed setting. The authors then propose an adaptive stepsize tracking protocol that involves transmitting two extra scalar variables to ensure consistency among stepsizes of different nodes. The authors give theoretical guarantees, matching (nearly) the convergence rate in the non-distributed setting. Furthermore, the authors exhibit their method on various real-world datasets to further underline their theoretical findings.

**Strengths:**

* The article is well written: the authors explain the problem clearly and the algorithm is clearly outlined.
* The result and formulation of Theorem 1 are insightful, and the synthetic example of (6) is simple yet very instructive. Together they outline the problem to be overcome in Section 2.2 nicely.
* The authors provide an (almost) rate optimal theoretical guarantee for their proposed method.
* The chosen real-world settings are both interesting and illustrative.
* The proposed method is attractive in terms of communication complexity, with only a couple of scalars being required on top of the gradients.

**Weaknesses:**

* Whilst I think the problem that the authors address is interesting, and within their scope they provide satisfying answers, I also think that this scope is rather limited. There are many things to investigate here in terms of e.g. network topology, communication efficiency, the influence of various noise sources. The authors stay within a set of rather stringent assumptions and do not capture the effect of these problem characteristics fully.

* In high-dimensional settings, which the authors aim to address, achieving zero bias for stochastic gradients is not always feasible. Additionally, confirming the absence of bias in the gradient estimator is challenging to verify in practice. Does the method still work without this assumption? What if the bias is very small?

* Similarly, the uniform bound on the gradient seems rather strong. Is this really required by the analysis?

**Questions:**

Some of the figures are rather difficult to read when printed on paper as they are rather small, but were readable digitally.

* Is there a reason the constant $C$ does not appear in e.g. (13)?

* I do not see the point of Corollary 1. As I understand it, this is an upper bound for D-TiAda. The authors say that this result is provided for "proper comparison". How does Corollary 1 provide any comparison if it is only an upper bound? Isn't Theorem 1 to provide the comparison that the authors are after?

**Limitations:**

Within the scope of the article, I think limitations are appropriately addressed.

---

> ### Author Rebuttal · Authors · 2024-08-06
>
> Thank you for the insightful and valuable comments. Please see below for a detailed point-by-point response.
>
> >__W1:__ Whilst I think the problem that the authors address is interesting, and within their scope they provide satisfying answers, I also think that this scope is rather limited. There are many things to investigate here in terms of e.g. network topology, communication efficiency, the influence of various noise sources. The authors stay within a set of rather stringent assumptions and do not capture the effect of these problem characteristics fully.
>
> __Response:__ First of all, we would like to further clarify the scope and contributions of this work. In this paper, we focus on a nonconvex-strongly-concave distributed minimax problem over networks, which subsumes many applications in machine learning and optimization. To mitigate the challenge of hyper-parameter tuning, we propose the first parameter-agnostic distributed adaptive minimax algorithm, D-AdaST, that efficiently resolves the issue of inconsistent adaptive step sizes with relatively low communication costs, achieving a near-optimal rate. Moreover, the obtained convergence results indeed highlight the dependence on network topology as well as the trade-offs between convergence speed and the length of the transition phase (cf. Remark 5), as demonstrated in the experiments. We concur with the reviewer on the importance of considering communication efficiency and the influence of different noise sources in future work.
>
> Regarding the assumptions used in this work, we will provide a more detailed discussion of the rationale and limitations of the assumptions in the revision. Please also refer to the subsequent responses to specific assumptions.
>
> >__W2:__ In high-dimensional settings, which the authors aim to address, achieving zero bias for stochastic gradients is not always feasible. Additionally, confirming the absence of bias in the gradient estimator is challenging to verify in practice. Does the method still work without this assumption? What if the bias is very small?
>
> __Response:__ We agree with the reviewer that obtaining the unbiased stochastic gradient is not always feasible in practice. Nevertheless, this remains one of the most widely used assumptions in optimization and machine learning when uniform sampling is performed in an independent and identically distributed (IID) manner. Without this assumption, the biased term in the gradient may introduce an extra constant steady-state error in the current convergence result, which cannot be mitigated by employing decreasing stepsizes [1]. Nevertheless, the algorithm still functions and if the bias is small relative to other terms or exhibits a certain structure, e.g., memory-biased (c.f., Definition 11 in [2]), more explicit convergence results can be obtained [2]. We believe that investigating these biased stochastic gradient models represents an interesting direction for future work.
>
> >__W3:__ Similarly, the uniform bound on the gradient seems rather strong. Is this really required by the analysis?.
>
> __Response:__ We remark that the assumption of bounded gradient is essential for the adaptive methods to achieve the property of being parameter-agnostic, thus reducing the burden of hyper-parameter tuning. In our proof, without the assumption, the term $S_1$ in Eq. (18) cannot be proved to be vanishing. As a result, one would need to employ the negative term $-\frac{4}{K}\sum_{k=0}^{K-1}{\mathbb{E} \left[ \|| \nabla_xf\left( \bar{x}_k,\bar{y}_k \right) \|| ^2 \right]}$ to absorb $S_1$ to ensure a sufficient descent. However, this typically imposes specific requirements on step size selection that may depend on the problem-dependent parameters, as also observed in [3], which contradicts the basic principle of a parameter-agnostic method. Indeed, to the best of our knowledge, in the field of distributed nonconvex-strongly-concave minimax optimization, there is no existing parameter-agnostic method that achieves the (near) optimal convergence rate while also eliminating the bounded gradient assumption. On the other hand, as detailed in Remark 3, this assumption is widely used and can be met by imposing constraints on the bounded domain of decision variables in many real-world tasks [4, 5].
>
> >__Q1:__ Some of the figures are rather difficult to read when printed on paper as they are rather small, but were readable digitally.
>
> __Response:__ To express the experimental results more clearly, we will adjust the text size and the thickness of the curves in the figure appropriately.
>
> >__Q2:__ Is there a reason the constant $C$ does not appear in e.g. (13)?
>
> __Response:__ For better readability of the convergence results, we show in the main text only the dependence of the convergence on some key parameters, while hiding other constant parameters such as $C$. The explicit convergence result with dependence on $C$ can be found in (75) in the Appendix. We will present a more detailed convergence result concerning other parameters in the revision.
>
> >__Q3:__ I do not see the point of Corollary 1. As I understand it, this is an upper bound for D-TiAda. The authors say that this result is provided for "proper comparison". How does Corollary 1 provide any comparison if it is only an upper bound? Isn't Theorem 1 to provide the comparison that the authors are after?
>
> __Response:__ Theorem 1 shows that the D-TiAda algorithm does not converge in certain scenarios, while Corollary 1 esentially shows that the underlying reason is due to the inconsistency of the adaptive stepsizes, i.e., $\zeta _{v}^{2}$ and $\zeta _{u}^{2}$ in (14), compared to the result for D-AdaST in Theorem 2. Together with these conclusions, we have demonstrated the impact of the stepsize inconsistency in the distributed adaptive minimax methods, as well as an efficient stepsize tracking mechanism to solve this problem. We will revise this part of the statements accordingly to avoid any possible confusion.

---

> > ### Comment · Reviewer_rQb7 · 2024-08-07
> >
> > I appreciate the detailed response by the authors. I wish to maintain my score.

---

> > > ### Author Response · Authors · 2024-08-09
> > > **Thanks to Reviewer rQb7**
> > >
> > > We appreciate the acknowledgment of the reviewer. Thanks for spending your time carefully reviewing our paper and responding to our clarifications.

---

### Author Rebuttal · Authors · 2024-08-06

__General Response to All Reviewers__

We would like to express our gratitude to  all the reviewers for evaluating our work positively and providing their insightful and valuable comments that have helped us greatly improve the quality of our paper.

We have carefully considered each of the reviewers' concerns and provided a detailed point-by-point response under each rebuttal section. Minor comments aside, our response includes the following major aspects: i) we have further clarified the scope of this work and discussed the rationale behind our assumptions; ii) we have conducted an additional experiment on a more complicated dataset CIFAR-100 (c.f., the attached PDF file in this general response block) and will add more experimental results with additional datasets in the final version; iii) we have provided detailed explanations to clarify certain concepts and statements; iv) we have discussed possible extensions to other general settings of the problem as well as the communication model. Unless otherwise noted, the citations in our response correspond to those in the submitted manuscript.

Below are the references used throughout the rebuttal, which have been properly cited and compared in the main text.

* [1] Hu, B., Seiler, P. and Lessard, L. Analysis of biased stochastic gradient descent using sequential semidefinite programs. Mathematical Programming, 2021.

* [2] Driggs, D., Liang, J. and Schonlieb C. On Biased Stochastic Gradient Estimation. JMLR, 2022.

* [3] Huang, F., Wu, X., and Hu, Z.. Adagda: Faster adaptive gradient descent ascent methods for minimax optimization. AISTATS, 2023.

* [4] Dinh, L., Pascanu, R., Bengio S. and Bengio Y. Sharp minima can generalize for deep nets. ICML 2017.

* [5] Arjovsky, M., Chintala S. and Bottou L. Wasserstein generative adversarial networks. ICML 2017.

* [6] Li, H., Tian, Y., Zhang, J., and Jadbabaie, A. Complexity lower bounds for nonconvex-strongly-concave min-max optimization. NeurIPS, 2021.

* [7] Chen, T., Sun, Y. and Yin W. Closing the gap: Tighter analysis of alternating stochastic gradient methods for bilevel problems. NeurIPS, 2021.

* [8] Li, X., Yang, J., and He, N. Tiada: A time-scale adaptive algorithm for nonconvex minimax optimization. ICLR, 2023.

* [9] Lin, T., Jin, C. and Jordan M. On gradient descent ascent for nonconvex-concave minimax problems. ICML, 2020.

* [10] Yang, J., Li, X., and He, N. Nest your adaptive algorithm for parameter-agnostic nonconvex minimax optimization. NeurIPS, 2022.

* [11] Tsaknakis, I., Hong, M., and Liu, S. Decentralized min-max optimization: Formulations, algorithms and applications in network poisoning attack. ICASSP, 2020.

* [12] Chen, L., Ye, H. and Luo, L. An Efficient Stochastic Algorithm for Decentralized Nonconvex-Strongly-Concave Minimax Optimization. AISTATS, 2024.

* [13] Khaled, A. and Jin, C. Faster federated optimization under second-order similarity. ICLR, 2023.

* [14] Ene, A. and Le Nguyen, H. Adaptive and Universal Algorithms for Variational Inequalities with Optimal Convergence. AAAI, 2022.

* [15] Jin, C., Netrapalli, P. and Jordan, M. What is local optimality in nonconvex-nonconcave minimax optimization? ICML, 2020.

* [16] Yang, J., Kiyavash, N. and He, N. Global convergence and variance-reduced optimization for a class of nonconvex-nonconcave minimax problems. NeurIPS, 2020.

* [17] Tian, Y., Sun, Y. and Scutari, G. Achieving linear convergence in distributed asynchronous multi-agent optimization. IEEE Transactions on Automatic Control, 2020.

---

### Decision · Program_Chairs · 2024-09-25

**Decision:**

Accept (poster)

**Comment:**

This paper introduces an adaptive-stepsize method for distributed minimax optimization. The authors construct counterexamples, showing that directly applying adaptive methods designed for centralized problems results in non-convergence in the distributed setting. Then, they propose an adaptive stepsize tracking protocol that involves transmitting two extra scalar variables to ensure consistency among stepsizes of different nodes. Theoretical guarantees are established, matching (nearly) the convergence rate in the non-distributed setting. This paper is well-written and the contributions are solid. Although the reviewers point several issues that need to be addressed in the future work, the strengths outweigh the weaknesses.